# THE SUPERPOSITION OF DIFFUSION MODELS USING THE ITÔ DENSITY ESTIMATOR

**Marta Skreta**[*,1,2]   **Lazar Atanackovic**[*,1,2]   **Avishek Joey Bose**[3,4]
**Alexander Tong**[4,5]   **Kirill Neklyudov**[4,5,†]
[1]University of Toronto   [2]Vector Institute   [3]University of Oxford
[4]Mila - Quebec AI Institute   [5]Université de Montréal

## ABSTRACT

The Cambrian explosion of easily accessible pre-trained diffusion models suggests a demand for methods that combine multiple different pre-trained diffusion models without incurring the significant computational burden of re-training a larger combined model. In this paper, we cast the problem of combining multiple pre-trained diffusion models at the generation stage under a novel proposed framework termed superposition. Theoretically, we derive superposition from rigorous first principles stemming from the celebrated continuity equation and design two novel algorithms tailor-made for combining diffusion models in SUPERDIFF. SUPERDIFF leverages a new scalable Itô density estimator for the log likelihood of the diffusion SDE which incurs *no additional overhead* compared to the well-known Hutchinson's estimator needed for divergence calculations. We demonstrate that SUPERDIFF is scalable to large pre-trained diffusion models as superposition is performed *solely through composition during inference*, and also enjoys painless implementation as it combines different pre-trained vector fields through an automated re-weighting scheme. Notably, we show that SUPERDIFF is efficient during inference time, and mimics traditional composition operators such as the logical **OR** and the logical **AND**. We empirically demonstrate the utility of using SUPERDIFF for generating more diverse images on CIFAR-10, more faithful prompt conditioned image editing using Stable Diffusion, as well as improved conditional molecule generation and unconditional *de novo* structure design of proteins. https://github.com/necludov/super-diffusion.

## 1  INTRODUCTION

The design and application of generative models at scale are arguably one of the fastest-growing use cases of machine learning, with generational leaps in performance that often exceed expert expectations (Steinhardt, 2022). A few of the central facilitators of this rapid progress are the availability of high-quality training data and large computing hardware (Kaplan et al., 2020); which in tandem provide a tried and trusted recipe to scale generative models in a variety of data modalities such as video generation (Brooks et al., 2024), natural language understanding (OpenAI, 2023; Achiam et al., 2023; Dong et al., 2022), and other challenging domains like mathematical reasoning (Trinh et al., 2024), or code assistance (Bubeck et al., 2023). As a result, it is not surprising that a driving force behind current generative modeling research is centered around developing open-source tooling (Dao et al., 2022; Kwon et al., 2023) to enable further scaling and understanding emergent behavior of such models (Schaeffer et al., 2023), including probing current limitations (Dziri et al., 2024).

Indeed, the rapid escalation of generative model development has also induced a democratizing effect, given the easy access to large pre-trained in the current AI climate (Stability AI, 2023; Midjourney, 2023; Ramesh et al., 2021). Furthermore, with the rise of open-source models, it is now easier than ever to host and deploy fine-tuned models. However, the current pace of progress also makes it infeasible to easily scale further models without confronting practical challenges. For instance, for continuous domains such as natural images current pre-trained diffusion models already exhaust all public data, with a growing proportion of the web already populated with synthetic data (Schuhmann et al., 2022). Compounding these challenges is the tremendous cost of pre-training these large

---

[*]Authors contributed equally.
[†]Correspondence to k.necludov@gmail.com

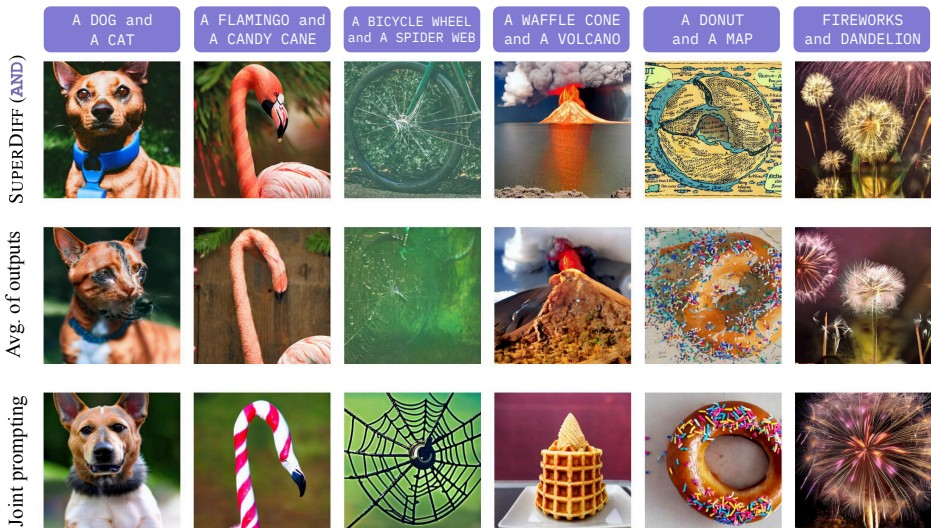

Figure 1: Concept interpolations via different methods: SUPERDIFF (top row), the averaging of outputs with different prompts (middle row), and joint prompting with standard Stable Diffusion (SD) (bottom row) for six different prompt combinations. Here we use SUPERDIFF with the **AND** operation (sampling equal densities).

diffusion models, which at present makes it computationally unattractive for individuals to build large pre-trained models on different datasets without re-training a larger combined model.

A compelling alternative to training ever larger models is to consider the efficacy of maximizing the utility of existing pre-trained models. In particular, it is interesting to consider the compositional benefits of combining pre-trained models at the generation stage in place of training a single mono-lithic model (Du & Kaelbling, 2024). For diffusion models, which are the current de facto modeling paradigm over continuous domains, compositional generation can be framed as modifying the inference mechanism through either guidance (Dhariwal & Nichol, 2021; Ho & Salimans, 2022) or applying complex MCMC correction schemes (Du et al., 2023). Despite advancements, methods that utilize guidance lack a firm theoretical underpinning (Bradley & Nakkiran, 2024) and MCMC tech-niques are prone to scaling issues, and more importantly, remain unproven at combining existing large pre-trained diffusion models. This motivates the following timely research question: ***Can we combine pre-trained diffusion models solely at inference in a theoretically sound and efficient manner?***

**Present work**. In this paper, we cast the problem of combining and composing multiple pre-trained diffusion models under a novel joint inference framework we term superposition. Intuitively we develop our framework by starting from the well-known principle of superposition in physical systems which summarizes the net response to multiple inputs in a linear system as the sum of individual inputs. Applying the superposition principle, even in the simplest case of a mixture model, requires constructing the resultant superimposed vector field; which can be built analytically by reweighting using the marginal density of generated samples along each diffusion model's inference trajectory.

We introduce SUPERDIFF for scalable superposition by proposing a novel estimator of the density of generated samples along an SDE trajectory, which we expect to be of independent interest beyond superposition. In particular, our Itô density estimator in Thm. 1 is general as it is not restricted to the vector field under which the trajectories are generated, it requires *no additional computation during inference*, and, when the ground true scores of the marginals are available, it is exact. This is in stark contrast to all prior density estimation approaches that involve computing expensive and high variance divergence estimates of a drift vector field associated with the probability flow ODE. Armed with our new Itô density estimator we propose to combine pre-trained diffusion models by guiding the joint inference process through fine-grained control of the relative superposition weights of the model outputs. More precisely, in Sec. 3.2, we propose algorithms for two specific instantiations, illustrated in (Fig. 1) of the superposition principle: (i) generating samples from a mixture of densities (an OR operation over models) or (ii) generating samples that are *equally* likely under *all* densities (an AND operation over models).

We test the applicability of our approach SUPERDIFF using the two proposed superimposition strategies for image and protein generation tasks. For images, we first demonstrate the ability to combine the outputs of two models trained on disjoint datasets such that they yield better performance

than the model trained on both of the datasets (see Sec. 4.1). In addition, we demonstrate the ability to interpolate between densities which correspond to concept interpolation in the image space (see Sec. 4.2). For proteins, we demonstrate that combining two different generative models leads to improvements in designability and novelty generation (see Sec. 4.3). Across all our experimental settings, we find that combining pre-trained diffusion models using SUPERDIFF leads to higher fidelity generated samples that better match task specification in text-conditioned image generation and also produce more diverse generated protein structures than comparable composition strategies.

## 2 PRELIMARIES

Generative models learn to approximate a target data distribution $p_{\text{data}} \in \mathbb{P}(\mathbb{R}^d)$ defined over $\mathbb{R}^d$ using a parametric model $q_\theta$ with learnable parameters $\theta$. In the conventional problem definition, the data distribution is realized as an empirical distribution that is provided as a training set of samples $\mathcal{D} = \{\mu^i\}_{i=1}^m$. Whilst there are multiple generative model families to choose from, we restrict our attention to diffusion models (Song et al., 2020; Ho et al., 2020) which are arguably the most popular modeling family driving current application domains. We next review the basics of the continuous time formulation of diffusion models before casting them within our superposition framework.

### 2.1 CONTINUOUS-TIME DIFFUSION MODELS

A diffusion model can be cast as the solution to the Stochastic Differential Equation (Øksendal, 2003),

$$dx_t = f_t(x)dt + g_t dW_t, \quad x_0 \sim q_0(x_0). \tag{1}$$

In the Itô SDE literature, the function $f_t : \mathbb{R}^d \to \mathbb{R}^d$ is known as the drift coefficient while $g_t : \mathbb{R} \to \mathbb{R}$ is a real-valued function called the diffusion coefficient and $W_t$ is the standard Wiener process. The subscript index $t \in [0, 1]$ indicates the time-valued nature of the stochastic process. Specifically, we fix the starting time $t = 0$ to correspond to the data distribution $p_{\text{data}} := q_0(x_0)$ and set $t = 1$ as the terminal time $t = 1$ to an easy to sample prior such as a standard Normal distribution $p_{\text{noise}} := q_1(x_1) = \mathcal{N}(x_1|0, I)$. As such the diffusion SDE, also called the forward process, can be seen as progressively corrupting the data distribution and ultimately hitting a terminal distribution devoid of any structure.

To generate samples from the marginal density $q_t(x_t)$ induced by the diffusion SDE in equation 1 we leverage the reverse-time SDE with the same marginal density as demonstrated below.

**Proposition 1.** [Reverse-time SDEs/ODE] *Marginal densities $q_t(x)$ induced by Eq. (1) correspond to the densities induced by the following SDE that goes back in time ($\tau = 1 - t$) with the corresponding initial condition*

$$dx_\tau = \left(-f_t(x_\tau) + \left(\frac{g_t^2}{2} + \xi_\tau\right)\nabla \log q_t(x_\tau)\right)d\tau + \sqrt{2\xi_\tau}d\overline{W}_\tau, \quad x_{\tau=0} \sim q_1(x_0), \tag{2}$$

*where $\overline{W}_\tau$ is the standard Wiener process in time $\tau$, and $\xi_\tau$ is any positive schedule.*

See proof in App. A.1. The reverse SDE flows backward in time $\tau = 1 - t$ and is linked to the diffusion SDE through the score $\nabla_x \log q_\tau(x)$, and $d\overline{W}_\tau$ is another Weiner process. As a result, a parametric model $\nabla \log q_\tau(x; \theta)$ may directly learn to approximate this score function for every point in time and then draw samples by simulating the reverse SDE in equation 2 by plugging back in the learned score. Notably, for $\xi_t \equiv 0$, the SDE becomes an ODE which defines a smooth change of measure corresponding to $p_{\text{noise}}$ into the measure corresponding to $p_{\text{data}}$.

In practice, for generative modelling, the forward SDE from Eq. (1) is chosen to be so simple that it can be integrated in time analytically without simulating the SDE itself. This is equivalent to choosing the noising schedule first, and then deriving the SDE that corresponds to this schedule. Namely, for every training sample $\mu^i$, we can define the density of the corrupted $\mu^i$ as a normal density with the mean scaled according to $\alpha_t$ and the variance $\sigma_t^2$, then the density of the entire corrupted dataset is simply a mixture over all training samples $\mu^i$, i.e.

$$q_t^i(x) = \mathcal{N}(x \,|\, \alpha_t \mu^i, \sigma_t^2 I), \quad q_t(x) = \frac{1}{N}\sum_{i=1}^N q_t^i(x). \tag{3}$$

Clearly, choosing $\alpha_t, \sigma_t$ such that $\alpha_0 = 1, \sigma_0 = 0$ and $\alpha_1 = 0, \sigma_1 = 1$, we guarantee $p_{\text{data}} := q_0(x_0)$ and $p_{\text{noise}} := q_1(x_1) = \mathcal{N}(x_1|0, I)$. This perturbation of the data distribution using a Gaussian kernel offers specific forms for the drift $f_t$, and diffusion coefficient $g_t$ as described next.

**Proposition 2.** [Ornstein–Uhlenbeck SDE] *The time-dependent densities in Eq. (3) correspond to the marginal densities of the following SDE, with the corresponding initial condition*

$$dx_t = \underbrace{\frac{\partial \log \alpha_t}{\partial t} x_t}_{f_t(x_t)} dt + \underbrace{\sqrt{2\sigma_t^2 \frac{\partial}{\partial t} \log \frac{\sigma_t}{\alpha_t}}}_{g_t} dW_t \,, \quad x_0 \sim q_0(x_0) \,. \tag{4}$$

See proof in App. A.2. We highlight the simplicity of the drift term, a linear scaling, that allows us to simulate efficiently the reverse SDE and is crucial for the proposed density estimators in Sec. 3.1. Altogether, the derivations of this section allow us to go from the noise schedules of samples in Eq. (3) used during the training of a given diffusion model to the corresponding forward SDE in Prop. 2, and finally to the reverse SDEs or ODE used in Prop. 1.

## 2.2 SUPERPOSITION OF ODES AND SDES

In this section, we introduce the superposition of multiple time-dependent densities that correspond to different stochastic processes. A suggestive view of these densities is as processes corresponding to different training data (e.g. different datasets), different conditions (e.g. different text prompts), or simply differently trained diffusion models. Namely, we consider $N$ forward noising process $\{q_t^i(x)\}_{i=1}^N$ that possibly start from different initial distributions $\{q_{t=0}^i(x)\}_{i=1}^N$ (e.g. different datasets). Assume that we know the individual vector fields $v_t^i(x)$ that define the change of corresponding densities $q_t^i(x)$ via the state-space ODEs and continuity equations,

$$\frac{dx_t}{dt} = v_t^i(x_t) \quad \Longrightarrow \quad \frac{\partial}{\partial t} q_t^i(x) = -\langle \nabla_x, q_t^i(x) v_t^i(x) \rangle \,, \quad \forall i \in [N]. \tag{5}$$

The *superposition* of the noising processes $\{q_t^i(x)\}_{i=1}^N$ is the mixture of corresponding densities:

$$q_t^{\mathrm{mix}}(x) := \sum_{j=1}^N \omega^j q_t^j(x), \quad \sum_{i=1}^N \omega^i = 1 \,, \quad \omega^i \geq 0 \,, \tag{6}$$

where $\omega^j$ is a mixing coefficient. Note that superimposed $q_t^{\mathrm{mix}}(x)$ also satisfies the continuity equation for the superposition of the vector fields $v_t^i(x)$ as demonstrated in the following proposition.

**Proposition 3.** [Superposition of ODEs (Liu, 2022)] *The mixture density in Eq. (6) follows the continuity equation with the superposed vector fields from Eq. (5), i.e.*

$$\frac{\partial}{\partial t} q_t^{\mathrm{mix}}(x) = -\langle \nabla_x, q_t^{\mathrm{mix}}(x) v_t(x) \rangle \,, \quad v_t(x) = \frac{1}{\sum_{j=1}^N \omega^j q_t^j(x)} \sum_{i=1}^N \omega^i q_t^i(x) v_t^i(x) \,. \tag{7}$$

We reproduce the proof for Prop. 3 in the context of our superposition in App. A.3. The superposition principle is the core principle that allows for efficient simulation-free learning of the flow-based models (Liu et al., 2022b; Lipman et al., 2022; Albergo et al., 2023) and diffusion models (Song et al., 2020). We discuss how these frameworks are derived from the superposition principle in App. B.

The superposition principle straightforwardly extends to the marginal densities of SDEs. That is, consider marginals densities $q_\tau^i(x)$ generated by the following SDEs

$$dx_\tau = u_\tau^i(x_\tau)d\tau + g_\tau d\overline{W}_\tau \,, \quad x_{\tau=0} = q_{\tau=0}^i(x_0) \,, \tag{8}$$

where one has to note the same diffusion coefficient for all the SDEs. Then the mixture of densities from Eq. (6) can be simulated by the SDE from the following proposition.

**Proposition 4.** [Superposition of SDEs] *The mixture $q_t^{\mathrm{mix}}(x) := \sum_{i=1}^N \omega^i q_t^i(x)$ of density marginals $\{q_t^i(x)\}_{i=1}^N$ induced by SDEs from Eq. (8) corresponds to the following SDE*

$$dx_\tau = u_\tau(x_\tau)d\tau + g_\tau d\overline{W}_\tau \,, \quad u_t(x) = \frac{1}{\sum_{j=1}^N \omega^j q_t^j(x)} \sum_{i=1}^N \omega^i q_t^i(x) u_t^i(x) \,. \tag{9}$$

See App. A.4 for the proof. Both Prop. 3 and Prop. 4 can be easily extended to the families of densities parameterized with a continuous variable (see Theorem 1 in Peluchetti (2023)).

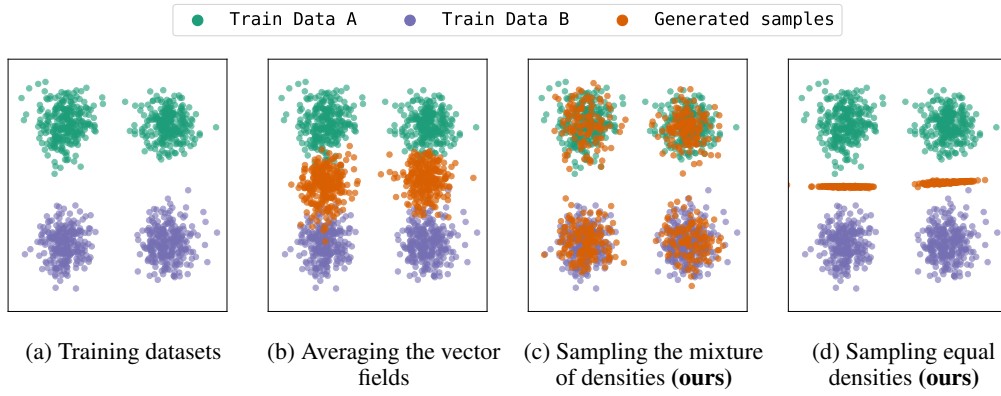

(a) Training datasets     (b) Averaging the vector fields     (c) Sampling the mixture of densities (**ours**)     (d) Sampling equal densities (**ours**)

Figure 2: An intuitive illustration of using model superposition for improving inference performance. We show an example of two disjoint datasets and train a model for each set. Each individual model learns to generate samples only from their respective datasets. Using model superposition enables sampling from both densities.

## 3 SUPERPOSITION OF DIFFERENT MODELS

We now introduce our method for combining pre-trained diffusion models using the principle of superposition. The result of this is a novel inference time algorithm SUPERDIFF which can be easily applied without further fine-tuning or post-processing of any of the pre-trained diffusion weights. Our proposed approach SUPERDIFF can be instantiated in two distinct ways that allow for the composition diffusion models that can be informally interpreted as logical composition operators in the logical `AND` and the logical `OR`. More precisely, given two pre-trained diffusion models that are trained on datasets `A` and `B` inference using SUPERDIFF can be done by either sampling from the mixture of the two learned densities, i.e. logical `AND` Fig. 2c, or sampling from the equal density locus (logical `OR` Fig. 2d). In such a manner, superposition using `AND` leads to generated samples that are equally likely under both pre-trained diffusion models while superposition using `OR` creates samples that are preferentially generated by either pre-trained model—and thus mimics the empirical distributions `A` or `B`.

**Method overview**. SUPERDIFF is applicable in settings where the modeler has access to $M$ pre-trained diffusion models, along with its learned score function $\nabla_x \log q_t^i(x)$. Each of the pre-trained models follows a marginal density $q_t^i(x)$ and as a result must admit a corresponding vector field $v_t^i(x)$ that satisfies the continuity equation in Equation 5. The key idea of our approach is an adaptive re-weighting scheme of the pre-trained model's vector fields that relies on the likelihood of a sample under different models. A naive approach to estimating each marginal density during generation immediately presents several technical challenges as it requires the estimation of the divergence of superimposed vector fields. In particular, these challenges can be stated as follows:

**(C1)** The marginal superpositioned vector field differs from the vector fields of either of the models.
**(C2)** The divergence operation requires backpropagation through the network and is computationally expensive even with Hutchinson's trace estimator (Hutchinson, 1989).

Our proposed approach SUPERDIFF overcomes these computational challenges by introducing a novel density estimator in Sec. 3.1. Crucially, this new estimator does not require divergence estimation and enjoys having the same variance as the computationally expensive Hutchinson's trace estimator, making it a favorable choice when generating using large pre-trained diffusion models. In section Sec. 3.2 we exploit this new density estimator to formally present our algorithm SUPERDIFF and derive connections to the composition operators that intuitively resemble logical `AND` and `OR`.

### 3.1 EVALUATING THE DENSITIES ON THE FLY

In this section, we introduce a novel method for evaluating the marginal density of a diffusion model during the inference process. The conventional way to evaluate the density uses the continuity equation and solves the same ODE that is used for generating samples. This, however, is not easily possible in the case of our superposition of vector fields framework as outlined in Prop. 3. To solve this, we present the following proposition that disentangles the vector field generating the sample ($u_t(x)$ in the proposition) and the vector fields corresponding to different generative models $v_t^i(x)$.

**Algorithm 1:** SUPERDIFF pseudocode (for `OR` and `AND` operations)

---

**Input :** $M$ pre-trained score models $\nabla_x \log q_t^i(x)$, the parameters of the schedule $\alpha_t, \sigma_t$, stepsize $d\tau > 0$, temperature parameter $T$, bias parameter $\ell$, and initial noise $z \sim \mathcal{N}(\mathbf{0}, \mathbf{I})$.

**for** $\tau = 0, \ldots, 1$ **do**

$\quad t = 1 - \tau, \ \varepsilon \sim \mathcal{N}(\mathbf{0}, \mathbf{I})$

$\quad \kappa_\tau^i \leftarrow \begin{cases} \texttt{softmax}(T \log q_t^i(x_\tau) + \ell) \ // \ \text{\small for OR according to Prop. 3} \\ \texttt{solve Linear Equations} \ // \ \text{\small for AND according to Prop. 6} \end{cases}$

$\quad u_t(x_\tau) \leftarrow \sum_{i=1}^M \kappa_\tau^i \nabla \log q_t^i(x_\tau)$

$\quad dx_\tau \leftarrow \left(-f_{1-\tau}(x_\tau) + g_{1-\tau}^2 u_t(x_\tau)\right) d\tau + g_{1-\tau} d\overline{W}_\tau \ // \ \text{\small using Prop. 1}$

$\quad x_{\tau+d\tau} \leftarrow x_\tau + dx_\tau$

$\quad d \log q_{1-\tau}(x_\tau) = \langle dx_\tau, \nabla \log q_{1-\tau}(x_\tau) \rangle + \Big( \langle \nabla, f_{1-\tau}(x_\tau) \rangle +$

$\quad\quad\quad\quad + \Big\langle f_{1-\tau}(x_\tau) - \frac{g_{1-\tau}^2}{2} \nabla \log q_{1-\tau}(x_\tau), \nabla \log q_{1-\tau}(x_\tau) \Big\rangle \Big) d\tau \ // \ \text{\small using Thm. 1}$

**return** $x_\tau$

---

**Proposition 5.** [Smooth density estimator] *For the integral curve $x(t)$ solving $dx/dt = u_t(x_t)$, and the density $q_t^i(x(t))$ satisfying the continuity equation $\frac{\partial}{\partial t} q_t^i(x) = -\langle \nabla_x, q_t^i(x) v_t^i(x) \rangle$, the log-density along the curve changes according to the following ODE*

$$\frac{d}{dt} \log q_t^i(x(t)) = -\langle \nabla_x, v_t^i(x) \rangle - \langle \nabla_x \log q_t^i(x), v_t^i(x) - u_t(x) \rangle. \tag{10}$$

See proof in App. C. We use this proposition for our experiments conducted in Sec. 4.1. However, as outlined previously, evaluating the marginal density via the continuity equation is restricted to small-scale models due to the computational challenges associated with efficiently estimating the divergence of the associated vector field. As a result, a common line of attack assumes constructing a stochastic unbiased estimator that trades for increased speed by introducing a bit of variance into the divergence estimate. This approach is known as Hutchinson's estimator and requires computing a Jacobian-vector product at every step of the inference.

Instead, we propose a new way to estimate density that allows for efficient computation while integrating the backward SDE from Prop. 1 with a specific choice of the diffusion coefficient.

**Theorem 1.** [Itô density estimator] *Consider time-dependent density $q_t(x)$ induced by the marginals of the following SDE*

$$dx_t = f_t(x_t) dt + g_t dW_t, \ \ x_{t=0} \sim q_0(x), \ \ t \in [0, 1], \tag{11}$$

*where $dW_t$ is the Wiener process. For the reverse-time ($\tau = 1 - t$) SDEs with **any** vector field $u_\tau$ and the same diffusion coefficient $g_t$, i.e.*

$$dx_\tau = u_\tau(x_\tau) d\tau + g_{1-\tau} d\overline{W}_\tau, \ \ \tau \in [0, 1], \tag{12}$$

*the change of the log-density $\log q_\tau(x_\tau)$ follows the following SDE*

$$d \log q_{1-\tau}(x_\tau) = \langle dx_\tau, \nabla \log q_{1-\tau}(x_\tau) \rangle + \Big( \langle \nabla, f_{1-\tau}(x_\tau) \rangle +$$
$$+ \Big\langle f_{1-\tau}(x_\tau) - \frac{g_{1-\tau}^2}{2} \nabla \log q_{1-\tau}(x_\tau), \nabla \log q_{1-\tau}(x_\tau) \Big\rangle \Big) d\tau. \tag{13}$$

We provide the proof in App. C. Notably, the SDE Eq. (13) that keeps track of the change of log-density includes only the divergence of the forward SDE drift $\langle \nabla, f_{1-\tau}(x_\tau) \rangle$. However, in practice, when using the Ornstein-Uhlenbeck SDE, this divergence is simply a constant due to a linear drift scaling. In App. D, we derive the same estimator but in discrete time using the detailed balance condition.

## 3.2 SUPERDIFF: SUPERPOSING PRE-TRAINED DIFFUSION MODELS

**Mixture of densities (logical `OR`).** For a mixture of the densities, superposition of the models follows directly from the propositions in Sec. 2.2. That is, for every $q_t^i(x)$, we assume that it can be generated from SDEs or an ODE from Prop. 1 and we assume the knowledge of scores $\nabla \log q_t^i(x)$.

Then, for the ODE simulation, according to Prop. 3, we can sample from the mixture of densities $q_t^{\text{mix}}(x) := 1/M \sum_{i=1}^{M} q_t^i(x)$ using the following vector field:

$$v_\tau(x) = -f_{1-\tau}(x) + \frac{g_{1-\tau}^2}{2} \sum_{i=1}^{M} \frac{q_t^i(x)}{\sum_j q_t^j(x)} \nabla \log q_{1-\tau}^i(x), \tag{14}$$

starting from samples $x_{\tau=0} \sim q_1(x_0)$. The densities are estimated along the trajectory using Prop. 5.

We highlight this analogously applies for simulation using the SDE $dx_\tau = u_\tau(x_\tau)d\tau + g_{1-\tau}d\overline{W}_\tau$. Namely, according to Prop. 4, we use the following vector field:

$$u_\tau(x) = -f_{1-\tau}(x) + g_{1-\tau}^2 \sum_{i=1}^{M} \frac{q_t^i(x)}{\sum_j q_t^j(x)} \nabla \log q_{1-\tau}^i(x), \tag{15}$$

starting from the samples $x_{\tau=0} \sim q_1(x_0)$. The densities are estimated along the trajectory using Thm. 1. We provide the pseudocode in Algorithm 1.

**Sampling equal densities (logical AND).** To produce the sample that have equal densities under different models we rely on our formula for the density update (see Thm. 1) to find the optimal weights for the vector fields. Indeed, for $M$ diffusion models, we have a system of $M$ equations: the equal change of density for every model and the normalization constraint for model weights, which is a linear system w.r.t. vector field weights as we show in the following proposition.

**Proposition 6.** [Density control] *For the SDE*

$$dx_\tau = \sum_{j=1}^{M} \kappa_j u_\tau^j(x_\tau)d\tau + g_{1-\tau}d\overline{W}_\tau, \tag{16}$$

*where $\kappa$ are the weights of different models and $\sum_j \kappa_j = 1$, one can find $\kappa$ that satisfies*

$$d \log q_{1-\tau}^i(x_\tau) = d \log q_{1-\tau}^j(x_\tau), \quad \forall \ i,j \in [M], \tag{17}$$

*by solving a system of $M + 1$ linear equations w.r.t. $\kappa$.*

We provide the proof and the formulas for the system of linear equations in App. C.1. This proposition allows us to find $\kappa$ that controls the densities to stay the same for all the models as described in Algorithm 1. This approach can also be straightforwardly extended to the case of diffusion models for satisfying different prescribed density ratios, i.e.

$$d \log q_{1-\tau}^1(x_\tau) = d \log q_{1-\tau}^i(x_\tau) + \ell^i. \tag{18}$$

## 4 EXPERIMENTS

### 4.1 JOINING MODELS WITH DISJOINT TRAINING DATA

We validate the proposed Algorithm 1 for the generation of the mixture of the distributions (**OR** setting). We split CIFAR-10 into two disjoint training sets of equal size (first 5 labels and last 5 labels), train two diffusion models on each part, and generate the samples jointly using both models. Namely, the stochastic inference is the **OR** implementation of Algorithm 1, whereas the deterministic setting is the integration of the ODEs (see Prop. 1) and the estimation of the log-density according to Prop. 5. For the choice of hyperparameters, architecture, and data preprocessing, we follow (Song et al., 2020).

In Table 1, we demonstrate that the performance of SUPERDIFF drastically outperforms the performance of the individual models and performs even better than the model trained on the union of both parts of the dataset. For comparison, we evaluate conventional image quality metrics: Fréchet Inception Distance (FID), Inception Score (IS), and Feature Likelihood Divergence (FLD) (Jiralerspong et al., 2023), which takes into account the generalization abilities of the model.

### 4.2 CONCEPT INTERPOLATION WITH SUPERDIFF (AND) AND STABLE DIFFUSION

Next, we evaluate the ability of SUPERDIFF to interpolate different concepts using Stable Diffusion (SD) v1-4 (logical **AND**). In this setup, we generate an image from SD by conditioning it on a prompt using classifier-free guidance. We define two models using two separate prompts that represent concepts — e.g. `"a sunflower"` and `"a lemon"`. We can thus consider each prompt-conditioned model as a separate diffusion model and apply Algorithm 1 to generate images sampled with

Table 1: Unconditional image generation performance for CIFAR-10 with models trained on two disjoint partitions of the training data (labeled A and B). We compare SUPERDIFF (**OR**) with the respective models with the model that is trained on the full dataset (model$_{A \cup B}$) and random choice between two models (model$_{A \text{ OR } B}$).

| | ODE inference | | | SDE inference | | |
|---|---|---|---|---|---|---|
| | FID ($\downarrow$) | IS ($\uparrow$) | FLD ($\downarrow$) | FID ($\downarrow$) | IS ($\uparrow$) | FLD ($\downarrow$) |
| model$_A$ | 14.00 | 8.73 | $15.50 \pm 0.17$ | 15.33 | 7.98 | $15.47 \pm 0.18$ |
| model$_B$ | 13.09 | 7.89 | $18.90 \pm 0.18$ | 13.50 | 7.98 | $18.54 \pm 0.23$ |
| model$_{A \cup B}$ | 6.00 | 8.95 | $8.06 \pm 0.12$ | **3.50** | 9.14 | $7.51 \pm 0.11$ |
| model$_{A \text{ OR } B}$ | 4.28 | 9.14 | $5.96 \pm 0.11$ | 3.99 | 9.36 | $5.29 \pm 0.14$ |
| SUPERDIFF (**OR**) | 4.41 | 9.12 | $6.10 \pm 0.11$ | 4.00 | 9.36 | $5.33 \pm 0.05$ |
| SUPERDIFF$_{T=100}$ (**OR**) | **4.11** | **9.21** | $\mathbf{5.89 \pm 0.11}$ | 4.00 | **9.48** | $\mathbf{5.20 \pm 0.11}$ |

Table 2: Quantatative evaluation of SD-generated images. We compare SUPERDIFF (**AND**), joint prompting, and averaging of outputs for two concept prompts. We report the average minimum CLIP, ImageReward, and TIFA scores of each prompt pair, which gives a measure of how well both concepts are represented.

| | Min. CLIP($\uparrow$) | Min. ImageReward ($\uparrow$) | Min. TIFA ($\uparrow$) |
|---|---|---|---|
| Joint prompting | 23.87 | $-1.62$ | 27.58 |
| Average of scores (Liu et al., 2022a) | 24.23 | $-1.57$ | 32.48 |
| SUPERDIFF (**AND**) | **24.79** | $\mathbf{-1.39}$ | **39.92** |

proportionally equal likelihood with respect to both prompt-conditioned models.[1] We generate 20 images for 20 different concept pairs (tasks) for our model and baselines (c.f. App. J).[2]

**Baselines for concept interpolation**. We consider two approaches for composing images as baselines. The first is simple averaging of SD outputs based on the approach in Liu et al. (2022a); we set $\kappa = 0.5$. The second guides SD generation with a single joint prompt. The prompts are constructed by joining two input concepts with the linking term `"that looks like"` — e.g. `"a sunflower that looks like a lemon"`. For fairness, we also flip the order of the prompt and keep the image with the higher score for all metrics listed below (Luo et al., 2024).

**SUPERDIFF qualitatively generates images with better concept interpolations**. We plot sample generated images for SUPERDIFF (**AND**) in Fig. 1. For the complete set of generated images, see App. J (Figs. A5–A24). We observe that SUPERDIFF (**AND**) can interpolate concepts while also maintaining high perceptual quality. In contrast, the averaging baseline either fails to interpolate concepts from both prompts fully or yields images with lower perceptual quality. We observe that SD using a single prompt struggles to interpolate both concepts.

**SUPERDIFF outperforms baselines on three image evaluation metrics**. To quantitatively evaluate the generated images, we consider three metrics: CLIP Score (Radford et al., 2021), ImageReward (Xu et al., 2024), and TIFA (Hu et al., 2023). CLIP Score measures the cosine similarity between an image embedding and text prompt embedding. ImageReward evaluates generated images by assigning a score that reflects how closely they align with human preferences. TIFA generates several question-answer pairs using a Large Language Model for a given prompt and assigns a score by answering the questions based on the image with a visual question-answering model. We report these metrics for SU-PERDIFF and all baselines in Table 2. For the logical **AND** setting, we evaluate the image against each concept prompt separately (i.e., `"a sunflower"` as one prompt and `"a lemon"` as the other) and take the minimum score for each metric, which measures how well *both* concepts are represented. We find that SUPERDIFF (**AND**) obtains the highest scores across all metrics, indicating that our method can better represent both concepts in the images, while the baseline methods typically only represent one concept or (especially in the case of averaging outputs), generate compositions with lower fidelity.

### 4.3 PROTEIN GENERATION

Next, we apply our method in the setting of unconditional *de novo* protein generation. Protein generation has critical implications in drug discovery (Abramson et al., 2024). A good understanding of the protein landscape is important to rationally find novel proteins. We evaluate proteins generated by the superposition of two existing protein diffusion models, Proteus (Wang et al., 2024a) and FrameDiff (Yim et al., 2023b). We report the results of our best model in Table 3, as well as results for each model individually and simply averaging them.

---

[1]In Table 2, we report results for inference only using SDEs. For the ODE setting, evaluating densities takes about $1082 \pm 7$ seconds per image with SD, whereas it only takes $209 \pm 2$ with our density estimator in the SDE setting (over a 5-fold decrease!), while also requiring almost 30% less memory.

[2]SUPERDIFF (**AND**) SD v1-4: https://huggingface.co/superdiff/superdiff-sd-v1-4

Table 3: Evaluation of SUPERDIFF and baseline models, Proteus & FrameDiff, for unconditional protein generation. We show results for three categories of metrics: designability, novelty, and diversity. We also include a baseline of simple averaging of scores ($\kappa = 0.5$). We use the parameter $\ell$ (see Eq. (18)) to control and bias model superposition towards Proteus, i.e. ($\ell > 0$) or FrameDiff ($\ell < 0$). We use temperature values $T = 1$ for all variants of SUPERDIFF. For each type of model composition (averaging or SUPERDIFF), we mark each metric with a (†) if it is better than both Proteus and FrameDiff on their own, and with a (⋆) if it better than either one of them.

| | Designability | | Novelty | | | Diversity | | |
|---|---|---|---|---|---|---|---|---|
| | < 2Å scRMSD (↑) | scRMSD (↓) | < 0.5 scTM (↑) | < 0.3 scTM (↑) | Max. scTM (↓) | Frac. $\beta$ (↑) | Pairwise scTM (↓) | Max. cluster (↑) |
| FrameDiff | $0.392 \pm 0.03$ | $4.315 \pm 0.25$ | $0.152 \pm 0.02$ | $0.016 \pm 0.01$ | $0.570 \pm 0.02$ | $\mathbf{0.175 \pm 0.01}$ | $0.337 \pm 0.02$ | $\mathbf{0.326 \pm 0.05}$ |
| Proteus | $0.928 \pm 0.02$ | $1.014 \pm 0.07$ | $0.360 \pm 0.03$ | $0.020 \pm 0.01$ | $0.536 \pm 0.01$ | $0.119 \pm 0.01$ | $0.312 \pm 0.01$ | $0.217 \pm 0.02$ |
| Average of scores (Liu et al., 2022a) | $0.740 \pm 0.03^\star$ | $1.960 \pm 0.14^\star$ | $0.360 \pm 0.03^\star$ | $0.024 \pm 0.01^\dagger$ | $\mathbf{0.511 \pm 0.01}^\dagger$ | $0.139 \pm 0.01^\star$ | $0.310 \pm 0.01^\dagger$ | $0.253 \pm 0.01^\star$ |
| SUPERDIFF$_{\ell=0}$(OR) | $0.752 \pm 0.03^\star$ | $1.940 \pm 0.14^\star$ | $0.276 \pm 0.03^\star$ | $0.008 \pm 0.01$ | $0.547 \pm 0.01^\star$ | $0.147 \pm 0.01^\star$ | $\mathbf{0.309 \pm 0.02}^\dagger$ | $0.268 \pm 0.02^\star$ |
| SUPERDIFF$_{\ell=1}$(OR) | $\mathbf{0.976 \pm 0.01}^\dagger$ | $\mathbf{0.929 \pm 0.05}^\dagger$ | $\mathbf{0.396 \pm 0.03}^\dagger$ | $0.024 \pm 0.01^\dagger$ | $0.528 \pm 0.01^\dagger$ | $0.127 \pm 0.01^\star$ | $\mathbf{0.307 \pm 0.02}^\dagger$ | $0.246 \pm 0.03^\star$ |
| SUPERDIFF$_{\ell=0}$(AND) | $0.752 \pm 0.03^\star$ | $2.079 \pm 0.16^\star$ | $0.296 \pm 0.03^\star$ | $\mathbf{0.040 \pm 0.01}^\dagger$ | $0.521 \pm 0.01^\dagger$ | $0.141 \pm 0.01^\star$ | $0.306 \pm 0.01^\dagger$ | $0.256 \pm 0.01^\star$ |

**Metrics for evaluations.** We consider *designability*, *novelty*, and *diversity* metrics for evaluation of unconditional generation of protein structures. Protein *designability* refers to the in-silico agreement between generated structures and refolded structures as computed using a purpose-built folding model e.g. ESMFold (Lin et al., 2022), which is positively correlated with the synthesizability of the protein in a wet-lab setting. Generally, if the root-mean-square distance between the generated and refolded proteins (scRMSD) is less than 2Å, it is considered to be designable. We compute several *novelty* metrics based on the similarity of generated proteins to those from the set of known proteins (the training set),

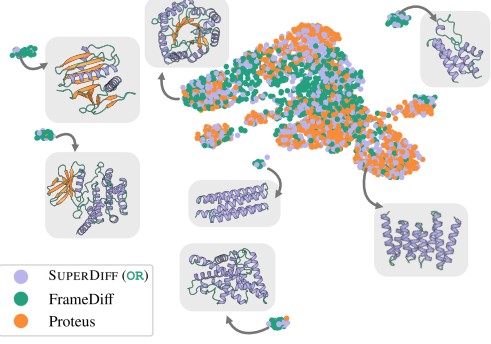

Figure 3: UMAP visualization of protein structures showing cluster archetypes where structure diversity is maintained with SUPERDIFF $_{\ell=0}$ (OR).

which are called scTM scores; the lower the score, the less similar the generated protein is to the training data. Lastly, we use *diversity* to assess the degree of heterogeneity present in the set of generated proteins. This is done by clustering the generated proteins in terms of sequence overlap, and measuring the fraction of proteins with challenging-to-generate secondary structures such as $\beta$ sheets (Bose et al., 2024). We provide details of all metrics in this section in App. H.2. All results are averages over 50 generated proteins at lengths $\{100, 150, 200, 250, 300\}$ for 500 timesteps.

**SUPERDIFF improves structure generation.** By combining two protein diffusion models, SUPERDIFF is able to outperform both of them. This is somewhat surprising as FrameDiff is substantially less designable than Proteus (0.392 vs. 0.928 scRMSD) (see Table 3). Nevertheless, by using SUPERDIFF (OR), we can increase designability, novelty, and maintain diversity, outperforming simple averaging. We further investigate the composition made by SUPERDIFF (OR) in Fig. 3. Here, we see a few modes (particularly on the plot's left-hand side) that Proteus does not generate; SUPERDIFF (OR) can maintain knowledge of these clusters (although to a lesser extent) while maintaining designability.

We also find that SUPERDIFF (AND) outperforms averaging in designability and diversity. Perhaps more impressively, SUPERDIFF (AND) can generate the most proteins that are furthest away from the set of known proteins (scTM score < 0.3) by almost two times more than the next best method. We visualize these proteins in Fig. A2 and explore composition in Fig. A1. This motivates the utility of applying our method in novel discovery settings.

## 4.4 SMALL MOLECULE GENERATION

We evaluate SUPERDIFF (AND) for multi-property molecule generation, where candidates must satisfy multiple constraints (e.g., binding to a target protein, non-toxicity, and bioavailability). Conventional approaches filter molecules post-hoc or optimize properties separately (Schneuing et al., 2024), but SUPERDIFF (AND) enables direct control. We investigate molecule generation with the following pairs of property pairs: (1) high binding score to the protein GSK3$\beta$ and high drug-likeness (QED) (Bickerton et al., 2012) and (2) high binding scores to the proteins GSK3$\beta$ and JNK3 (an example of dual-target drug design (Jin et al., 2020; Chen et al., 2024)). We generate molecules using LDMol (Chang & Ye, 2024), a latent diffusion model that generates a SMILES string (Weininger, 1988) conditioned on a text prompt. Again, we take each prompt-conditioned model as a separate diffusion model. Following Wang et al. (2024b), we use the prompts `"This molecule inhibits {GSK3B/JNK3}"` and `"This molecule looks like a drug"`.

Table 4: Multi-property molecule generation results. For a set of two target properties ($P_1$ and $P_2$), we report the average top-10 and top-1 best performing molecules (for the joint methods, "best" is taken as the largest $P_1*P_2$ scores). We also report the average minimum scores of each property pair, as well as the diversity, validity & uniqueness (V&U), and quality of all molecules. Metrics are taken from 5 runs of batch-size 512.

| | $P_1$/$P_2$ | Top-10 (↑) | | | | Top-1 (↑) | | All | | |
| --- | --- | --- | --- | --- | --- | --- | --- | --- | --- | --- |
| | | ($P_1*P_2$) | min($P_1$,$P_2$) | $P_1$ | $P_2$ | $P_1$ | $P_2$ | Div. (↑) | V&U (↑) | Qual. (↑) |
| $P_1$ only | | $0.107_{\pm0.024}$ | $0.240_{\pm0.049}$ | $\mathbf{0.411}_{\pm\mathbf{0.034}}$ | $0.266_{\pm0.058}$ | $\mathbf{0.550}_{\pm\mathbf{0.037}}$ | $0.258_{\pm0.085}$ | $0.907$ | $0.742$ | $0.07$ |
| $P_2$ only | | $0.030_{\pm0.010}$ | $0.033_{\pm0.011}$ | $0.033_{\pm0.011}$ | $\mathbf{0.884}_{\pm\mathbf{0.008}}$ | $0.032_{\pm0.029}$ | $\mathbf{0.919}_{\pm\mathbf{0.008}}$ | $\mathbf{0.921}$ | $\mathbf{0.830}$ | $\mathbf{0.191}$ |
| Joint prompting | GSK3$\beta$ | $0.171_{\pm0.029}$ | $\mathbf{0.273}_{\pm\mathbf{0.034}}$ | $0.287_{\pm0.040}$ | $0.631_{\pm0.050}$ | $0.338_{\pm0.106}$ | $0.743_{\pm0.102}$ | $0.901$ | $0.741$ | $0.114$ |
| Avg. of scores (Liu et al., 2022a) | QED | $0.154_{\pm0.012}$ | $0.269_{\pm0.023}$ | $0.287_{\pm0.014}$ | $0.580_{\pm0.020}$ | $0.392_{\pm0.043}$ | $0.566_{\pm0.114}$ | $0.916$ | $0.773$ | $0.137$ |
| SUPERDIFF (AND) | | $\mathbf{0.177}_{\pm\mathbf{0.024}}$ | $\mathbf{0.273}_{\pm\mathbf{0.022}}$ | $0.277_{\pm0.024}$ | $0.668_{\pm0.027}$ | $0.440_{\pm0.064}$ | $0.680_{\pm0.087}$ | $0.914$ | $0.783$ | $0.141$ |
| $P_1$ only | | $0.057_{\pm0.007}$ | $0.135_{\pm0.011}$ | $\mathbf{0.410}_{\pm\mathbf{0.034}}$ | $0.136_{\pm0.011}$ | $\mathbf{0.550}_{\pm\mathbf{0.037}}$ | $0.208_{\pm0.087}$ | $0.907$ | $\mathbf{0.741}$ | $0.070$ |
| $P_2$ only | | $0.056_{\pm0.011}$ | $0.183_{\pm0.014}$ | $0.275_{\pm0.048}$ | $0.195_{\pm0.009}$ | $0.340_{\pm0.187}$ | $0.326_{\pm0.053}$ | $\mathbf{0.910}$ | $0.690$ | $0.082$ |
| Joint prompting | GSK3$\beta$ | $0.071_{\pm0.022}$ | $0.186_{\pm0.045}$ | $0.367_{\pm0.028}$ | $0.187_{\pm0.046}$ | $0.534_{\pm0.108}$ | $0.296_{\pm0.126}$ | $0.910$ | $0.714$ | $\mathbf{0.098}$ |
| Avg. of scores (Liu et al., 2022a) | JNK3 | $0.073_{\pm0.013}$ | $0.199_{\pm0.012}$ | $0.351_{\pm0.037}$ | $0.210_{\pm0.012}$ | $0.446_{\pm0.124}$ | $0.348_{\pm0.099}$ | $0.909$ | $0.702$ | $0.077$ |
| SUPERDIFF (AND) | | $\mathbf{0.080}_{\pm\mathbf{0.025}}$ | $\mathbf{0.209}_{\pm\mathbf{0.035}}$ | $0.361_{\pm0.060}$ | $\mathbf{0.214}_{\pm\mathbf{0.038}}$ | $0.466_{\pm0.139}$ | $\mathbf{0.360}_{\pm\mathbf{0.104}}$ | $0.908$ | $0.726$ | $0.088$ |

**Evaluation metrics**. We evaluate properties using oracles from Therpeutic Data Commons (Huang et al., 2021); we also compute diversity (Tanimoto distance on Morgan fingerprints (Rogers & Hahn, 2010)) and quality (Lee et al., 2025), which is the fraction of molecules that are valid, unique, have QED $\geq 0.6$ and synthetic accessibility (SA) $\leq 4.0$ (Ertl & Schuffenhauer, 2009).

**SUPERDIFF improves multi-property molecule generation**. In Table 4, we evaluate molecules generated by prompting for each for property separately ($P_1$/$P_2$ only), taking the average of scores, and with SUPERDIFF (AND). We also compare to joint prompting, where we generate using a prompt of both properties: `"This molecule {prop_1} and {prop_2}"`. We create a prompt for both property orderings and randomly sample 50% of the final molecules from each ordering to evaluate the same number of molecules. We find that SUPERDIFF (AND) is able to generate molecules that have higher products of predicted properties compared with averaging scores and joint prompting. SUPERDIFF (AND) also has a higher average minimum score for each property pair, indicating that the molecules better satisfy *both* properties simultaneously. We visualize the best molecules in App. I.

## 5 RELATED WORK

**Compositional diffusion models**. Combining multiple density models into one model with better properties is a classical subject in machine learning (Hinton, 2002). For diffusion models, the most straightforward combination is via averaging their respective learned score functions (Liu et al., 2022a; Kong et al., 2024), which can also be viewed as a form of guidance (Ho et al., 2020; Dhariwal & Nichol, 2021; Bansal et al., 2023). Another way to combine diffusion models comes from their connections to energy-based models (Du et al., 2023; 2020; Nie et al., 2021; Ajay et al., 2024). This, however, comes under a strong assumption that the marginal densities of the noising process are given, which is not the case in most of the modern applications (e.g. (Rombach et al., 2022)); our method resolves this. Density estimation, also, is the main bottleneck for the continual learning of diffusion models, e.g. Golatkar et al. (2023) proposes to learn a separate model for the densities in order to simulate the vector fields from Prop. 3. Finally, Zhong et al. (2024); Biggs et al. (2024) propose to combine models by averaging their weights which, however, does not allow for merging of models with different architectures or different conditions—both resolved by SUPERDIFF in Sec. 4.3 and Sec. 4.2 respectively.

The concurrent work (Karczewski et al., 2024) independently proposed the smooth density estimator (Prop. 5) and an estimator similar to the Itô density estimator (Thm. 1). However, for the latter, the authors do not consider other types of dynamics besides the reverse-time SDEs from Prop. 1. For additional related works regarding protein generation, see App. E.

## 6 CONCLUSION

Despite the ubiquity of diffusion models, many possible ways of performing generation remain unexplored, with classifier-free guidance being the only practical option (Ho & Salimans, 2022). In this paper, we address this shortcoming by proposing two novel methods for combining different models (or the same model with different condition variables) using SUPERDIFF for joint generation.

**Limitations**. While computationally efficient SUPERDIFF is still limited by the computational budget required to produce the outputs of each model. In particular, the combination at the level of model outputs cannot be simply done via cheap combinations of pre-trained model weights. This, however, invites us to develop a more principled—architecture and training-agnostic—method that does not require any assumptions prior assumptions which makes SUPERDIFF a general purpose method.

**Future Work**. Our method unlocks novel research directions by allowing for the principled generation of novel samples that were not previously possible to generate easily Sec. 4.2. Furthermore, we argue that the proposed way to estimate the density during the generation enables numerous new potential ways to control the generation process by providing information about the likelihood.

## REPRODUCIBILITY STATEMENT

To facilitate reproducibility of our empirical results and findings, we intend to make our code publicly available in the final version. We describe all mathematical and algorithmic details necessary to reproduce our results throughout this paper. In Sec. 3 we outline the theoretical basis and mathematical framework for our method. Furthermore, we provide pseudocode for our method in Algorithm 1. For our theoretical contributions, we provide detailed proofs for all theorems and propositions in App. A, App. B and App. C. We provide experimental details for the CIFAR-10 image experiment results in Sec. 4.1. Details regarding experiments for concept interpolation via Stable Diffusion are discussed in Sec. 4.2 and App. J. Experimental details for unconditional protein generation are described in Sec. 4.3 and App. H.1.

## ACKNOWLEDGMENTS

The authors thank Rob Brekelmans, Francisco Vargas, and the anonymous reviewers for providing the feedback significantly improving the paper. The authors thank Viktor Ohanesian for implementing the algorithm for Stable Diffusion XL https://huggingface.co/superdiff/superdiff-sdxl-v1-0. The research was enabled in part by by the Province of Ontario and companies sponsoring the Vector Institute (http://vectorinstitute.ai/partners/), the computational resources provided by the Digital Research Alliance of Canada (https://alliancecan.ca), and Mila (https://mila.quebec). KN is supported by IVADO. AJB is partially supported by an NSERC Post-doc fellowship. This research is partially supported by EPSRC Turing AI World-Leading Research Fellowship No. EP/X040062/1.

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

# APPENDIX

## A PROOFS FOR SEC. 2

### A.1 PROOF OF PROP. 1

**Proposition 1.** [Reverse-time SDEs/ODE] *Marginal densities $q_t(x)$ induced by Eq. (1) correspond to the densities induced by the following SDE that goes back in time ($\tau = 1 - t$) with the corresponding initial condition*

$$dx_\tau = \left(-f_t(x_\tau) + \left(\frac{g_t^2}{2} + \xi_\tau\right)\nabla \log q_t(x_\tau)\right)d\tau + \sqrt{2\xi_\tau}d\overline{W}_\tau, \quad x_{\tau=0} \sim q_1(x_0), \quad (2)$$

*where $\overline{W}_\tau$ is the standard Wiener process in time $\tau$, and $\xi_\tau$ is any positive schedule.*

For the forward SDE we can write the corresponding Fokker-Planck equation

$$\frac{\partial}{\partial t}q_t(x) = -\langle\nabla, q_t(x)f_t(x)\rangle + \frac{g_t^2}{2}\Delta q_t(x), \quad (19)$$

then for the inverse time $\tau = 1 - t$, we have

$$\frac{\partial}{\partial \tau}q_{1-\tau}(x) = \langle\nabla, q_{1-\tau}(x)f_{1-\tau}(x)\rangle - \frac{g_{1-\tau}^2}{2}\Delta q_{1-\tau}(x) \quad (20)$$

$$= \left\langle\nabla, q_{1-\tau}(x)\left(f_{1-\tau}(x) - \frac{g_{1-\tau}^2}{2}\nabla \log q_{1-\tau}(x)\right)\right\rangle \quad (21)$$

$$= -\left\langle\nabla, q_{1-\tau}(x)\left(-f_{1-\tau}(x) + \left(\frac{g_{1-\tau}^2}{2} + \xi_\tau\right)\nabla \log q_{1-\tau}(x)\right)\right\rangle + \xi_\tau\Delta q_{1-\tau}(x), \quad (22)$$

which corresponds to the SDE

$$dx_\tau = d\tau\left(-f_{1-\tau}(x_\tau) + \left(\frac{g_{1-\tau}^2}{2} + \xi_\tau\right)\nabla \log q_{1-\tau}(x_\tau)\right) + \sqrt{2\xi_\tau}d\overline{W}_\tau. \quad (23)$$

### A.2 PROOF OF PROP. 2

**Proposition 2.** [Ornstein–Uhlenbeck SDE] *The time-dependent densities in Eq. (3) correspond to the marginal densities of the following SDE, with the corresponding initial condition*

$$dx_t = \underbrace{\frac{\partial \log \alpha_t}{\partial t}x_t}_{f_t(x_t)}dt + \underbrace{\sqrt{2\sigma_t^2\frac{\partial}{\partial t}\log\frac{\sigma_t}{\alpha_t}}}_{g_t}dW_t, \quad x_0 \sim q_0(x_0). \quad (4)$$

*Proof.* For individual components $q_t^i$ from Eq. (3), we have to find the vector field satisfying the continuity equation, which we can rewrite as

$$\frac{\partial}{\partial t}\log q_t^i(x) = -\langle\nabla_x, v_t^i(x)\rangle - \langle\nabla_x \log q_t^i(x), v_t^i(x)\rangle. \quad (24)$$

Using the formula for the density of the normal distribution $q_t^i(x) = \mathcal{N}(x \mid \alpha_t\mu^i, \sigma_t^2)$, we have

$$\nabla_x \log q_t^i(x) = -\frac{1}{\sigma_t^2}(x - \alpha_t\mu^i), \quad (25)$$

$$\frac{\partial}{\partial t}\log q_t^i(x) = d\frac{\partial}{\partial t}\log \sigma_t + \frac{1}{\sigma_t^3}\frac{\partial \sigma_t}{\partial t}\|x - \alpha_t\mu^i\|^2 + \frac{1}{\sigma_t^2}\langle x - \alpha_t\mu^i, \mu^i\rangle\frac{\partial \alpha_t}{\partial t} \quad (26)$$

$$= \left\langle\nabla_x, \frac{\partial \log \sigma_t}{\partial t}x\right\rangle - \left\langle\nabla_x \log q_t^i(x), \underbrace{\frac{\partial \log \sigma_t}{\partial t}(x - \alpha_t\mu^i) + \frac{\partial \alpha_t}{\partial t}\mu^i}_{v_t^i(x)}\right\rangle \quad (27)$$

$$= \langle\nabla_x, v_t^i(x)\rangle - \langle\nabla_x \log q_t^i(x), v_t^i(x)\rangle. \quad (28)$$

For the mixture of densities $q_t$ from Eq. (3), we can use Prop. 3 and write

$$v_t(x) = \frac{1}{q_t(x)N} \sum_{i=1}^{N} q_t^i(x) \left[ \frac{\partial \log \sigma_t}{\partial t}(x - \alpha_t \mu^i) + \frac{\partial \alpha_t}{\partial t} \mu^i \right] \tag{29}$$

$$= \frac{\partial \log \sigma_t}{\partial t} x + \left[ \frac{\partial \alpha_t}{\partial t} - \frac{\partial \log \sigma_t}{\partial t} \alpha_t \right] \frac{1}{q_t(x)N} \sum_{i=1}^{N} q_t^i(x) \mu^i . \tag{30}$$

At the same time

$$\nabla_x \log q_t(x) = \frac{1}{q_t(x)N} \sum_{i=1}^{N} q_t^i(x) \left[ -\frac{1}{\sigma_t^2}(x - \alpha_t \mu^i) \right] , \tag{31}$$

$$\frac{1}{q_t(x)N} \sum_{i=1}^{N} q_t^i(x) \mu^i = \frac{1}{\alpha_t} \left[ \sigma_t^2 \nabla_x \log q_t(x) + x \right] . \tag{32}$$

Using this formula in Eq. (30), we have

$$v_t(x) = \left[ \frac{\partial \log \sigma_t}{\partial t} + \frac{1}{\alpha_t} \frac{\partial \alpha_t}{\partial t} - \frac{\partial \log \sigma_t}{\partial t} \right] x + \left[ \frac{\partial \alpha_t}{\partial t} - \frac{\partial \log \sigma_t}{\partial t} \alpha_t \right] \frac{\sigma_t^2}{\alpha_t} \nabla_x \log q_t(x) \tag{33}$$

$$= \frac{\partial \log \alpha_t}{\partial t} x - \sigma_t^2 \frac{\partial}{\partial t} \log \frac{\sigma_t}{\alpha_t} \nabla_x \log q_t(x) . \tag{34}$$

Hence, we have

$$\frac{\partial}{\partial t} q_t(x) = -\left\langle \nabla, q_t(x) v_t(x) \right\rangle = -\left\langle \nabla, q_t(x) \left( \frac{\partial \log \alpha_t}{\partial t} x \right) \right\rangle + \sigma_t^2 \frac{\partial}{\partial t} \log \frac{\sigma_t}{\alpha_t} \Delta q_t(x) , \tag{35}$$

which corresponds to the SDE in the statement. $\square$

## A.3 PROOF OF PROP. 3

**Proposition 3.** [Superposition of ODEs (Liu, 2022)] *The mixture density in Eq. (6) follows the continuity equation with the superposed vector fields from Eq. (5), i.e.*

$$\frac{\partial}{\partial t} q_t^{\mathrm{mix}}(x) = -\left\langle \nabla_x, q_t^{\mathrm{mix}}(x) v_t(x) \right\rangle , \quad v_t(x) = \frac{1}{\sum_{j=1}^{N} \omega^j q_t^j(x)} \sum_{i=1}^{N} \omega^i q_t^i(x) v_t^i(x) . \tag{7}$$

*Proof.* By the straightforward substitution, we have

$$\frac{\partial}{\partial t} q_t^{\mathrm{mix}}(x) = \sum_{i=1}^{m} \omega^i \frac{\partial}{\partial t} q_t^i(x) = -\sum_{i=1}^{m} \omega^i \left\langle \nabla_x, q_t^i(x) v_t^i(x) \right\rangle \tag{36}$$

$$= -\left\langle \nabla_x, \sum_{i=1}^{m} \omega^i q_t^i(x) v_t^i(x) \right\rangle = -\left\langle \nabla_x, \frac{q_t^{\mathrm{mix}}(x)}{q_t^{\mathrm{mix}}(x)} \sum_{i=1}^{m} \omega^i q_t^i(x) v_t^i(x) \right\rangle \tag{37}$$

$$= -\left\langle \nabla_x, q_t^{\mathrm{mix}}(x) \underbrace{\frac{1}{\sum_{j=1}^{m} \omega^j q_t^j(x)} \sum_{i=1}^{m} \omega^i q_t^i(x) v_t^i(x)}_{v_t(x)} \right\rangle . \tag{38}$$

$\square$

## A.4 PROOF OF PROP. 4

**Proposition 4.** [Superposition of SDEs] *The mixture $q_t^{\mathrm{mix}}(x) := \sum_{i=1}^{N} \omega^i q_t^i(x)$ of density marginals $\{q_t^i(x)\}_{i=1}^{N}$ induced by SDEs from Eq. (8) corresponds to the following SDE*

$$dx_\tau = u_\tau(x_\tau) d\tau + g_\tau d\overline{W}_\tau , \quad u_t(x) = \frac{1}{\sum_{j=1}^{N} \omega^j q_t^j(x)} \sum_{i=1}^{N} \omega^i q_t^i(x) u_t^i(x) . \tag{9}$$

*Proof.* By the straightforward substitution, we have

$$\frac{\partial}{\partial t} q_t^{\mathrm{mix}}(x) = \sum_{i=1}^{m} \omega^i \frac{\partial}{\partial t} q_t^i(x) = \sum_{i=1}^{m} \omega^i \left( -\left\langle \nabla_x, q_t^i(x) u_t^i(x) \right\rangle + \frac{g_t^2}{2} \Delta q_t^i(x) \right). \tag{39}$$

The first term is analogous to Prop. 3, i.e.

$$\sum_{i=1}^{m} \omega^i \left( -\left\langle \nabla_x, q_t^i(x) u_t^i(x) \right\rangle \right) = -\left\langle \nabla_x, q_t^{\mathrm{mix}}(x) \underbrace{\frac{1}{\sum_{j=1}^{m} \omega^j q_t^j(x)} \sum_{i=1}^{m} \omega^i q_t^i(x) u_t^i(x)}_{u_t(x)} \right\rangle. \tag{40}$$

The second term is

$$\sum_{i=1}^{m} \omega^i \frac{g_t^2}{2} \Delta q_t^i(x) = \frac{g_t^2}{2} \Delta \sum_{i=1}^{m} \omega^i \Delta q_t^i(x) = \frac{g_t^2}{2} \Delta q_t^{\mathrm{mix}}(x). \tag{41}$$

This results in the following PDE

$$\frac{\partial}{\partial t} q_t^{\mathrm{mix}}(x) = -\left\langle \nabla_x, q_t^{\mathrm{mix}}(x) \underbrace{\frac{1}{\sum_{j=1}^{m} \omega^j q_t^j(x)} \sum_{i=1}^{m} \omega^i q_t^i(x) u_t^i(x)}_{u_t(x)} \right\rangle + \frac{g_t^2}{2} \Delta q_t^{\mathrm{mix}}(x). \tag{42}$$

$\square$

# B  FLOWS AND DIFFUSION AS A SUPERPOSITION OF ELEMENTARY VECTOR FIELDS

From Prop. 3, one can immediately get the main principle for the simulation-free training of generative flow models, as demonstrated in the following proposition.

**Proposition 7.** [Superposition of $L_2$-losses] *For the parametric vector field model $v_t(x; \theta)$ with parameters $\theta$, the $L_2$-loss for the vector field from Prop. 3 can be decomposed into the losses for the vector fields from Eq. (5), i.e.*

$$\int_0^1 dt \, \mathbb{E}_{q_t^{\mathrm{mix}}(x)} \| v_t(x) - v_t(x; \theta) \|^2 = \sum_{i=1}^{N} \omega_i \int_0^1 dt \, \mathbb{E}_{q_t^i(x)} \| v_t^i(x) - v_t(x; \theta) \|^2 + \text{constant},$$

*where the constant does not depend on the parameters $\theta$.*

*Proof.* By the straightforward calculation, we have

$$\int_0^1 dt \, \mathbb{E}_{q_t^{\mathrm{mix}}(x)} \| v_t(x) - v_t(x; \theta) \|^2 = \int_0^1 dt \, \mathbb{E}_{q_t^{\mathrm{mix}}(x)} \left[ \| v_t(x) \|^2 - 2\langle v_t(x), v_t(x; \theta) \rangle + \| v_t(x; \theta) \|^2 \right],$$

where the first term is constant w.r.t. $\theta$ and the last term is amenable for a straightforward estimation. The middle term, according to Prop. 3, is

$$\int dx \, q_t^{\mathrm{mix}}(x) \langle v_t(x), v_t(x; \theta) \rangle = \int dx \, \sum_{i=1}^{m} \omega^i q_t^i(x) \langle v_t^i(x), v_t(x; \theta) \rangle. \tag{43}$$

Thus, we have

$$\int_0^1 dt \, \mathbb{E}_{q_t^{\mathrm{mix}}(x)} \left[ \| v_t(x) \|^2 - 2\langle v_t(x), v_t(x; \theta) \rangle + \| v_t(x; \theta) \|^2 \right] \tag{44}$$

$$= \int_0^1 dt \, \left[ \underbrace{\mathbb{E}_{q_t^{\mathrm{mix}}(x)} \| v_t(x) \|^2 - \sum_{i=1}^{m} \omega^i \mathbb{E}_{q_t^i(x)} \| v_t^i(x) \|^2 + \sum_{i=1}^{m} \omega^i \mathbb{E}_{q_t^i(x)} \| v_t^i(x) \|^2}_{\text{constant}} - \right. \tag{45}$$

$$\left. - 2 \sum_{i=1}^{m} \omega^i \mathbb{E}_{q_t^i(x)} \langle v_t^i(x), v_t(x; \theta) \rangle + \sum_{i=1}^{m} \omega^i \mathbb{E}_{q_t^i(x)} \| v_t(x; \theta) \|^2 \right] \tag{46}$$

$$= \int_0^1 dt \sum_{i=1}^m \omega^i \mathbb{E}_{q_t^i(x)} \big\| v_t^i(x) - v_t(x;\theta) \big\|^2 + \text{constant} . \tag{47}$$

$\square$

The proof for proposition follows a simple extension of Liu (2022). Consequently, the generative modeling problem explicitly boils down to learning a time-dependent vector field $v_t(x,\theta)$. Moreover, for the generative modeling task, the noising process is required to satisfy the following boundary conditions $q_0^i(x) = \delta(x - \mu^i)$ and $q_1^i(x) = p_{\text{noise}}(x)$, $\forall i \in \mathcal{D}$. Generating samples that resemble $p_{\text{data}}(\mu)$ during inference then simply amounts to solving the reverse-time ODE with the learned model $v_t(x;\theta)$ starting from a noisy sample $x_1 \sim p_{\text{noise}}(x)$.

Diffusion models can be incorporated into the flow-based framework by selecting a Gaussian forward process which gives the marginal density at a timestep $t$ as the mixture of corresponding Gaussians,

$$q_t^i(x) = \mathcal{N}(x \,|\, \alpha_t \mu^i, \sigma_t^2), \quad q_t(x) = \frac{1}{N} \sum_{i=1}^N q_t^i(x). \tag{48}$$

Under this Gaussian diffusion framework, we can express the vector fields analytically using the score, which is formalized by the following proposition.

**Proposition 8.** [Diffusion processes] *The time-dependent densities in Eq. (48) satisfy the following continuity equations*

$$\frac{\partial}{\partial t} q_t^i(x) = -\big\langle \nabla_x, q_t^i(x) v_t^i(x) \big\rangle, \quad v_t^i(x) = \frac{\partial \log \sigma_t}{\partial t}(x - \alpha_t \mu^i) + \frac{\partial \alpha_t}{\partial t} \mu^i, \tag{49}$$

$$\frac{\partial}{\partial t} q_t(x) = -\big\langle \nabla_x, q_t(x) v_t(x) \big\rangle, \quad v_t(x) = \frac{\partial \log \alpha_t}{\partial t} x - \left( \sigma_t^2 \frac{\partial}{\partial t} \log \frac{\sigma_t}{\alpha_t} \right) \nabla_x \log q_t(x). \tag{50}$$

The proof repeats the proof for Prop. 2 in App. A.2 since Eq. (50) corresponds to the Ornstein-Uhlenbeck process:

$$dx_t = \frac{\partial \log \alpha_t}{\partial t} x_t dt + \sqrt{2 \left( \sigma_t^2 \frac{\partial}{\partial t} \log \frac{\sigma_t}{\alpha_t} \right)} dW_t. \tag{51}$$

Diffusion models using a Gaussian perturbation kernel enjoy the benefit of giving an exact expression for the Stein score of the perturbed data which leads to the celebrated denoising score matching objective (Vincent, 2011; Ho et al., 2020). Analogously to Prop. 7 we can write the denoising score matching objective as the superposition of the scores as demonstrated in the following proposition.

**Proposition 9.** [Denoising Score Matching (Vincent, 2011)] *For the parametric score model* $\nabla \log q_t(x;\theta)$ *with parameters* $\theta$, *the score matching objective can be decomposed into the corresponding objectives for the individual scores, i.e.*

$$\int_0^1 dt \ \mathbb{E}_{q_t(x)} \| \nabla \log q_t(x) - \nabla \log q_t(x;\theta) \|^2 =$$

$$= \frac{1}{N} \sum_{i=1}^N \int_0^1 dt \ \mathbb{E}_{q_t^i(x)} \big\| \nabla \log q_t^i(x) - \nabla \log q_t(x;\theta) \big\|^2 + \text{constant},$$

*where the constant does not depend on the parameters* $\theta$ *and* $\nabla \log q_t^i(x) = -\frac{1}{\sigma_t^2}(x - \alpha_t \mu^i)$.

*Proof.* From Prop. 8, we have

$$v_t^i(x) = \frac{\partial \log \sigma_t}{\partial t}(x - \alpha_t \mu^i) + \frac{\partial \alpha_t}{\partial t} \mu^i \tag{52}$$

$$= \frac{\partial \log \sigma_t}{\partial t}(x - (x + \sigma_t^2 \nabla \log q_t^i(x))) + \frac{\partial \alpha_t}{\partial t} \frac{1}{\alpha_t}(x + \sigma_t^2 \nabla \log q_t^i(x)) \tag{53}$$

$$= \frac{\partial \log \alpha_t}{\partial t} x + \sigma_t^2 \frac{\partial}{\partial t} \log \frac{\alpha_t}{\sigma_t} \nabla_x \log q_t^i(x) \tag{54}$$

$$v_t(x) = \frac{\partial \log \alpha_t}{\partial t} x + \sigma_t^2 \frac{\partial}{\partial t} \log \frac{\alpha_t}{\sigma_t} \nabla_x \log q_t(x) \,. \tag{55}$$

Applying the same change of variables to the parametric model

$$v_t(x; \theta) = \frac{\partial \log \alpha_t}{\partial t} x + \sigma_t^2 \frac{\partial}{\partial t} \log \frac{\alpha_t}{\sigma_t} \nabla_x \log q_t(x; \theta) \,, \tag{56}$$

and using Prop. 7, we have

$$\int_0^1 dt \ \mathbb{E}_{q_t(x)} \|v_t(x) - v_t(x; \theta)\|^2 = \sum_{i=1}^m \frac{1}{N} \int_0^1 dt \ \mathbb{E}_{q_t^i(x)} \|v_t^i(x) - v_t(x; \theta)\|^2 + \text{constant} \,, \tag{57}$$

$$\int_0^1 dt \ \mathbb{E}_{q_t(x)} \|\nabla_x \log q_t(x) - \nabla_x \log q_t(x; \theta)\|^2 \tag{58}$$

$$= \sum_{i=1}^m \frac{1}{N} \int_0^1 dt \ \mathbb{E}_{q_t^i(x)} \|\nabla_x \log q_t^i(x) - \nabla_x \log q_t(x; \theta)\|^2 + \frac{\text{constant}}{\left( \sigma_t^2 \frac{\partial}{\partial t} \log \frac{\alpha_t}{\sigma_t} \right)} \,, \tag{59}$$

which concludes the proof. $\square$

## C   DENSITY ESTIMATORS

**Proposition 5.** [Smooth density estimator] *For the integral curve $x(t)$ solving $dx/dt = u_t(x_t)$, and the density $q_t^i(x(t))$ satisfying the continuity equation $\frac{\partial}{\partial t} q_t^i(x) = -\langle \nabla_x, q_t^i(x) v_t^i(x) \rangle$, the log-density along the curve changes according to the following ODE*

$$\frac{d}{dt} \log q_t^i(x(t)) = -\langle \nabla_x, v_t^i(x) \rangle - \langle \nabla_x \log q_t^i(x), v_t^i(x) - u_t(x) \rangle \,. \tag{10}$$

*Proof.* By straightforward computation, we have

$$\frac{d}{dt} \log q_t^i(x(t)) = \frac{\partial}{\partial t} \log q_t^i(x) + \left\langle \nabla_x \log q_t^i(x), \frac{dx}{dt} \right\rangle \,, \tag{60}$$

and using the continuity equation $\frac{\partial}{\partial t} q_t^i(x) = -\langle \nabla_x, q_t^i(x) v_t^i(x) \rangle$, we have

$$\frac{\partial}{\partial t} \log q_t^i(x) = -\langle \nabla_x, v_t^i(x) \rangle - \langle \nabla_x \log q_t^i(x), v_t^i(x) \rangle \,, \tag{61}$$

$$\frac{d}{dt} \log q_t^i(x(t)) = -\langle \nabla_x, v_t^i(x) \rangle - \langle \nabla_x \log q_t^i(x), v_t^i(x) - u_t(x) \rangle \,. \tag{62}$$

$\square$

**Theorem 1.** [Itô density estimator] *Consider time-dependent density $q_t(x)$ induced by the marginals of the following SDE*

$$dx_t = f_t(x_t) dt + g_t dW_t \,, \quad x_{t=0} \sim q_0(x) \,, \quad t \in [0, 1] \,, \tag{11}$$

*where $dW_t$ is the Wiener process. For the reverse-time ($\tau = 1 - t$) SDEs with **any** vector field $u_\tau$ and the same diffusion coefficient $g_t$, i.e.*

$$dx_\tau = u_\tau(x_\tau) d\tau + g_{1-\tau} d\overline{W}_\tau \,, \quad \tau \in [0, 1] \,, \tag{12}$$

*the change of the log-density $\log q_\tau(x_\tau)$ follows the following SDE*

$$d \log q_{1-\tau}(x_\tau) = \langle dx_\tau, \nabla \log q_{1-\tau}(x_\tau) \rangle + \Big( \langle \nabla, f_{1-\tau}(x_\tau) \rangle +$$

$$+ \left\langle f_{1-\tau}(x_\tau) - \frac{g_{1-\tau}^2}{2} \nabla \log q_{1-\tau}(x_\tau), \nabla \log q_{1-\tau}(x_\tau) \right\rangle \Big) d\tau \,. \tag{13}$$

*Proof.* The Fokker-Planck equation describing the evolution of the marginals for the forward process is

$$\frac{\partial q_t(x)}{\partial t} = -\langle \nabla, f_t(x)q_t(x) \rangle + \frac{g_t^2}{2}\Delta q_t(x), \tag{63}$$

hence, for the inverse time $\tau = 1 - t$, we have

$$\frac{\partial}{\partial \tau}q_{1-\tau}(x) = \langle \nabla, f_{1-\tau}(x)q_{1-\tau}(x) \rangle - \frac{g_{1-\tau}^2}{2}\Delta q_{1-\tau}(x) \tag{64}$$

$$\frac{\partial}{\partial \tau}\log q_{1-\tau}(x) = \langle \nabla, f_{1-\tau}(x) \rangle + \langle \nabla \log q_{1-\tau}(x), f_{1-\tau}(x) \rangle - $$
$$- \frac{g_{1-\tau}^2}{2}\Delta \log q_{1-\tau}(x) - \frac{g_{1-\tau}^2}{2}\|\nabla \log q_{1-\tau}(x)\|^2. \tag{65}$$

For the following reverse-time SDE

$$dx_\tau = u_\tau(x_\tau)d\tau + g_{1-\tau}d\overline{W}_\tau, \tag{66}$$

using Itô's lemma for $\log q_{1-\tau}(x_\tau)$, we have

$$d\log q_{1-\tau}(x_\tau) = \left(\frac{\partial}{\partial \tau}\log q_{1-\tau}(x_\tau) + \langle \nabla \log q_{1-\tau}(x_\tau), u_\tau(x_\tau) \rangle + \right. \tag{67}$$

$$\left. + \frac{g_{1-\tau}^2}{2}\Delta \log q_{1-\tau}(x_\tau)\right)d\tau + g_{1-\tau}\langle \nabla \log q_{1-\tau}(x_\tau), d\overline{W}_\tau \rangle. \tag{68}$$

Using Eq. (65), we have

$$d\log q_{1-\tau}(x_\tau) = \left(\left\langle \nabla \log q_{1-\tau}(x_\tau), f_{1-\tau}(x_\tau) - \frac{g_{1-\tau}^2}{2}\nabla \log q_{1-\tau}(x_\tau) \right\rangle + \right. \tag{69}$$

$$\left. + \langle \nabla, f_{1-\tau}(x_\tau) \rangle\right)d\tau + \left\langle \nabla \log q_{1-\tau}(x_\tau), \underbrace{u_\tau(x_\tau)d\tau + g_{1-\tau}d\overline{W}_\tau}_{dx_\tau} \right\rangle. \tag{70}$$

$\square$

### C.1 PROOF OF PROP. 6

**Proposition 6.** [Density control] *For the SDE*

$$dx_\tau = \sum_{j=1}^{M}\kappa_j u_\tau^j(x_\tau)d\tau + g_{1-\tau}d\overline{W}_\tau, \tag{16}$$

*where $\kappa$ are the weights of different models and $\sum_j \kappa_j = 1$, one can find $\kappa$ that satisfies*

$$d\log q_{1-\tau}^i(x_\tau) = d\log q_{1-\tau}^j(x_\tau), \quad \forall \ i,j \in [M], \tag{17}$$

*by solving a system of $M + 1$ linear equations w.r.t. $\kappa$.*

*Proof.* Using the result of Thm. 1, for the density change of every model, we have

$$d\log q_{1-\tau}^i(x_\tau) = \langle dx_\tau, \nabla \log q_{1-\tau}^i(x_\tau) \rangle + \left(\langle \nabla, f_{1-\tau}(x_\tau) \rangle + \right. \tag{71}$$

$$\left. + \left\langle f_{1-\tau}(x_\tau) - \frac{g_{1-\tau}^2}{2}\nabla \log q_{1-\tau}^i(x_\tau), \nabla \log q_{1-\tau}^i(x_\tau) \right\rangle\right)d\tau \tag{72}$$

$$= \sum_{j=1}^{M}\kappa_j d\tau \langle u_\tau^j(x_\tau), \nabla \log q_{1-\tau}^i(x_\tau) \rangle + \left(\langle \nabla, f_{1-\tau}(x_\tau) \rangle + \right. \tag{73}$$

$$+ \left\langle g_{1-\tau}d\overline{W}_\tau + \left(f_{1-\tau}(x_\tau) - \frac{g_{1-\tau}^2}{2}\nabla \log q_{1-\tau}^i(x_\tau)\right)d\tau, \nabla \log q_{1-\tau}^i(x_\tau) \right\rangle. \tag{74}$$

Let us denote

$$a_{ij} = d\tau \langle u_\tau^j(x_\tau), \nabla \log q_{1-\tau}^i(x_\tau) \rangle, \tag{75}$$

$$b_i = \langle \nabla, f_{1-\tau}(x_\tau) \rangle + \left\langle g_{1-\tau} d\overline{W}_\tau + \left( f_{1-\tau}(x_\tau) - \frac{g_{1-\tau}^2}{2} \nabla \log q_{1-\tau}^i(x_\tau) \right) d\tau, \nabla \log q_{1-\tau}^i(x_\tau) \right\rangle. \tag{76}$$

Then the change of densities can be written as the following linear transformation

$$\begin{bmatrix} d \log q_{1-\tau}^1(x_\tau) \\ \dots \\ d \log q_{1-\tau}^i(x_\tau) \\ \dots \\ d \log q_{1-\tau}^M(x_\tau) \end{bmatrix} = A\kappa + b. \tag{77}$$

Note that we want to find $\kappa$ such that $d \log q_{1-\tau}^i = d \log q_{1-\tau}^j$, $\forall\, i, j \in [M]$ and $\sum_j \kappa_j = 1$. This is equivalent to the following system of $M+1$ linear equations

$$\begin{pmatrix} a_{11} & \dots & a_{1M} & -1 \\ \vdots & \ddots & \vdots & -1 \\ a_{M1} & \dots & a_{MM} & -1 \\ 1 & \dots & 1 & 0 \end{pmatrix} \begin{pmatrix} \kappa_1 \\ \vdots \\ \kappa_M \\ d \log \end{pmatrix} = \begin{pmatrix} -b_1 \\ \vdots \\ -b_M \\ 1 \end{pmatrix}, \tag{78}$$

where we have introduced new variable $d \log = d \log q_{1-\tau}^i$, $\forall\, i \in [M]$. $\qquad\square$

## D   DISCRETE-TIME PERSPECTIVE ON THM. 1

In this section we derive Thm. 1 from another perspective by operating with discrete time transition kernels and the detailed balance. Namely, we aim to compute the marginal density by starting from the detailed balance equation, which states the equivalence of the following joint densities:

$$q_t(y)k_{\Delta t}(x \mid y) = q_{t+\Delta t}(x)r_{\Delta t}(y \mid x). \tag{79}$$

The detailed balance condition simply states the pair $(x, y)$ can be sampled in two equivalent ways: 1.) sampling $y$ from the marginal $q_t(y)$ and then sampling $x$ via the noising kernel $k_{\Delta t}(x \mid y)$, or 2.) sampling $x$ from $q_{t+\Delta t}(y)$ and then denoising it via $r_{\Delta t}(y \mid x)$. Indeed, there are infinitely many valid kernels that may satisfy the detailed balance; we make a specific choice informed by the following principle. We aim to construct a universal noising kernel that is independent of the data distribution or other densities, i.e. $p_{\text{data}}(\cdot), q_t(\cdot), q_{t+\Delta t}(\cdot)$. Remarkably, this principle results in a unique kernel choice, which we formalize in the following proposition.

**Proposition 10.** [Universal noising kernel] *For any data density $p_{\text{data}}(\mu)$, for the continuous noising process $q_t(x) = \int d\mu\, \mathcal{N}(x \mid \alpha_t \mu, \sigma_t^2)p_{\text{data}}(\mu)$, there exists unique noising kernel*

$$k_{\Delta t}(x \mid y) = \mathcal{N}\left( x \,\Big|\, \frac{\alpha_{t+\Delta t}}{\alpha_t}y, S_{t+\Delta t}^2 \right), \quad q_{t+\Delta t}(x) = \int dy\, k_{\Delta t}(y \mid x)q_t(y), \tag{80}$$

*where $S_{t+\Delta t}^2 = \sigma_{t+\Delta t}^2 - \sigma_t^2 \frac{\alpha_{t+\Delta t}^2}{\alpha_t^2}$ and it is independent of the densities $p_{\text{data}}(\cdot), q_t(\cdot), q_{t+\Delta t}(\cdot)$.*

The proof for this proposition is presented in App. D.1. Note that the proposition holds exactly for any $\Delta t$—i.e. there are no any assumptions on the scale of $\Delta t$ or any approximations.

Once the noising kernel is fixed, the detailed balance Eq. (79) uniquely defines the denoising kernel $r_{\Delta t}(y \mid x)$ that propagates samples back in time. However, the analytic form of this kernel depends on the densities $q_{t+\Delta t}(x)$ and $q_t(y)$, which are unavailable for the modern large-scale diffusion models that are served using limited API endpoints. Therefore, we consider the Gaussian approximation of the reverse kernel and justify its applicability in the following theorem.

**Theorem 2.** [Denoising kernel] *For the universal noising kernel $k_{\Delta t}(x \mid y)$, the Gaussian approximation of the corresponding reverse kernel $r_{\Delta t}(y \mid x) = k_{\Delta t}(x \mid y)q_t(y)/q_{t+\Delta t}(x)$ is*

$$\tilde{r}_{\Delta t}(y \mid x) = \mathcal{N}\left( y \,\Big|\, \frac{\alpha_t}{\alpha_{t+\Delta t}}x + \frac{\alpha_t}{\alpha_{t+\Delta t}}S_{t+\Delta t}^2 \nabla \log q_{t+\Delta t}(x), \left( \frac{\alpha_t}{\alpha_{t+\Delta t}}S_{t+\Delta t} \right)^2 \right), \tag{81}$$

*which corresponds to a single step of the Euler discretization of the following SDE*

$$dx_\tau = -\left(\frac{\partial}{\partial t}\log\alpha_t x - 2\sigma_t^2\frac{\partial}{\partial t}\log\frac{\sigma_t}{\alpha_t}\nabla_x\log q_t(x)\right)\cdot d\tau + \sqrt{2\sigma_t^2\frac{\partial}{\partial t}\log\frac{\sigma_t}{\alpha_t}}dW_\tau\,, \quad (82)$$

*and*

$$D_{\mathrm{KL}}(\tilde r_{\Delta t}(y\,|\,x)\|r_{\Delta t}(y\,|\,x)) = o(\Delta t)\,. \tag{83}$$

See proof in App. D.3. Given the noising kernel from Prop. 10 and the approximation of the reverse kernel from Thm. 2, we propose to estimate the log marginal density of the current sample using the detailed balance as follows,

$$\log q_t(y) - \log q_{t+\Delta t}(x) = -\log k_{\Delta t}(x\,|\,y) + \log\tilde r_{\Delta t}(y\,|\,x) + \log\frac{r_{\Delta t}(y\,|\,x)}{\tilde r_{\Delta t}(y\,|\,x)}\,, \tag{84}$$

where the last term involves the unknown reversed kernel. We argue that this term can be disregarded due to Lemma 1. Namely, in the following theorem, we study the distribution of the last term for different samples $y$ generated from $x$ and argue that this term can be ignored.

**Theorem 3.** [Detailed balance density estimator error] *For $y = x + \Delta t\cdot v + g_{t+\Delta t}\sqrt{\Delta t}\varepsilon$, where $g_{t+\Delta t} = \sqrt{2\sigma_t^2\frac{\partial}{\partial t}\log\frac{\sigma_t}{\alpha_t}}$ and $\varepsilon$ is a standard normal random variable, the following estimator is unbiased w.r.t. the generated samples $y$, i.e.*

$$\mathbb{E}_\varepsilon\left[\log\frac{r_{\Delta t}(y\,|\,x)}{\tilde r_{\Delta t}(y\,|\,x)}\right] = o(\Delta t)\,, \tag{85}$$

*and the variance is*

$$\mathbf{D}_\varepsilon\left[\log\frac{r_{\Delta t}(y\,|\,x)}{\tilde r_{\Delta t}(y\,|\,x)}\right] = \Delta t^2\frac{g_{t+\Delta t}^4}{4}\mathbf{D}_\varepsilon\left(\varepsilon^T\frac{\partial^2\log q_t(x)}{\partial x^2}\varepsilon\right) + o(\Delta t^2)\,, \tag{86}$$

*where $\frac{\partial^2\log q_{t+\Delta t}(x)}{\partial x^2}$ is the Hessian matrix of the log-density.*

We postpone the proof of this theorem until App. D.4. Note that the next variable $y$ is completely defined by the random variable $\varepsilon$. Thus, the mean and the variance are evaluated taking into account different outcomes of $y$. However, the variance of this distribution is related to the variance of Hutchinson's estimator. Indeed, for the Euler integration scheme of the log-density $\log q_{t+\Delta t}(x_{t+\Delta t}) = \log q(x_t) - \Delta t\cdot\langle\nabla_x, v_t(x_t)\rangle$, Hutchinson's estimator is

$$\langle\nabla_x, v_t(x_t)\rangle = \mathbb{E}_\varepsilon\left[\varepsilon^T\frac{\partial v_t(x_t)}{\partial x_t}\varepsilon\right] \simeq \varepsilon^T\frac{\partial v_t(x_t)}{\partial x_t}\varepsilon\,, \ \varepsilon\sim\mathcal{N}(0,1)\,, \tag{87}$$

where $\frac{\partial v_t(x_t)}{\partial x_t}$ is the Jacobian matrix. The variance of this estimator is

$$\mathbf{D}_\varepsilon[\log q_{t+\Delta t}(x_{t+\Delta t})] = \Delta t^2\mathbf{D}_\varepsilon\left[\varepsilon^T\frac{\partial v_t(x_t)}{\partial x_t}\varepsilon\right]\,. \tag{88}$$

Note that here the variable $\varepsilon$ is newly introduced and is not related anyhow to $x_{t+\Delta t}$.

Thus, we arrive at the following estimator of the change of density

$$\log q_t(y) - \log q_{t+\Delta t}(x) \approx -\log k_{\Delta t}(x\,|\,y) + \log\tilde r_{\Delta t}(y\,|\,x)\,. \tag{89}$$

We further highlight that since the generation via the inference process happens in many steps with small $\Delta t$, we approximate the relation Eq. (89) up to the second order terms in $\Delta t$.

**Proposition 11.** [Recurrence relation for the log-density] *For $y = x + \Delta t\cdot v + g_{t+\Delta t}\sqrt{\Delta t}\varepsilon$, where $g_{t+\Delta t} = \sqrt{2\sigma_t^2\frac{\partial}{\partial t}\log\frac{\sigma_t}{\alpha_t}}$ and $\varepsilon$ is a standard normal random variable, the estimator from Eq. (89) can be expanded as follows*

$$-\log k_{\Delta t}(x\,|\,y) + \log\tilde r_{\Delta t}(y\,|\,x) = d\Delta t\frac{\partial}{\partial t}\log\alpha_{t+\Delta t} - \Delta t\frac{g_{t+\Delta t}^2}{2}\|\nabla\log q_{t+\Delta t}(x)\|^2 +$$

$$+\left\langle\Delta t\cdot v + g_{t+\Delta t}\sqrt{\Delta t}\varepsilon + \Delta t\frac{\partial}{\partial t}\log\alpha_{t+\Delta t}x, \nabla\log q_{t+\Delta t}(x)\right\rangle + o(\Delta t)$$

$$\tag{90}$$

The proof for this proposition is included in App. D.5. For practical use cases we note that Eq. (90) is the final form of our density estimator.

## D.1   PROOF OF PROP. 10

**Proposition 10.** [Universal noising kernel] *For any data density $p_{\mathrm{data}}(\mu)$, for the continuous noising process $q_t(x) = \int d\mu \, \mathcal{N}(x \,|\, \alpha_t \mu, \sigma_t^2) p_{\mathrm{data}}(\mu)$, there exists unique noising kernel*

$$k_{\Delta t}(x \,|\, y) = \mathcal{N}\left(x \,\middle|\, \frac{\alpha_{t+\Delta t}}{\alpha_t} y, S_{t+\Delta t}^2\right), q_{t+\Delta t}(x) = \int dy \, k_{\Delta t}(y \,|\, x) q_t(y), \qquad (80)$$

*where $S_{t+\Delta t}^2 = \sigma_{t+\Delta t}^2 - \sigma_t^2 \frac{\alpha_{t+\Delta t}^2}{\alpha_t^2}$ and it is independent of the densities $p_{\mathrm{data}}(\cdot), q_t(\cdot), q_{t+\Delta t}(\cdot)$.*

*Proof.*

$$q_t(x) = \int d\mu \, \mathcal{N}(x \,|\, \alpha_t \mu, \sigma_t^2) p_{\mathrm{data}}(\mu) \qquad (91)$$

Let's derive the incremental kernel $k_{\Delta t}(x \,|\, y)$ from this formula, i.e.

$$q_{t+\Delta t}(x) = \int dy \, k_{\Delta t}(x \,|\, y) q_t(y) \qquad (92)$$

$$\int d\mu \, \mathcal{N}(x \,|\, \alpha_{t+\Delta t}\mu, \sigma_{t+\Delta t}^2) p_{\mathrm{data}}(\mu) = \int dy \, k_{\Delta t}(x \,|\, y) \int d\mu \, \mathcal{N}(y \,|\, \alpha_t \mu, \sigma_t^2) p_{\mathrm{data}}(\mu) \qquad (93)$$

$$\int d\mu \left( \mathcal{N}(x \,|\, \alpha_{t+\Delta t}\mu, \sigma_{t+\Delta t}^2) - \int dy \, k_{\Delta t}(x \,|\, y) \mathcal{N}(y \,|\, \alpha_t \mu, \sigma_t^2) \right) p_{\mathrm{data}}(\mu) = 0 \qquad (94)$$

$$\mathcal{N}(x \,|\, \alpha_{t+\Delta t}\mu, \sigma_{t+\Delta t}^2) = \int dy \, k_{\Delta t}(x \,|\, y) \mathcal{N}(y \,|\, \alpha_t \mu, \sigma_t^2) \qquad (95)$$

Obviously, one can perform the following change of variables

$$x = \alpha_{t+\Delta t}\mu + \frac{y - \alpha_t \mu}{\sigma_t} S + R\varepsilon, \text{ where } y \sim \mathcal{N}(y \,|\, \alpha_t \mu, \sigma_t^2), \text{ and } \varepsilon \sim \mathcal{N}(\varepsilon \,|\, 0, 1), \qquad (96)$$

$$\text{where we have to take } S^2 + R^2 = \sigma_{t+\Delta t}^2. \qquad (97)$$

To make the kernel independent of $\mu$ we have to choose $R = \sigma_t \frac{\alpha_{t+\Delta t}}{\alpha_t}$, then the kernel is

$$k_{\Delta t}(x \,|\, y) = \mathcal{N}\left(x \,\middle|\, \frac{\alpha_{t+\Delta t}}{\alpha_t} y, \underbrace{\sigma_{t+\Delta t}^2 - \sigma_t^2 \frac{\alpha_{t+\Delta t}^2}{\alpha_t^2}}_{S_{t+\Delta t}^2}\right). \qquad (98)$$

$\square$

## D.2   LEMMA 1

**Lemma 1.** [Reverse kernel lemma] *For the universal noising kernel $k_{\Delta t}(x \,|\, y)$, the corresponding reverse kernel $r_{\Delta t}(y \,|\, x) = k_{\Delta t}(x \,|\, y) q_t(y) / q_{t+\Delta t}(x)$, and its Gaussian approximation*

$$\tilde{r}_{\Delta t}(y \,|\, x) = \mathcal{N}\left(y \,\middle|\, \frac{\alpha_t}{\alpha_{t+\Delta t}} x + \frac{\alpha_t}{\alpha_{t+\Delta t}} S_{t+\Delta t}^2 \nabla \log q_{t+\Delta t}(x), \left(\frac{\alpha_t}{\alpha_{t+\Delta t}} S_{t+\Delta t}\right)^2\right), \qquad (99)$$

*we have*

$$\log \frac{r_{\Delta t}(x + \Delta t \cdot v + g_{t+\Delta t}\sqrt{\Delta t}\varepsilon \,|\, x)}{\tilde{r}_{\Delta t}(x + \Delta t \cdot v + g_{t+\Delta t}\sqrt{\Delta t}\varepsilon \,|\, x)} = \qquad (100)$$

$$= \Delta t \cdot \frac{g_{t+\Delta t}^2}{2} \left( \Delta \log q_{t+\Delta t}(x) - \varepsilon^T \frac{\partial^2 \log q_t(x)}{\partial x^2} \varepsilon \right) + o(\Delta t), \qquad (101)$$

*where $g_{t+\Delta t} = \sqrt{2\sigma_{t+\Delta t}^2 \frac{\partial}{\partial t} \log \frac{\sigma_{t+\Delta t}}{\alpha_{t+\Delta t}}}$.*

*Proof.* From the detailed balance equation, we have

$$\log r_{\Delta t}(y \mid x) = \log k_{\Delta t}(x \mid y) + \log q_t(y) - \log q_{t+\Delta t}(x) \tag{102}$$

$$= \frac{-1}{2S^2_{t+\Delta t}}\left(x - \frac{\alpha_{t+\Delta t}}{\alpha_t}y\right)^2 - \frac{d}{2}\log 2\pi S^2_{t+\Delta t} + \log q_t(y) - \log q_{t+\Delta t}(x), \tag{103}$$

where we use the definition of the forward kernel. Thus, the difference between kernels is

$$\log \frac{r_{\Delta t}(y \mid x)}{\tilde{r}_{\Delta t}(y \mid x)} = \frac{-1}{2S^2_{t+\Delta t}}\left(x - \frac{\alpha_{t+\Delta t}}{\alpha_t}y\right)^2 - \frac{d}{2}\log 2\pi S^2_{t+\Delta t} + \log q_t(y) - \log q_{t+\Delta t}(x) + \tag{104}$$

$$+ \frac{1}{2(S_{t+\Delta t}\frac{\alpha_t}{\alpha_{t+\Delta t}})^2}\left(y - \frac{\alpha_t}{\alpha_{t+\Delta t}}x - \frac{\alpha_t}{\alpha_{t+\Delta t}}S^2_{t+\Delta t}\nabla \log q_{t+\Delta t}(x)\right)^2 + \tag{105}$$

$$+ \frac{d}{2}\log 2\pi S^2_{t+\Delta t} + \frac{d}{2}\log\left(\frac{\alpha_t}{\alpha_{t+\Delta t}}\right)^2. \tag{106}$$

Opening the brackets, we have

$$\log \frac{r_{\Delta t}(y \mid x)}{\tilde{r}_{\Delta t}(y \mid x)} = \frac{-1}{2S^2_{t+\Delta t}}\left(x - \frac{\alpha_{t+\Delta t}}{\alpha_t}y\right)^2 + \log q_t(y) - \log q_{t+\Delta t}(x) + \tag{107}$$

$$+ \frac{1}{2S^2_{t+\Delta t}}\left(\frac{\alpha_{t+\Delta t}}{\alpha_t}y - x\right)^2 - \left\langle \frac{\alpha_{t+\Delta t}}{\alpha_t}y - x, \nabla \log q_{t+\Delta t}(x)\right\rangle + \tag{108}$$

$$+ \frac{S^2_{t+\Delta t}}{2}\|\nabla \log q_{t+\Delta t}(x)\|^2 + \frac{d}{2}\log\left(\frac{\alpha_t}{\alpha_{t+\Delta t}}\right)^2. \tag{109}$$

The constants can be estimated as follows

$$S^2_{t+\Delta t} = \sigma^2_{t+\Delta t} - \sigma^2_t\frac{\alpha^2_{t+\Delta t}}{\alpha^2_t} = \sigma^2_{t+\Delta t} - (\sigma^2_{t+\Delta t} - 2dt\sigma_{t+\Delta t}\frac{\partial \sigma_{t+\Delta t}}{\partial t} + o(\Delta t))\frac{\alpha^2_{t+\Delta t}}{\alpha^2_t} \tag{110}$$

$$= \sigma^2_{t+\Delta t} - (\sigma^2_{t+\Delta t} - 2dt\sigma_{t+\Delta t}\frac{\partial \sigma_{t+\Delta t}}{\partial t} + o(\Delta t))\left(1 + 2dt\frac{\alpha^2_{t+\Delta t}}{\alpha^3_{t+\Delta t}}\frac{\partial \alpha_{t+\Delta t}}{\partial t} + o(\Delta t)\right) \tag{111}$$

$$= 2dt\sigma^2_{t+\Delta t}\frac{\partial}{\partial t}\log\frac{\sigma_{t+\Delta t}}{\alpha_{t+\Delta t}} + o(\Delta t), \tag{112}$$

and

$$\frac{\alpha_{t+\Delta t}}{\alpha_t} = 1 - \frac{\alpha_{t+\Delta t}}{\alpha^2_{t+\Delta t}}\frac{\partial \alpha_{t+\Delta t}}{\partial t}(-\Delta t) + o(\Delta t) = 1 + \Delta t\frac{\partial \log \alpha_{t+\Delta t}}{\partial t} + o(\Delta t) \tag{113}$$

$$\frac{d}{2}\log\left(\frac{\alpha_t}{\alpha_{t+\Delta t}}\right)^2 = -dtd\frac{\partial}{\partial t}\log \alpha_{t+\Delta t} + o(\Delta t). \tag{114}$$

Thus, we have

$$\log \frac{r_{\Delta t}(y \mid x)}{\tilde{r}_{\Delta t}(y \mid x)} = \log q_t(y) - \log q_{t+\Delta t}(x) + \Delta t\sigma^2_{t+\Delta t}\frac{\partial}{\partial t}\log\frac{\sigma_{t+\Delta t}}{\alpha_{t+\Delta t}}\|\nabla \log q_{t+\Delta t}(x)\|^2 - \tag{115}$$

$$- dtd\frac{\partial}{\partial t}\log \alpha_{t+\Delta t} - \left\langle (1 + \Delta t\frac{\partial \log \alpha_{t+\Delta t}}{\partial t})y - x, \nabla \log q_{t+\Delta t}(x)\right\rangle + o(\Delta t). \tag{116}$$

From [Prop. 8](#), the time-derivative of the density is

$$\frac{\partial}{\partial t}\log q_t(x) = -\langle \nabla, v_t(x)\rangle - \langle \nabla \log q_t(x), v_t(x)\rangle \tag{117}$$

$$= -d\frac{\partial \log \alpha_t}{\partial t} + \left(\sigma^2_t\frac{\partial}{\partial t}\log\frac{\sigma_t}{\alpha_t}\right)\Delta \log q_t(x) - \tag{118}$$

$$- \frac{\partial \log \alpha_t}{\partial t}\langle \nabla \log q_t(x), x\rangle + \left(\sigma^2_t\frac{\partial}{\partial t}\log\frac{\sigma_t}{\alpha_t}\right)\|\nabla \log q_t(x)\|^2. \tag{119}$$

Hence, we have

$$\log q_t(x) - \log q_{t+\Delta t}(x) = -\Delta t \frac{\partial}{\partial t} \log q_{t+\Delta t}(x) + o(\Delta t) \tag{120}$$

$$= \Delta t d \frac{\partial \log \alpha_{t+\Delta t}}{\partial t} - \Delta t \left( \sigma_{t+\Delta t}^2 \frac{\partial}{\partial t} \log \frac{\sigma_{t+\Delta t}}{\alpha_{t+\Delta t}} \right) \Delta \log q_{t+\Delta t}(x) + \tag{121}$$

$$+ \Delta t \frac{\partial \log \alpha_{t+\Delta t}}{\partial t} \left\langle \nabla \log q_{t+\Delta t}(x), x \right\rangle - \Delta t \left( \sigma_{t+\Delta t}^2 \frac{\partial}{\partial t} \log \frac{\sigma_{t+\Delta t}}{\alpha_{t+\Delta t}} \right) \| \nabla \log q_{t+\Delta t}(x) \|^2 . \tag{122}$$

Using it in Eq. (116), we have

$$\log \frac{r_{\Delta t}(y \mid x)}{\tilde{r}_{\Delta t}(y \mid x)} = \log q_t(y) - \log q_t(x) - \Delta t \left( \sigma_{t+\Delta t}^2 \frac{\partial}{\partial t} \log \frac{\sigma_{t+\Delta t}}{\alpha_{t+\Delta t}} \right) \Delta \log q_{t+\Delta t}(x) - \tag{123}$$

$$- (1 + \Delta t \frac{\partial \log \alpha_{t+\Delta t}}{\partial t}) \langle y - x, \nabla \log q_{t+\Delta t}(x) \rangle + o(\Delta t) . \tag{124}$$

Finally, we consider $y = x + \Delta t \cdot v + g_{t+\Delta t} \sqrt{\Delta t} \varepsilon$. Then we can estimate

$$\log q_t(y) = \log q_t(x) + \langle \nabla \log q_t(x), y - x \rangle + \frac{1}{2} (y - x)^T \frac{\partial^2 \log q_t(x)}{\partial x^2} (y - x) \tag{125}$$

$$+ o(\|y - x\|^2) \tag{126}$$

$$= \log q_t(x) + \left\langle \nabla \log q_t(x), \Delta t \cdot v + g_{t+\Delta t} \sqrt{\Delta t} \varepsilon \right\rangle \tag{127}$$

$$+ \Delta t \frac{g_{t+\Delta t}^2}{2} \varepsilon^T \frac{\partial^2 \log q_t(x)}{\partial x^2} \varepsilon + o(\Delta t) , \tag{128}$$

where $\frac{\partial^2 \log q_t(x)}{\partial x^2}$ denotes the Hessian of the log-density. Thus, for $y = x + \Delta t \cdot v + g_{t+\Delta t} \sqrt{\Delta t} \varepsilon$, we have

$$\log \frac{r_{\Delta t}(y \mid x)}{\tilde{r}_{\Delta t}(y \mid x)} = \Delta t \frac{g_{t+\Delta t}^2}{2} \varepsilon^T \frac{\partial^2 \log q_t(x)}{\partial x^2} \varepsilon - \Delta t \left( \sigma_{t+\Delta t}^2 \frac{\partial}{\partial t} \log \frac{\sigma_{t+\Delta t}}{\alpha_{t+\Delta t}} \right) \Delta \log q_{t+\Delta t}(x) + \tag{129}$$

$$+ \left\langle \nabla \log q_t(x) - \nabla \log q_{t+\Delta t}(x), \Delta t \cdot v + g_{t+\Delta t} \sqrt{\Delta t} \varepsilon \right\rangle + o(\Delta t) . \tag{130}$$

The last term is $o(\Delta t)$ by expanding the scores in time, hence, for $\frac{g_{t+\Delta t}^2}{2} = \sigma_{t+\Delta t}^2 \frac{\partial}{\partial t} \log \frac{\sigma_{t+\Delta t}}{\alpha_{t+\Delta t}}$, we have

$$\log \frac{r_{\Delta t}(x + \Delta t \cdot v + g_{t+\Delta t} \sqrt{\Delta t} \varepsilon \mid x)}{\tilde{r}_{\Delta t}(x + \Delta t \cdot v + g_{t+\Delta t} \sqrt{\Delta t} \varepsilon \mid x)} = \tag{131}$$

$$= \Delta t \left( \sigma_{t+\Delta t}^2 \frac{\partial}{\partial t} \log \frac{\sigma_{t+\Delta t}}{\alpha_{t+\Delta t}} \right) \left( \Delta \log q_{t+\Delta t}(x) - \varepsilon^T \frac{\partial^2 \log q_t(x)}{\partial x^2} \varepsilon \right) + o(\Delta t) . \tag{132}$$

$\square$

## D.3 PROOF OF THM. 2

**Theorem 2.** [Denoising kernel] *For the universal noising kernel $k_{\Delta t}(x \mid y)$, the Gaussian approximation of the corresponding reverse kernel $r_{\Delta t}(y \mid x) = k_{\Delta t}(x \mid y) q_t(y) / q_{t+\Delta t}(x)$ is*

$$\tilde{r}_{\Delta t}(y \mid x) = \mathcal{N} \left( y \left| \frac{\alpha_t}{\alpha_{t+\Delta t}} x + \frac{\alpha_t}{\alpha_{t+\Delta t}} S_{t+\Delta t}^2 \nabla \log q_{t+\Delta t}(x), \left( \frac{\alpha_t}{\alpha_{t+\Delta t}} S_{t+\Delta t} \right)^2 \right. \right), \tag{81}$$

*which corresponds to a single step of the Euler discretization of the following SDE*

$$dx_\tau = -\left( \frac{\partial}{\partial t} \log \alpha_t x - 2 \sigma_t^2 \frac{\partial}{\partial t} \log \frac{\sigma_t}{\alpha_t} \nabla_x \log q_t(x) \right) \cdot d\tau + \sqrt{2 \sigma_t^2 \frac{\partial}{\partial t} \log \frac{\sigma_t}{\alpha_t}} dW_\tau , \tag{82}$$

*and*

$$D_{\mathrm{KL}}(\tilde{r}_{\Delta t}(y \mid x) \| r_{\Delta t}(y \mid x)) = o(\Delta t) . \tag{83}$$

*Proof.* To find the reverse the kernel, we first use Tweedie's formula to find the expectation, i.e.

$$\nabla \log q_{t+\Delta t}(x) = -\frac{1}{q_{t+\Delta t}(x)} \int dy \, \frac{1}{S_{t+\Delta t}^2} \left( x - \frac{\alpha_{t+\Delta t}}{\alpha_t} y \right) k_{\Delta t}(x \,|\, y) q_t(y) \quad (133)$$

$$\frac{\alpha_t}{\alpha_{t+\Delta t}} S_{t+\Delta t}^2 \nabla \log q_{t+\Delta t}(x) = -\frac{1}{q_{t+\Delta t}(x)} \int dy \left( \frac{\alpha_t}{\alpha_{t+\Delta t}} x - y \right) k_{\Delta t}(x \,|\, y) q_t(y) \quad (134)$$

$$\frac{\alpha_t}{\alpha_{t+\Delta t}} x + \frac{\alpha_t}{\alpha_{t+\Delta t}} S_{t+\Delta t}^2 \nabla \log q_{t+\Delta t}(x) = \int dy \, y \frac{k_{\Delta t}(x \,|\, y) q_t(y)}{q_{t+\Delta t}(x)} = \int dy \, y r_{\Delta t}(y \,|\, x) \,. \quad (135)$$

Thus, we are going to approximate

$$\tilde{r}_{\Delta t}(y \,|\, x) = \mathcal{N}\left( y \,\middle|\, \frac{\alpha_t}{\alpha_{t+\Delta t}} x + \frac{\alpha_t}{\alpha_{t+\Delta t}} S_{t+\Delta t}^2 \nabla \log q_{t+\Delta t}(x), \left( \frac{\alpha_t}{\alpha_{t+\Delta t}} S_{t+\Delta t} \right)^2 \right) \quad (136)$$

where $\left( \frac{\alpha_t}{\alpha_{t+\Delta t}} S_{t+\Delta t} \right)^2$ is chosen to match the leading term of $k_{\Delta t}(x \,|\, y)$. The constants can be estimated as follows

$$S_{t+\Delta t}^2 = \sigma_{t+\Delta t}^2 - \sigma_t^2 \frac{\alpha_{t+\Delta t}^2}{\alpha_t^2} = \sigma_{t+\Delta t}^2 - (\sigma_{t+\Delta t}^2 - 2\Delta t \sigma_{t+\Delta t} \frac{\partial \sigma_{t+\Delta t}}{\partial t} + o(\Delta t)) \frac{\alpha_{t+\Delta t}^2}{\alpha_t^2} \quad (137)$$

$$= \sigma_{t+\Delta t}^2 - (\sigma_{t+\Delta t}^2 - 2\Delta t \sigma_{t+\Delta t} \frac{\partial \sigma_{t+\Delta t}}{\partial t} + o(\Delta t)) \left( 1 + 2\Delta t \frac{\alpha_{t+\Delta t}^2}{\alpha_{t+\Delta t}^3} \frac{\partial \alpha_{t+\Delta t}}{\partial t} + o(\Delta t) \right) \quad (138)$$

$$= 2\Delta t \sigma_{t+\Delta t}^2 \frac{\partial}{\partial t} \log \frac{\sigma_{t+\Delta t}}{\alpha_{t+\Delta t}} + o(\Delta t) \,, \quad (139)$$

and

$$\frac{\alpha_t}{\alpha_{t+\Delta t}} = 1 + \frac{1}{\alpha_{t+\Delta t}} \frac{\partial \alpha_{t+\Delta t}}{\partial t} (-\Delta t) + o(\Delta t) = 1 - \Delta t \frac{\partial \log \alpha_{t+\Delta t}}{\partial t} + o(\Delta t) \,. \quad (140)$$

Thus, $y$ can be generated as

$$y = x - \Delta t \frac{\partial \log \alpha_{t+\Delta t}}{\partial t} x + 2\Delta t \sigma_{t+\Delta t}^2 \frac{\partial}{\partial t} \log \frac{\sigma_{t+\Delta t}}{\alpha_{t+\Delta t}} \nabla \log q_{t+\Delta t}(x) + \quad (141)$$

$$+ \sqrt{2\Delta t \sigma_{t+\Delta t}^2 \frac{\partial}{\partial t} \log \frac{\sigma_{t+\Delta t}}{\alpha_{t+\Delta t}}} \varepsilon \,, \quad (142)$$

where $\varepsilon$ is a standard normal random variable. This corresponds to the single step of the Euler discretization of the following SDE

$$dx_t = -\left( \frac{\partial}{\partial t} \log \alpha_t x - 2\sigma_t^2 \frac{\partial}{\partial t} \log \frac{\sigma_t}{\alpha_t} \nabla_x \log q_t(x) \right) \cdot \Delta t + \sqrt{2\sigma_t^2 \frac{\partial}{\partial t} \log \frac{\sigma_t}{\alpha_t}} dW_t \,. \quad (143)$$

For the KL-divergence between the reverse kernel and its Gaussian approximation, we have

$$D_{\mathrm{KL}}(\tilde{r}_{\Delta t}(y \,|\, x) \| r_{\Delta t}(y \,|\, x)) = \mathbb{E}_\varepsilon \log \frac{\tilde{r}_{\Delta t}(x + \Delta t \cdot v + g_{t+\Delta t} \sqrt{\Delta t} \varepsilon \,|\, x)}{r_{\Delta t}(x + \Delta t \cdot v + g_{t+\Delta t} \sqrt{\Delta t} \varepsilon \,|\, x)} + o(\Delta t) \,, \quad (144)$$

where

$$v = -\left( \frac{\partial}{\partial t} \log \alpha_t x - 2\sigma_t^2 \frac{\partial}{\partial t} \log \frac{\sigma_t}{\alpha_t} \nabla_x \log q_t(x) \right), \quad \text{and} \quad g_{t+\Delta t} = \sqrt{2\sigma_t^2 \frac{\partial}{\partial t} \log \frac{\sigma_t}{\alpha_t}} \,. \quad (145)$$

From Lemma 1, we have

$$D_{\mathrm{KL}} \left( \tilde{r}_{\Delta t}(y \,|\, x) \| r_{\Delta t}(y \,|\, x) \right) = \quad (146)$$

$$= \Delta t \left( \sigma_{t+\Delta t}^2 \frac{\partial}{\partial t} \log \frac{\sigma_{t+\Delta t}}{\alpha_{t+\Delta t}} \right) \mathbb{E}_\varepsilon \left( \varepsilon^T \frac{\partial^2 \log q_t(x)}{\partial x^2} \varepsilon - \Delta \log q_{t+\Delta t}(x) \right) + o(\Delta t) \quad (147)$$

$$= o(\Delta t) \,. \quad (148)$$

□

## D.4 PROOF OF THM. 3

**Theorem 3.** [Detailed balance density estimator error] *For $y = x + \Delta t \cdot v + g_{t+\Delta t}\sqrt{\Delta t}\varepsilon$,*
*where $g_{t+\Delta t} = \sqrt{2\sigma_t^2 \frac{\partial}{\partial t} \log \frac{\sigma_t}{\alpha_t}}$ and $\varepsilon$ is a standard normal random variable, the following*
*estimator is unbiased w.r.t. the generated samples y, i.e.*

$$\mathbb{E}_\varepsilon \left[ \log \frac{r_{\Delta t}(y \mid x)}{\tilde{r}_{\Delta t}(y \mid x)} \right] = o(\Delta t), \tag{85}$$

*and the variance is*

$$\mathbf{D}_\varepsilon \left[ \log \frac{r_{\Delta t}(y \mid x)}{\tilde{r}_{\Delta t}(y \mid x)} \right] = \Delta t^2 \frac{g_{t+\Delta t}^4}{4} \mathbf{D}_\varepsilon \left( \varepsilon^T \frac{\partial^2 \log q_t(x)}{\partial x^2} \varepsilon \right) + o(\Delta t^2), \tag{86}$$

*where $\frac{\partial^2 \log q_{t+\Delta t}(x)}{\partial x^2}$ is the Hessian matrix of the log-density.*

*Proof.* From [Lemma 1](#), we have

$$\mathbb{E}_\varepsilon \log \frac{r_{\Delta t}(y \mid x)}{\tilde{r}_{\Delta t}(y \mid x)} = \mathbb{E}_\varepsilon \Delta t \frac{g_{t+\Delta t}^2}{2} \left( \Delta \log q_{t+\Delta t}(x) - \varepsilon^T \frac{\partial^2 \log q_t(x)}{\partial x^2} \varepsilon \right) + o(\Delta t) \tag{149}$$

$$= \Delta t \frac{g_{t+\Delta t}^2}{2} \left( \Delta \log q_{t+\Delta t}(x) - \mathbb{E}_\varepsilon \operatorname{Tr} \left[ \varepsilon^T \frac{\partial^2 \log q_t(x)}{\partial x^2} \varepsilon \right] \right) + o(\Delta t) \tag{150}$$

$$= \Delta t \frac{g_{t+\Delta t}^2}{2} \left( \Delta \log q_{t+\Delta t}(x) - \mathbb{E}_\varepsilon \operatorname{Tr} \left[ \frac{\partial^2 \log q_t(x)}{\partial x^2} \right] \right) + o(\Delta t) \tag{151}$$

$$= o(\Delta t). \tag{152}$$

For the variance, we have

$$\mathbf{D}_\varepsilon \log \frac{r_{\Delta t}(y \mid x)}{\tilde{r}_{\Delta t}(y \mid x)} = \Delta t^2 \frac{g_{t+\Delta t}^4}{4} \mathbb{E}_\varepsilon \left( \Delta \log q_{t+\Delta t}(x) - \varepsilon^T \frac{\partial^2 \log q_t(x)}{\partial x^2} \varepsilon \right)^2 + o(\Delta t^2) \tag{153}$$

$$= \Delta t^2 \frac{g_{t+\Delta t}^4}{4} \mathbb{E}_\varepsilon \left( \mathbb{E}_\eta \left[ \eta^T \frac{\partial^2 \log q_t(x)}{\partial x^2} \eta \right] - \varepsilon^T \frac{\partial^2 \log q_t(x)}{\partial x^2} \varepsilon \right)^2 + o(\Delta t^2) \tag{154}$$

$$= \Delta t^2 \frac{g_{t+\Delta t}^4}{4} \mathbf{D}_\varepsilon \left( \varepsilon^T \frac{\partial^2 \log q_t(x)}{\partial x^2} \varepsilon \right) + o(\Delta t^2), \tag{155}$$

where we used a standard normal random variable $\eta$. □

### D.5 PROOF OF PROP. 11

**Proposition 11.** [Recurrence relation for the log-density] *For $y = x + \Delta t \cdot v + g_{t+\Delta t}\sqrt{\Delta t}\varepsilon$,*
*where $g_{t+\Delta t} = \sqrt{2\sigma_t^2 \frac{\partial}{\partial t} \log \frac{\sigma_t}{\alpha_t}}$ and $\varepsilon$ is a standard normal random variable, the estimator*
*from [Eq. (89)](#) can be expanded as follows*

$$-\log k_{\Delta t}(x \mid y) + \log \tilde{r}_{\Delta t}(y \mid x) = d\Delta t \frac{\partial}{\partial t} \log \alpha_{t+\Delta t} - \Delta t \frac{g_{t+\Delta t}^2}{2} \|\nabla \log q_{t+\Delta t}(x)\|^2 +$$

$$+ \left\langle \Delta t \cdot v + g_{t+\Delta t}\sqrt{\Delta t}\varepsilon + \Delta t \frac{\partial}{\partial t} \log \alpha_{t+\Delta t} x, \nabla \log q_{t+\Delta t}(x) \right\rangle + o(\Delta t) \tag{90}$$

*Proof.* Indeed,

$$\log \frac{\tilde{r}_{\Delta t}(y \mid x)}{k_{\Delta t}(x \mid y)} = \frac{1}{2B_{t+\Delta t}^2} \left( x - \frac{\alpha_{t+\Delta t}}{\alpha_t} y \right)^2 + \frac{d}{2} \log 2\pi S_{t+\Delta t}^2 - \tag{156}$$

$$- \frac{1}{2(S_{t+\Delta t \frac{\alpha_t}{\alpha_{t+\Delta t}}})^2} \left( y - \frac{\alpha_t}{\alpha_{t+\Delta t}} x - \frac{\alpha_t}{\alpha_{t+\Delta t}} S_{t+\Delta t}^2 \nabla \log q_{t+\Delta t}(x) \right)^2 - \tag{157}$$

$$- \frac{d}{2} \log 2\pi S_{t+\Delta t}^2 - d \log \frac{\alpha_t}{\alpha_{t+\Delta t}} \tag{158}$$

$$= -d \log \frac{\alpha_t}{\alpha_{t+\Delta t}} + \left\langle \frac{\alpha_{t+\Delta t}}{\alpha_t} y - x, \nabla \log q_{t+\Delta t}(x) \right\rangle - \tag{159}$$

$$- \frac{S_{t+\Delta t}^2}{2} \|\nabla \log q_{t+\Delta t}(x)\|^2 \tag{160}$$

Expanding the time around $t + \Delta t$, we have

$$\log \frac{\tilde{r}_{\Delta t}(y \mid x)}{k_{\Delta t}(x \mid y)} = d\Delta t \frac{\partial}{\partial t} \log \alpha_{t+\Delta t} + \left\langle (1 + \Delta t \frac{\partial}{\partial t} \log \alpha_{t+\Delta t}) y - x, \nabla \log q_{t+\Delta t}(x) \right\rangle - \tag{161}$$

$$- \Delta t \left( \sigma_{t+\Delta t}^2 \frac{\partial}{\partial t} \log \frac{\sigma_{t+\Delta t}}{\alpha_{t+\Delta t}} \right) \|\nabla \log q_{t+\Delta t}(x)\|^2 + o(\Delta t). \tag{162}$$

For $y = x + \Delta t \cdot v + \sqrt{\Delta t} g_{t+\Delta t} \varepsilon$, $g_{t+\Delta t} = \sqrt{2\sigma_{t+\Delta t}^2 \frac{\partial}{\partial t} \log \frac{\sigma_{t+\Delta t}}{\alpha_{t+\Delta t}}}$, we have

$$\log \frac{\tilde{r}_{\Delta t}(y \mid x)}{k_{\Delta t}(x \mid y)} = d\Delta t \frac{\partial}{\partial t} \log \alpha_{t+\Delta t} - \Delta t \frac{g_{t+\Delta t}^2}{2} \|\nabla \log q_{t+\Delta t}(x)\|^2 + \tag{163}$$

$$+ \left\langle \Delta t \cdot v + g_{t+\Delta t} \sqrt{\Delta t} \varepsilon + \Delta t \frac{\partial}{\partial t} \log \alpha_{t+\Delta t} x, \nabla \log q_{t+\Delta t}(x) \right\rangle + o(\Delta t). \tag{164}$$

$\square$

# E    ADDITIONAL RELATED WORK

**Protein generation**. Structure-based de novo protein design using deep generative models has recently seen a surge in interest, with a particular emphasis on diffusion-based approaches (Watson et al., 2023; Yim et al., 2023b), and also flow matching methods (Bose et al., 2024; Yim et al., 2023a; 2024; Huguet et al., 2024). Moreover, building on the initial SE(3) equivariant diffusion paradigm multiple recent approaches have sought to increase the performance of the methods through architectural innovations (Wang et al., 2024a), conditioning on auxiliary modalities such as sequence or sidechains (Ingraham et al., 2023; Lin et al., 2024). Finally, recent approaches tackle the problem of co-generation which seeks to define a joint inference procedure over both structure and sequences (Campbell et al., 2024; Ren et al., 2024; Lisanza et al., 2023), but remains distinct from the setting of this work which attempts to combine different pre-trained models using superposition.

# F    BROADER IMPACTS

In this paper, we present theoretical results and demonstrate use cases in generation tasks such as image generation and unconditional protein generation. Because of the theoretical nature of our contributions, this work carries little societal impact. SUPERDIFF can be used to generated protein structure from a composition of existing protein diffusion models. Better protein generation methods can potentially lead to negative use in generating bio-hazardous molecules and proteins. We do not perceive this as a great risk at the current stage of these models.

# G    ADDITIONAL RESULTS FOR CIFAR-10

# H    PROTEIN GENERATION

## H.1    EXPERIMENTAL DETAILS

In our setting, we consider two state-of-the-art diffusion models for protein generation: Proteus (Wang et al., 2024a) and FrameDiff (Yim et al., 2023b), which were trained on protein structures from

Table A1: Image generation performance for CIFAR-10 with *conditional* models trained on two random partitions of the training data (labeled A and B). We compare SUPERDIFF (**OR**) with the respective models (model$_A$ and model$_B$) with the model that is trained on the full dataset (model$_{A \cup B}$) and random choice between two models (model$_{A \text{ OR } B}$).

| | ODE inference | | | SDE inference | | |
|---|---|---|---|---|---|---|
| | FID ($\downarrow$) | IS ($\uparrow$) | FLD ($\downarrow$) | FID ($\downarrow$) | IS ($\uparrow$) | FLD ($\downarrow$) |
| model$_A$ | 4.75 | 8.98 | $6.95 \pm 0.12$ | 4.66 | 9.35 | $6.39 \pm 0.13$ |
| model$_B$ | 4.78 | 8.97 | $6.86 \pm 0.15$ | 4.36 | **9.45** | $\mathbf{6.20 \pm 0.14}$ |
| model$_{A \cup B}$ | 5.30 | **9.04** | $6.82 \pm 0.09$ | **2.83** | 9.44 | $6.26 \pm 0.11$ |
| model$_{A \text{ OR } B}$ | 4.75 | 8.94 | $6.86 \pm 0.15$ | 4.41 | 9.40 | $6.3 \pm 0.18$ |
| SUPERDIFF (**OR**) | 4.74 | 9.00 | $6.98 \pm 0.10$ | 4.46 | 9.40 | $6.22 \pm 0.15$ |
| SUPERDIFF $_{T=100}$ (**OR**) | **4.63** | 8.98 | $\mathbf{6.81 \pm 0.19}$ | 4.23 | 9.42 | $6.27 \pm 0.12$ |

Protein Data Bank (PDB, (Berman et al., 2000)) to estimate the special Euclidean group (SE(3)) equivariant score over multiple diffusion steps. The models' outputs predict the coordinates of a monomeric protein backbone.

We use pre-trained checkpoints from Proteus[3] and FrameDiff[4]. During protein generation at inference, we separately combine scores for translations and rotations from both models using Algorithm 1.

We investigate the use of different temperature ($T$) settings to scale $\kappa$ for controlling the densities. We also found that adding a small bias ($\omega$) towards Proteus log densities improved designability.

## H.2 METRICS FOR EVALUATING GENERATED PROTEINS

**Designability**. We assess designability using the *self-consistency* evaluation from Trippe et al. (2023), where given a generated backbone, we predict its scaffold using a sequence prediction model (we use ProteinMPNN (Dauparas et al., 2022)) and re-fold the sequence using a structure prediction model (we use ESMFold (Lin et al., 2022)). We then compare the re-folded protein to the original generated backbone by computing their template modeling score (scTM) and root-mean-square-distance (scRMSD). A protein is considered to be designable if its scRMSD is $< 2\text{Å}$ to the refolded structure. For each protein, we repeat this process 8 times and keep the sequence with the lowest scRMSD. We report the fraction of designable proteins for each method in Table 3 as well as the scRMSD mean of the resulting *designable* proteins.

**Novelty**. A significant impetus for generative modelling in biology and chemistry is to propose compounds that have not been previously identified (i.e., different from the training data), but are also possible to make. We compute three novelty metrics: the fraction of designable proteins with a scTM score $< 0.5$ (higher is better), the fraction of designable proteins with a scTM score $< 0.3$, and the mean scTM score between generated proteins and the proteins from PDB that the original diffusion models were trained on, which represents the collective human knowledge of protein structures; a lower score is indicates higher distance from the training set and is desirable since it shows generalization ability.

**Diversity**. To measure how diverse the generated proteins are, we compute their mean pairwise scTM score (lower is better), and also report the fraction of unique clusters formed after clustering them with MaxCluster (Herbert & Sternberg, 2008) (higher is better). Finally, we report the fraction of proteins that contain $\beta$-sheet secondary structures, as it has been found that these structures are typically more rare to generate (Bose et al., 2024).

## H.3 PROTEIN DIVERSITY EXPLORATION

To embed and cluster the generated protein backbones we use the Foldseek (Van Kempen et al., 2024) package to compute pairwise aligned TM-scores. We then use the UMAP (McInnes et al., 2018) package to compute a 2D embedding. Where proteins are represented as points that are close to each other if they are structurally similar (by aligned TM-Score).

We then clustered the proteins again with the Foldseek tool to find representative structures. Finally we used KMeans to explore the space and narrow down which protein structures belong where on the protein structure manifold.

---

[3] https://github.com/Wangchentong/Proteus
[4] https://github.com/jasonkyuyim/se3_diffusion

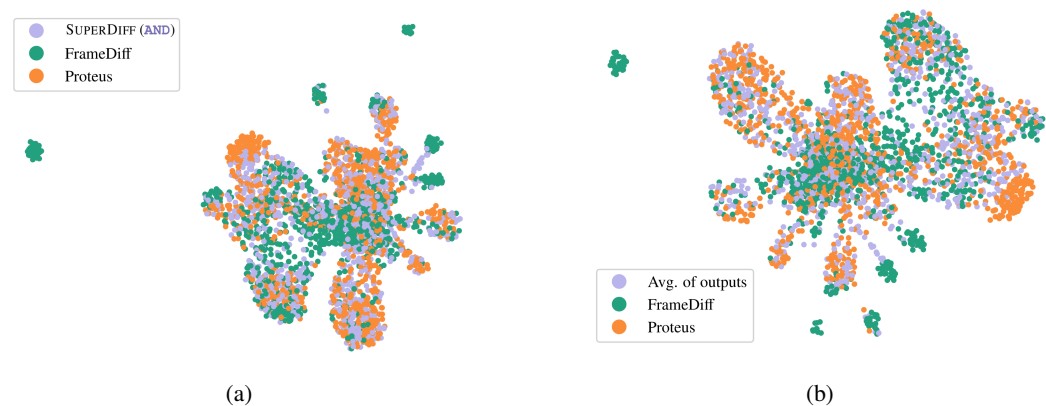

Figure A1: UMAP visualizations of protein structures with (a) SUPERDIFF (AND) and (b) averaging of outputs.

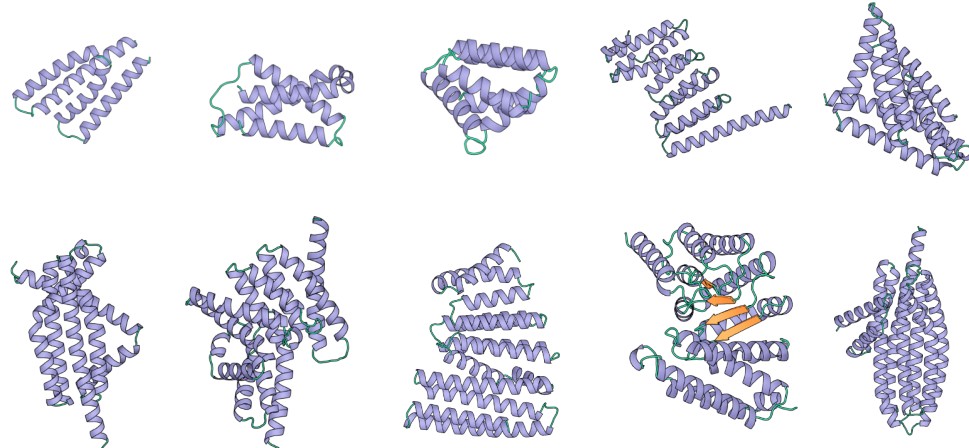

Figure A2: Proteins generated by SUPERDIFF (AND) with scTM score $< 0.3$.

In Fig. A1, we show UMAP visualizations of proteins generated with SUPERDIFF (AND) and averaging of outputs (Liu et al., 2022a).

### H.4 PROTEIN NOVELTY EXPLORATION

In Fig. A2, we visualize the proteins generated by SUPERDIFF (AND) that are furthest away from the set of known proteins (scTM score $< 0.3$).

## I SMALL MOLECULE GENERATION

### I.1 VISUALIZING TOP-PERFORMING MOLECULES

In Fig. A3, we display the best molecules generated with the objective of GSK3$\beta$ inhibition and QED. We evaluate the performance of molecules according to both objectives by taking the product of the individual objectives. In Fig. A4, we display the best molecules generated with the objective of high GSK3$\beta$ inhibition and JNK3 inhibition.

## J GENERATING IMAGE COMPOSITIONS WITH STABLE DIFFUSION

For our concept interpolation experiments, we use publicly-available pre-trained weights, models, and schedulers from Stable Diffusion v1-4 https://huggingface.co/CompVis/stable-diffusion-v1-4.

In Figs. A5–A24, we show examples of image compositions generated by SUPERDIFF (AND), averaging of outputs, and joint prompting. Prompts are shown in each image caption. We show

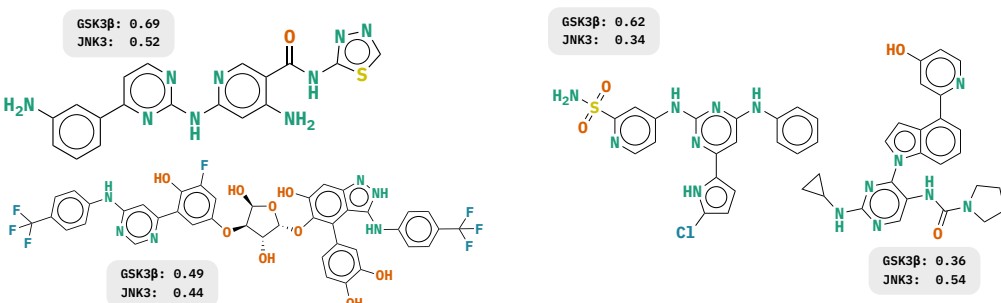

(a) Molecules generated by SUPERDIFF (AND).

(b) Molecules generated by averaging scores.

Figure A3: Best molecules generated by (a) SUPERDIFF (AND) and (b) averaging scores (Liu et al., 2022a) where target properties are high GSK3$\beta$ inhibition and high QED.

(a) Molecules generated by SUPERDIFF (AND).

(b) Molecules generated by averaging scores.

Figure A4: Best molecules generated by (a) SUPERDIFF (AND) and (b) averaging scores (Liu et al., 2022a) where where target properties are high GSK3$\beta$ inhibition and high JNK3 inhibition.

images generated by the first 6 seeds (uniform sampling), as well as our favourite images generated from 20 seeds. For joint prompting, we generate prompts from two concepts using the linking term `"that looks like"`. For example, given the concepts `"a lemon"` and `"a sunflower"`, the resulting prompt would be `"a sunflower that looks like a lemon"`. We also generate images with the reversed prompt (`"a lemon that looks like a sunflower"`), and keep the images generated by the prompt resulting in the higher mean score for each metric. The order of the concepts in each image caption reflects the ordering that obtained the higher TIFA score.

## J.1 STABLE DIFFUSION WITH SUPERDIFF (OR)

**Baselines for concept selection (OR).** For the OR setting, our baseline is prompting SD that prompts the model to select between two concepts: `"a sunflower or a lemon"`. As with AND, we also flip the prompt and keep the better image for each metric. As an upper bound, we generate images from SD by prompting it with a random choice between the two concepts (Prompt$_{A \text{ OR } B}$).

Table A2: Quantative evaluation of SD-generated images for logical **OR**. We compare SUPERDIFF (**OR**), joint prompting, and an upper bound of randomly selecting a single concept prompt (Prompt$_{A \text{ OR } B}$). We report the maximum scores from each prompt pair, as well as the absolute difference between the maximum and minimum scores ($|\Delta|$). These metrics reflect how well a method can select one concept to generate.

| | $|\Delta|$/Max. CLIP ($\uparrow$) | $|\Delta|$/Max. ImageReward ($\uparrow$) | $|\Delta|$/Max. TIFA ($\uparrow$) | |
|---|---|---|---|---|
| Prompt$_{A \text{ OR } B}$ (uncorrelated random choice) | 9.13/29.72 | 2.87/0.70 | 88.21/97.58 | |
| Joint prompting | 7.20/29.80 | 2.47/0.59 | 79.46/**97.92** | |
| SUPERDIFF (**OR**) | **8.58**/**29.87** | **2.76**/**0.64** | **84.10**/95.83 | |

**SUPERDIFF qualitatively generates images with better concept interpolations and selections**. In Fig. A25, we show examples of image compositions generated by SUPERDIFF (**OR**), joint prompting, and an upper bound of randomly selecting a prompt of one concept (coin flip). As with SUPERDIFF (**AND**), we keep the prompt order that resulted in higher scores for the baseline. We find that SUPERDIFF (**OR**) can faithfully generate images with a single concept. The joint prompting baseline can sometimes generate images that combine fragments of both concepts, but other times it also generates images of a single concept, typically the first concept in the joint prompt (this also underscores why this method struggles with concept interpolation).

We evaluate SUPERDIFF (**OR**) using the same three metrics as SUPERDIFF (**AND**) (CLIP Score, ImageReward, and TIFA) and display the results in Table A2. We again evaluate the image against each concept prompt separately and take both the maximum score and the absolute difference between both scores for each metric. This is so that we can measure how well *one* concept is represented. The upper bound for this setting is randomly prompting SD with either of the prompts; we find that we are almost able to match this setting across all scores, indicating that our method is able to faithfully select a single concept. SD with joint prompting does not perform as well, as nothing prevents it from combining components from both concepts.

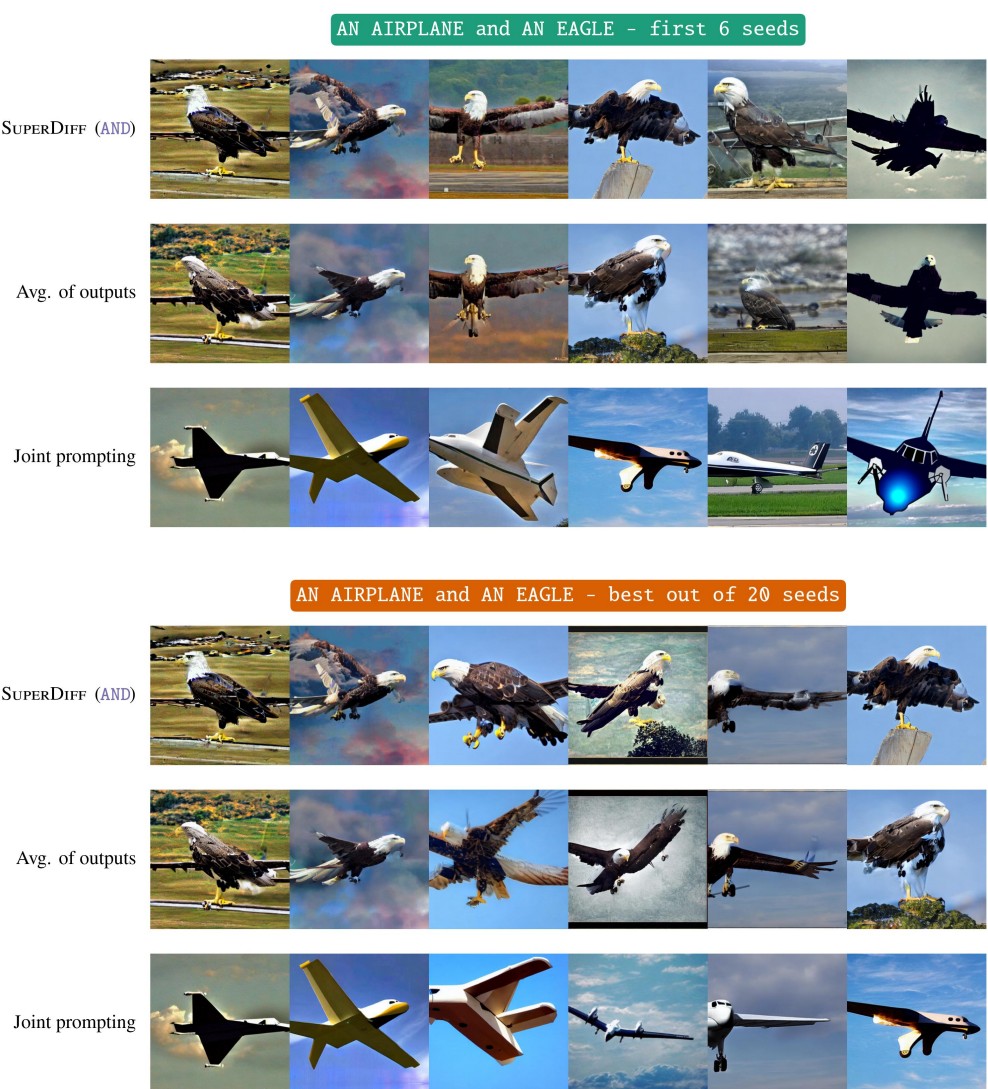

Figure A5: Image compositions generated using SUPERDIFF (AND), averaging of outputs, or joint prompting for the concepts "an airplane" and "an eagle".

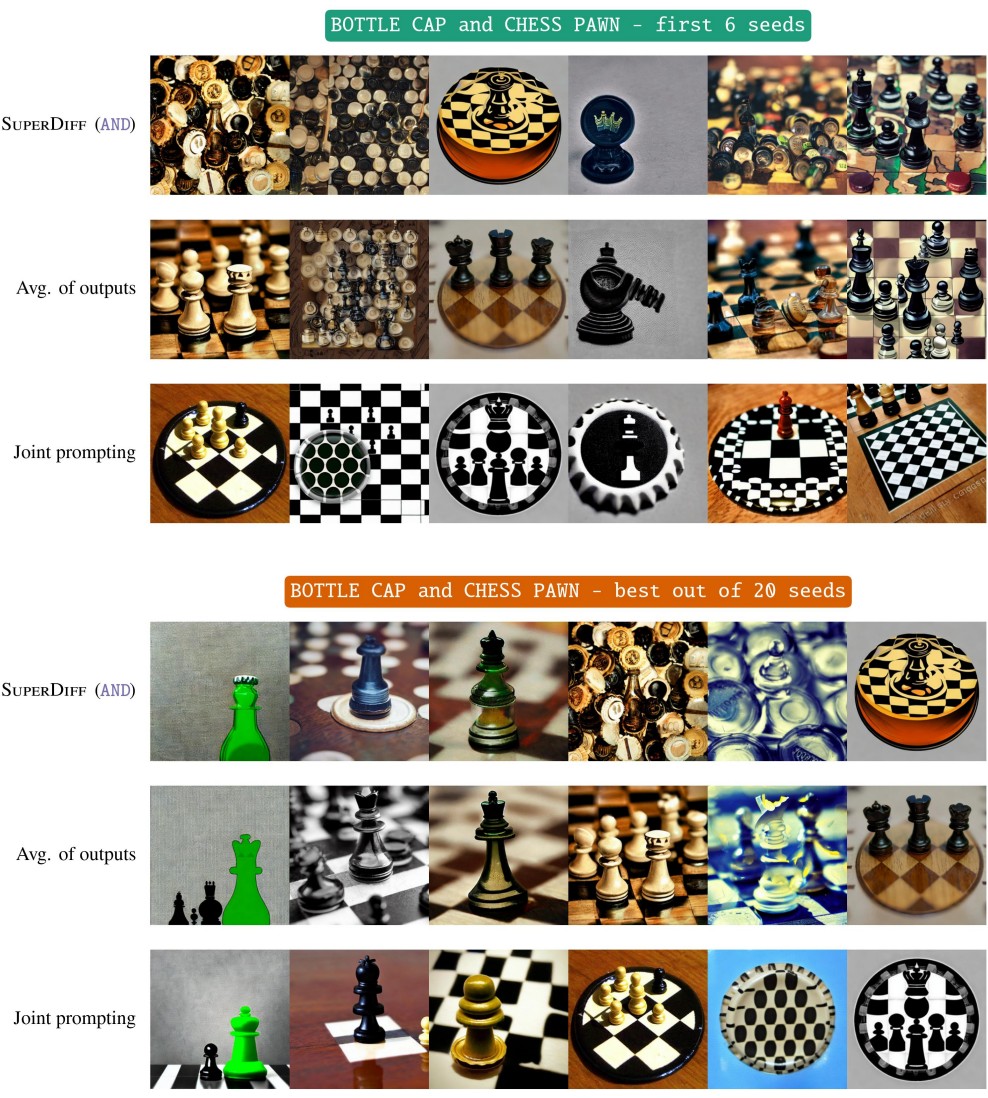

Figure A6: Image compositions generated using SUPERDIFF (AND), averaging of outputs, or joint prompting for the concepts "bottle cap" and "chess pawn".

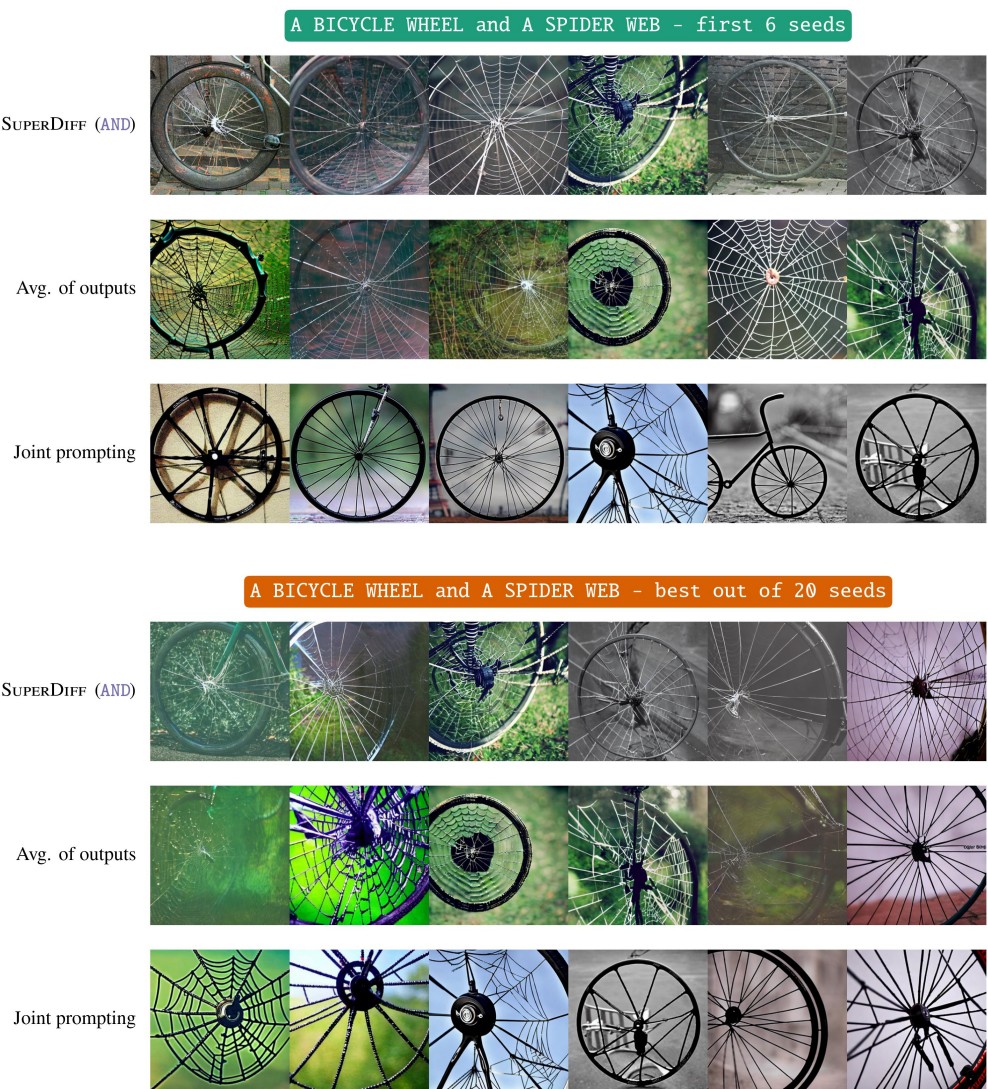

Figure A7: Image compositions generated using SUPERDIFF (AND), averaging of outputs, or joint prompting for the concepts "a bicycle wheel" and "a spider web".

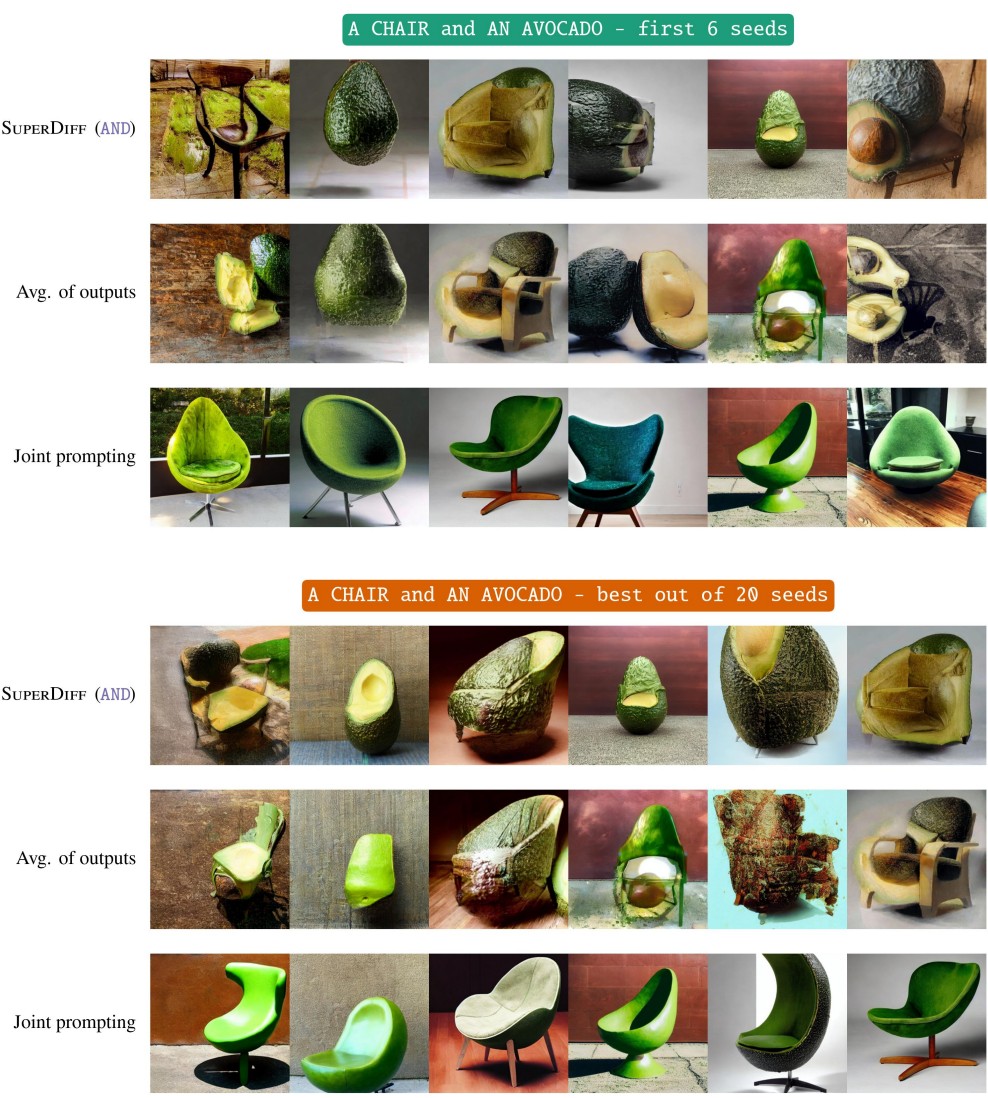

Figure A8: Image compositions generated using SUPERDIFF (AND), averaging of outputs, or joint prompting for the concepts "a chair" and "an avocado".

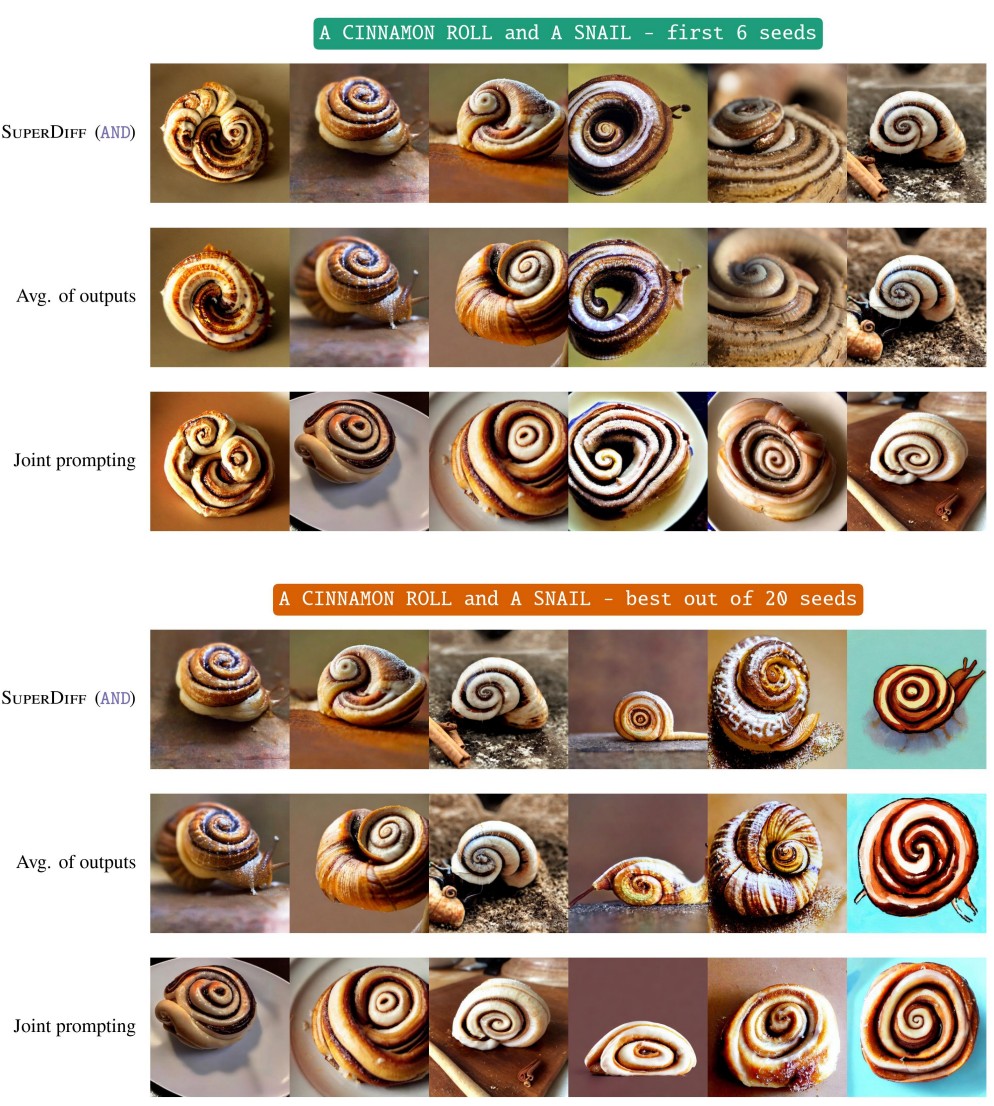

Figure A9: Image compositions generated using SUPERDIFF (**AND**), averaging of outputs, or joint prompting for the concepts `"a cinnamon roll"` and `"a snail"`.

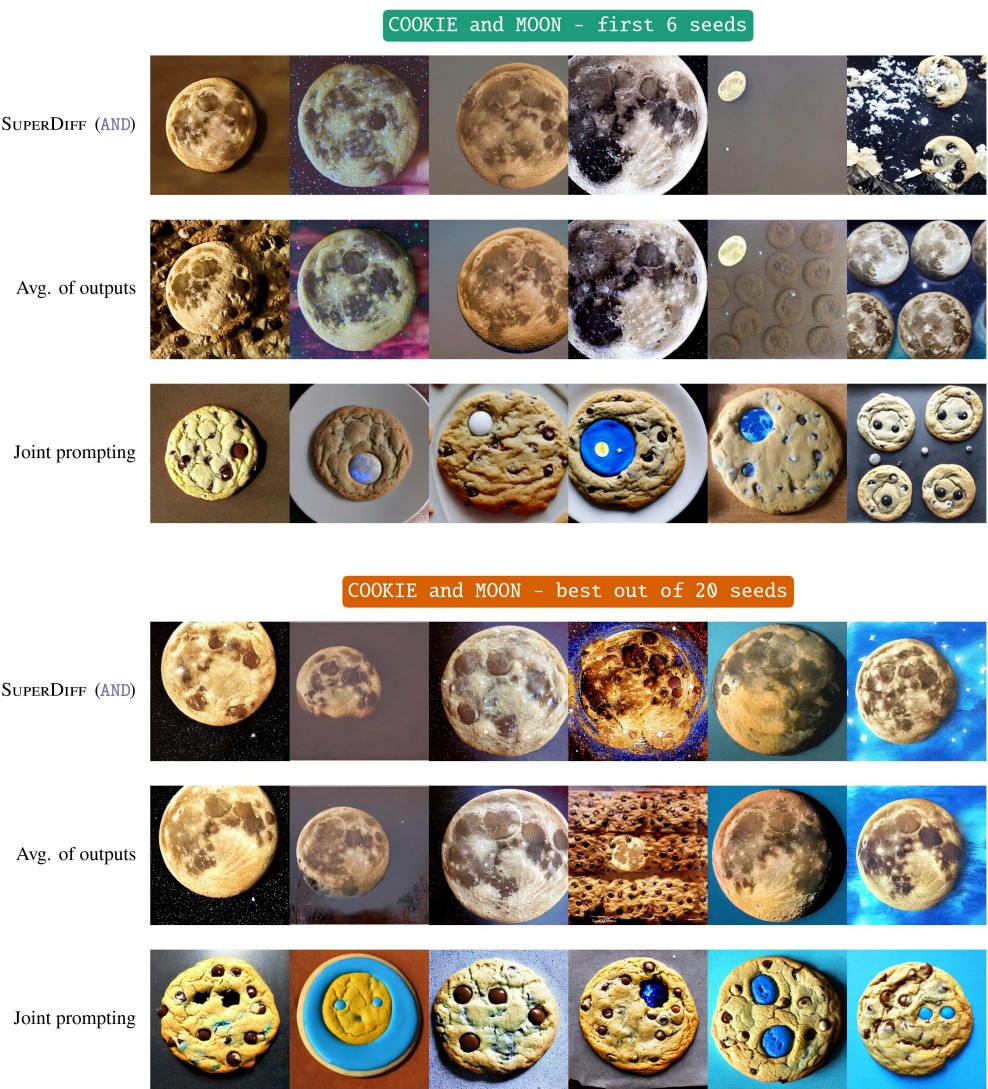

Figure A10: Image compositions generated using SUPERDIFF (AND), averaging of outputs, or joint prompting for the concepts "cookie" and "moon".

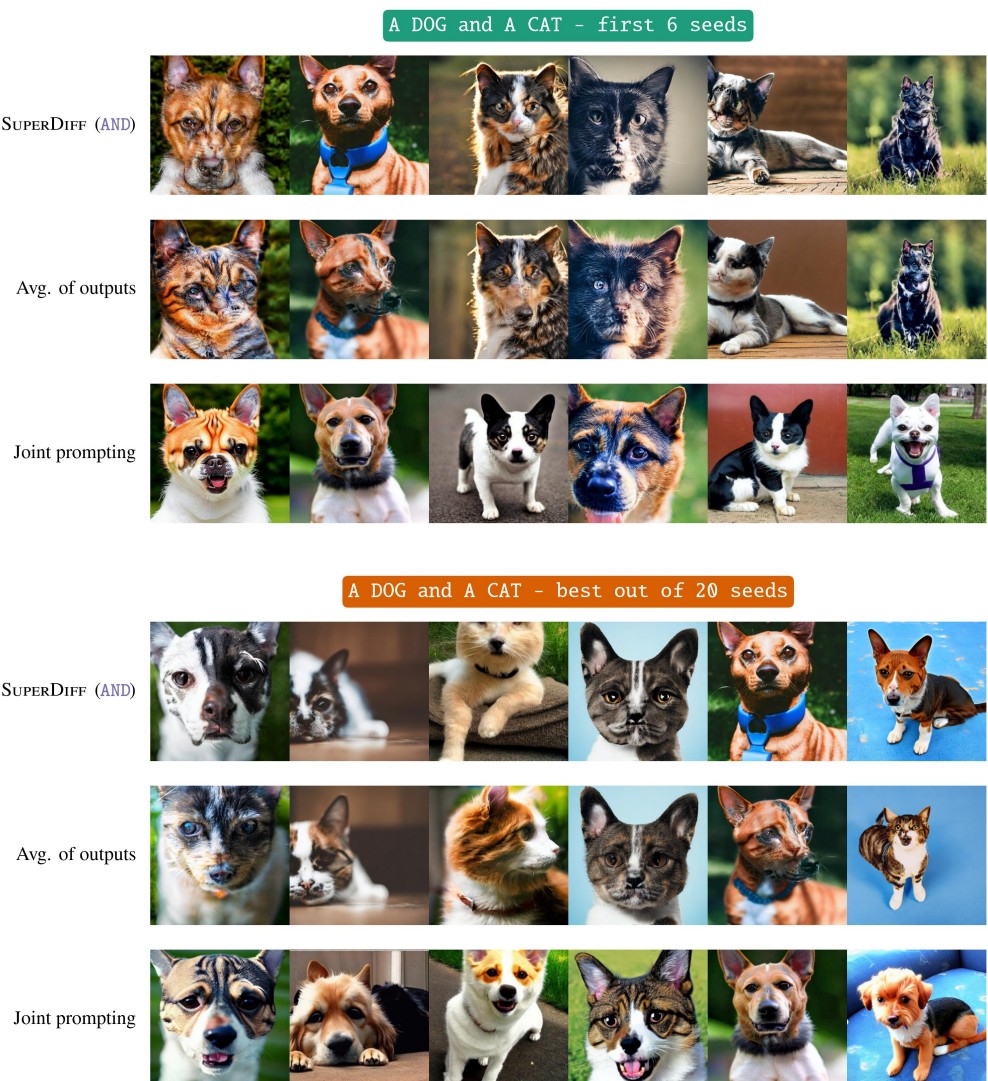

Figure A11: Image compositions generated using SUPERDIFF (AND), averaging of outputs, or joint prompting for the concepts "a dog" and "a cat".

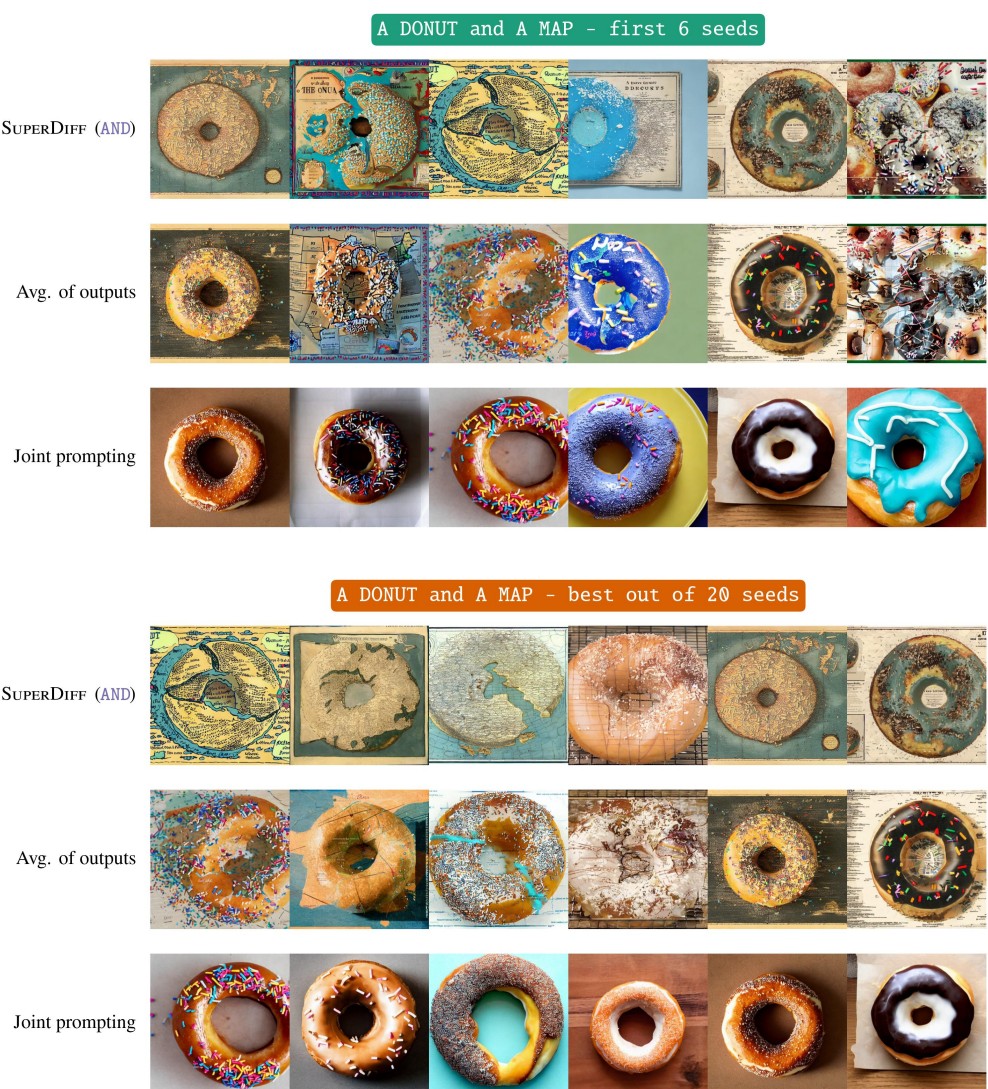

Figure A12: Image compositions generated using SUPERDIFF (AND), averaging of outputs, or joint prompting for the concepts "a donut" and "a map".

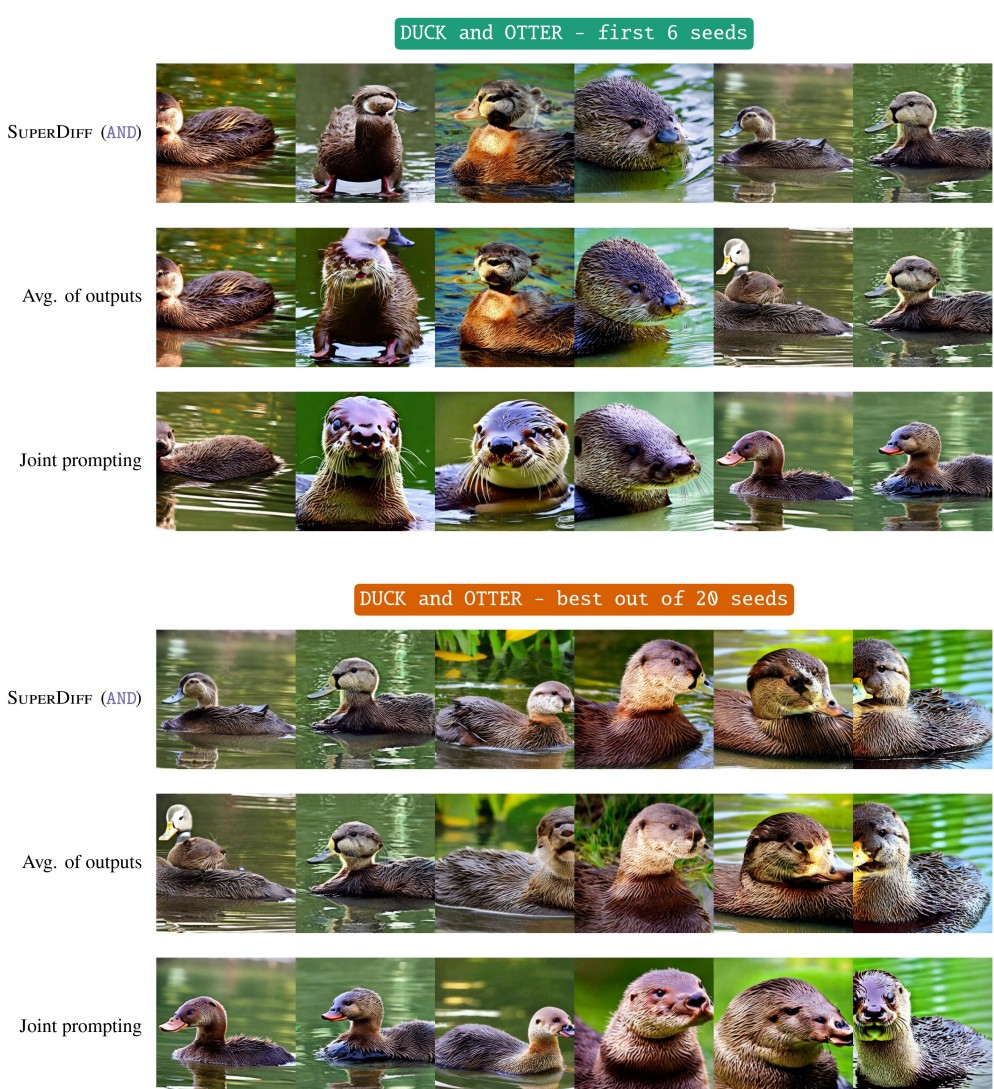

Figure A13: Image compositions generated using SUPERDIFF (AND), averaging of outputs, or joint prompting for the concepts "duck" and "otter".

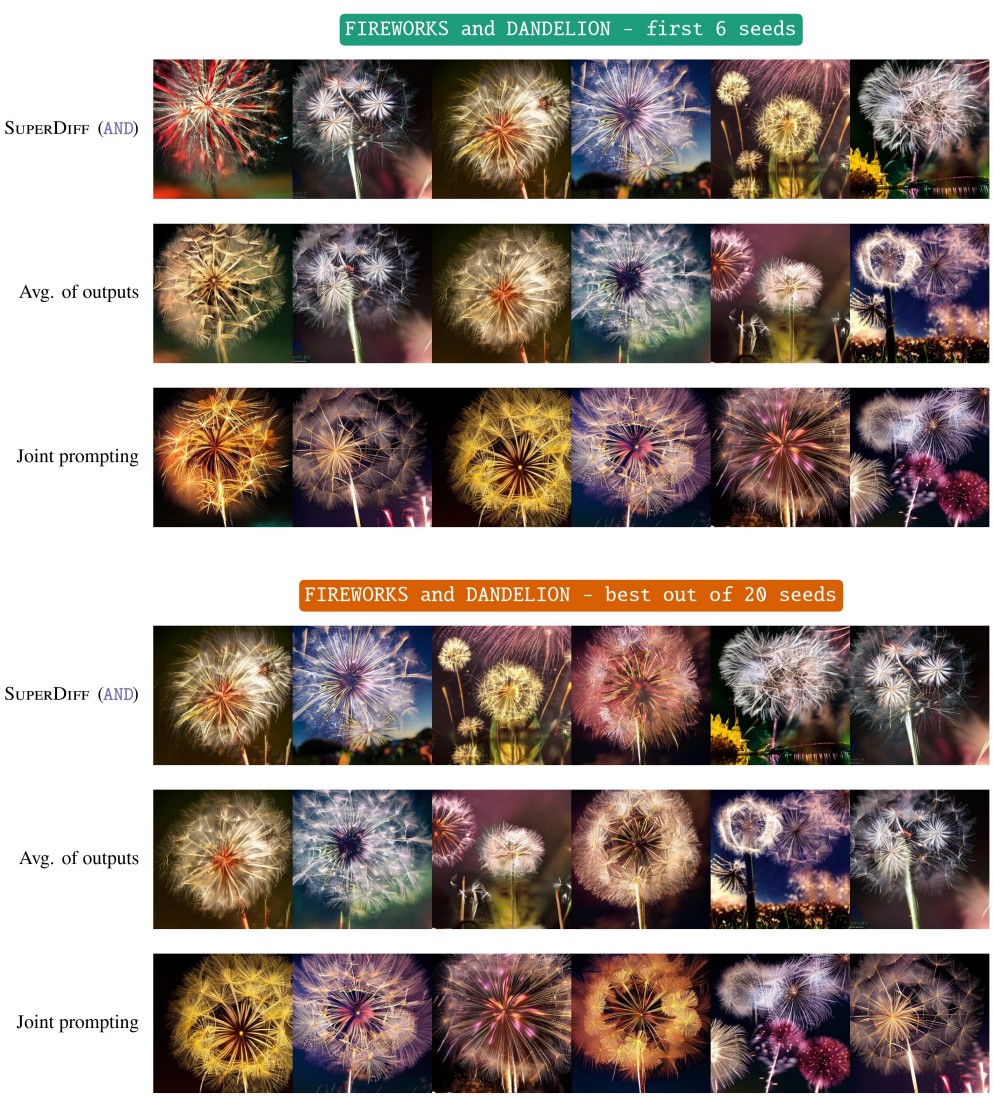

Figure A14: Image compositions generated using SUPERDIFF (AND), averaging of outputs, or joint prompting for the concepts "fireworks" and "dandelion".

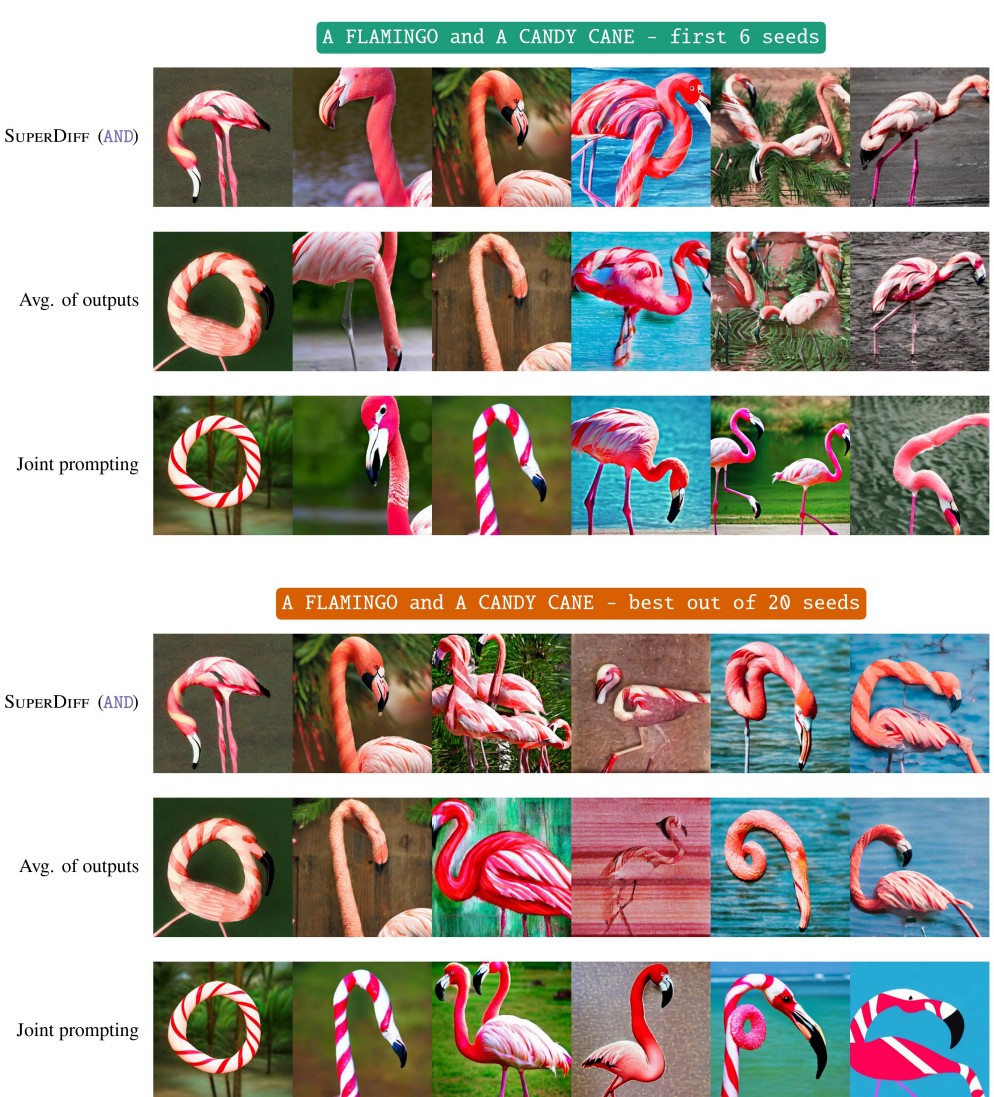

Figure A15: Image compositions generated using SUPERDIFF (AND), averaging of outputs, or joint prompting for the concepts "a flamingo" and "a candy cane".

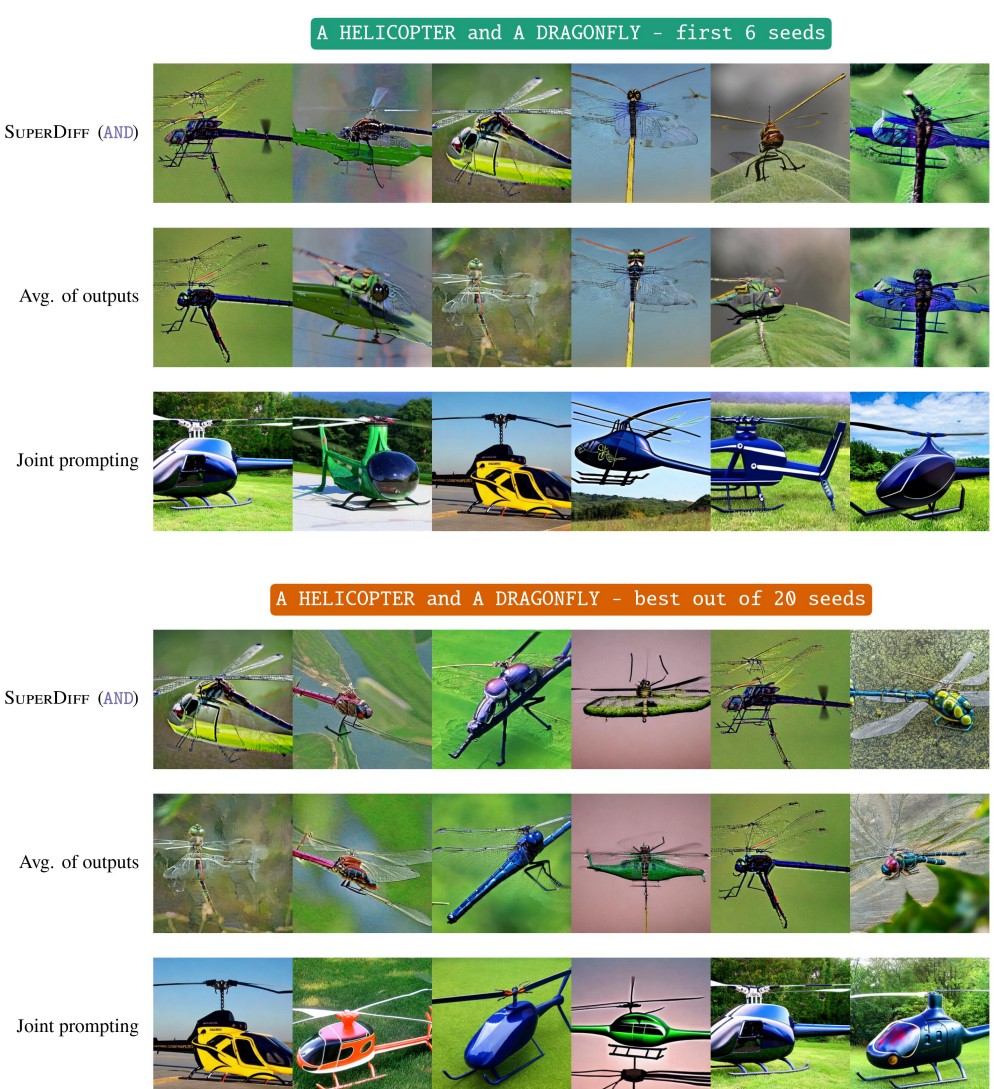

Figure A16: Image compositions generated using SUPERDIFF (AND), averaging of outputs, or joint prompting for the concepts "a helicopter" and "a dragonfly".

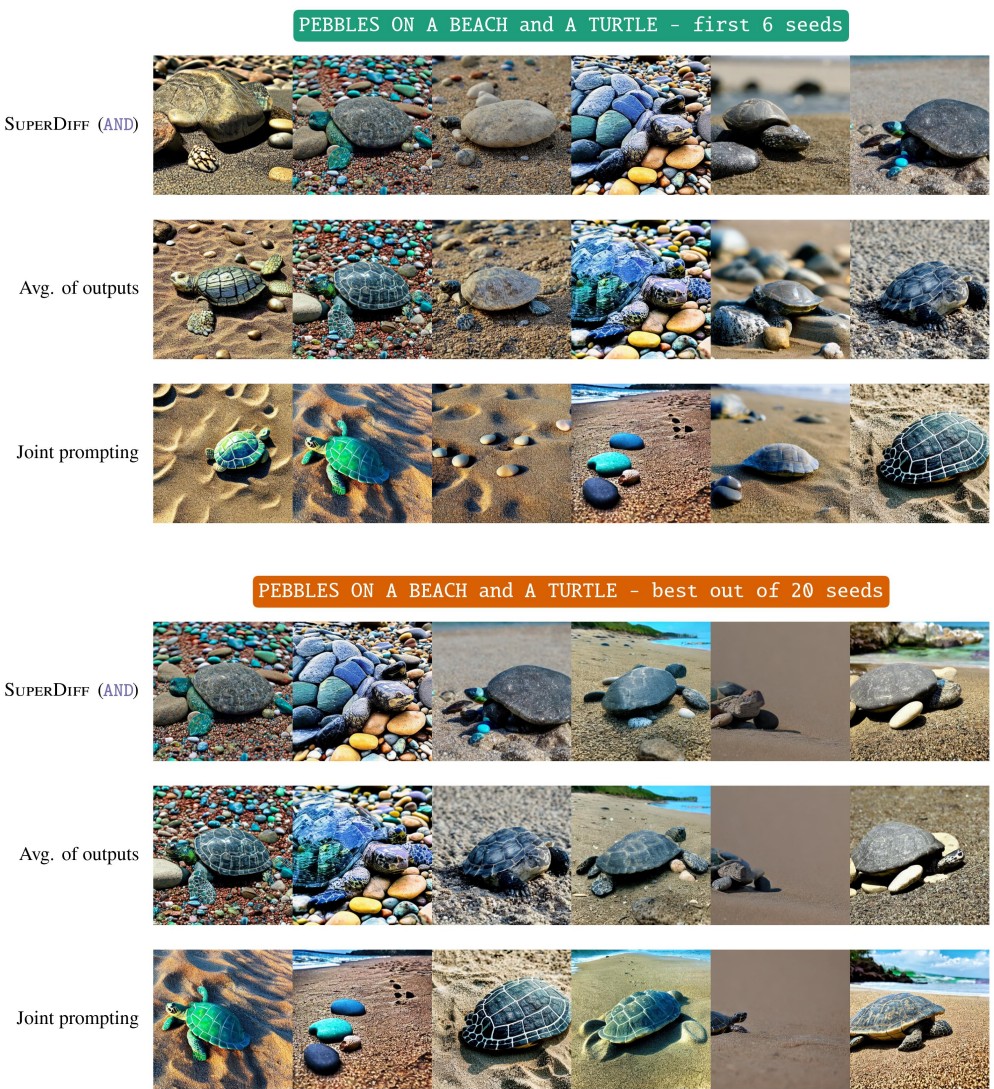

Figure A17: Image compositions generated using SUPERDIFF (AND), averaging of outputs, or joint prompting for the concepts "pebbles on a beach" and "a turtle".

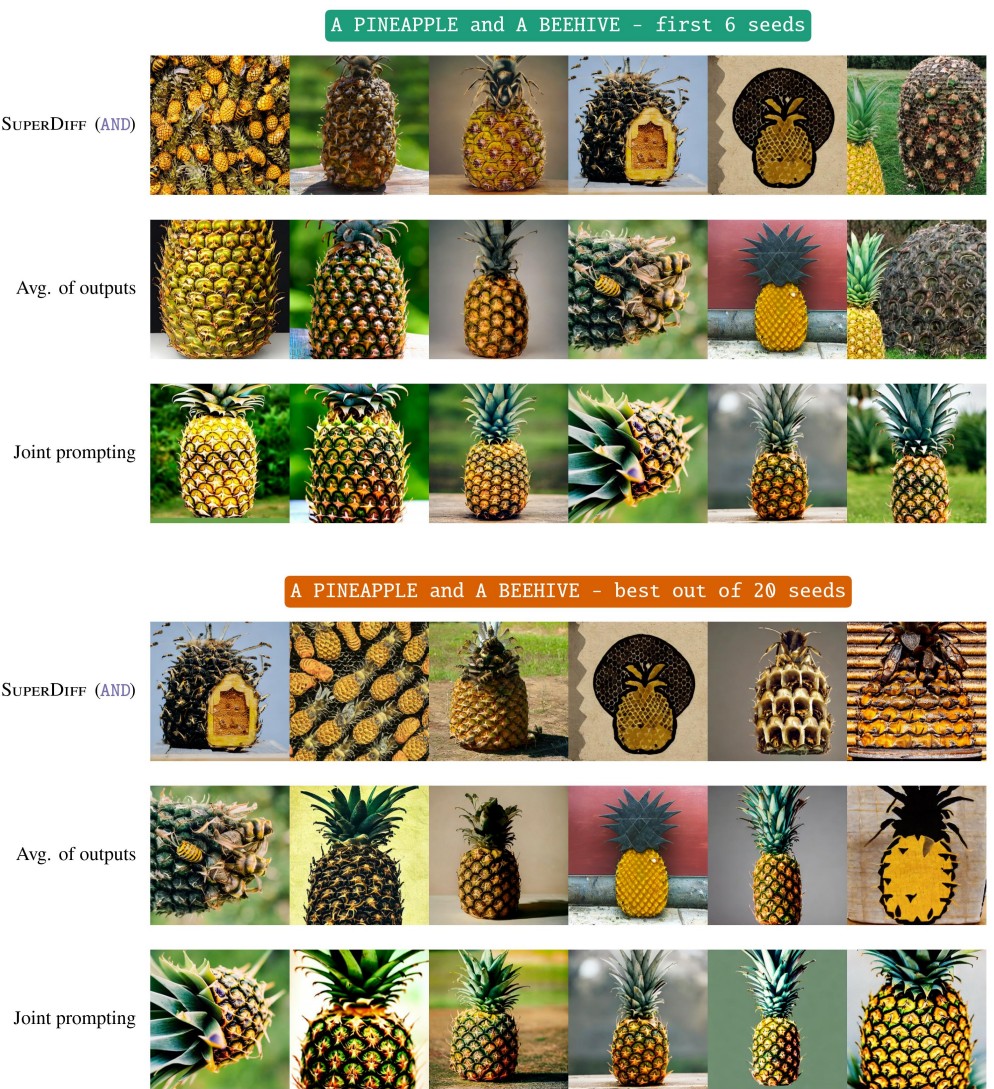

Figure A18: Image compositions generated using SUPERDIFF (AND), averaging of outputs, or joint prompting for the concepts "a pineapple" and "a beehive".

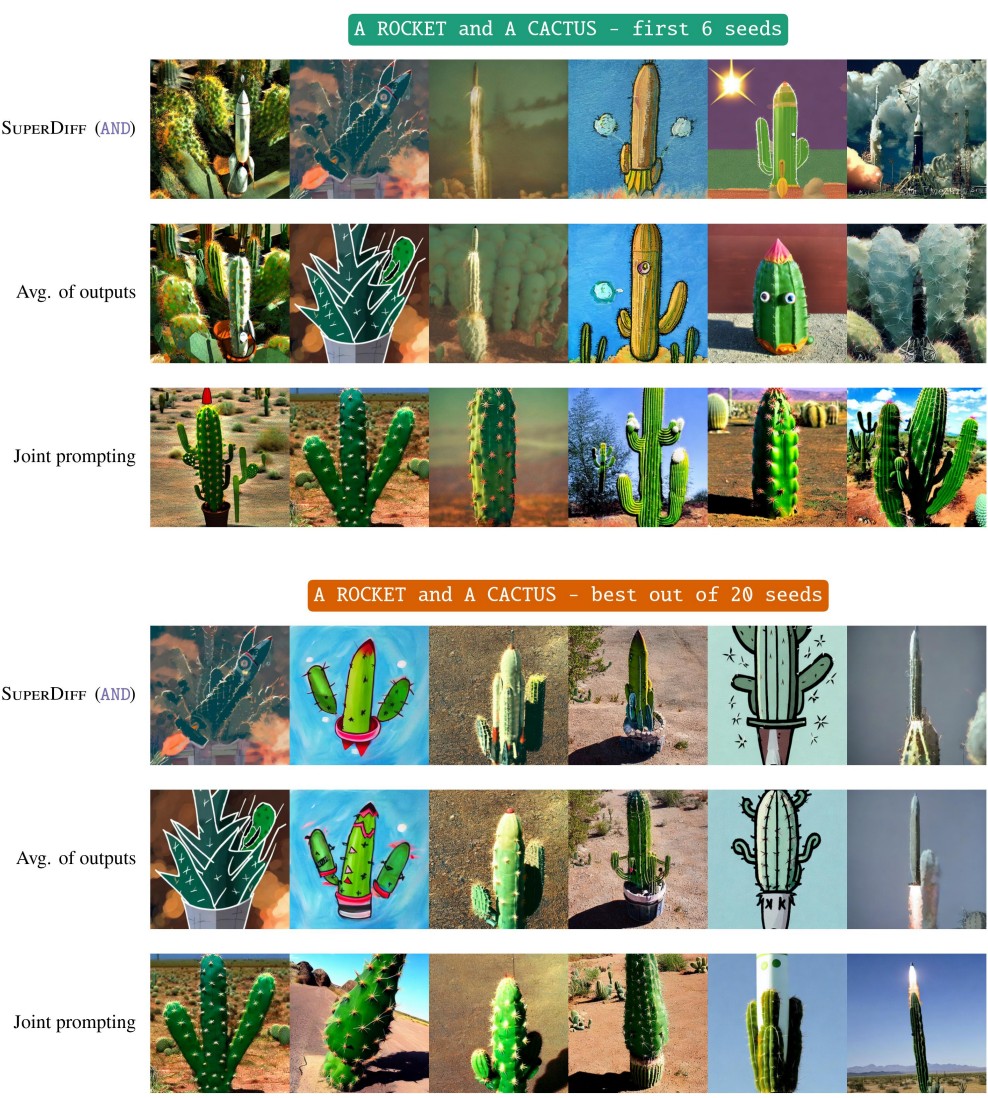

Figure A19: Image compositions generated using SUPERDIFF (AND), averaging of outputs, or joint prompting for the concepts "a rocket" and "a cactus".

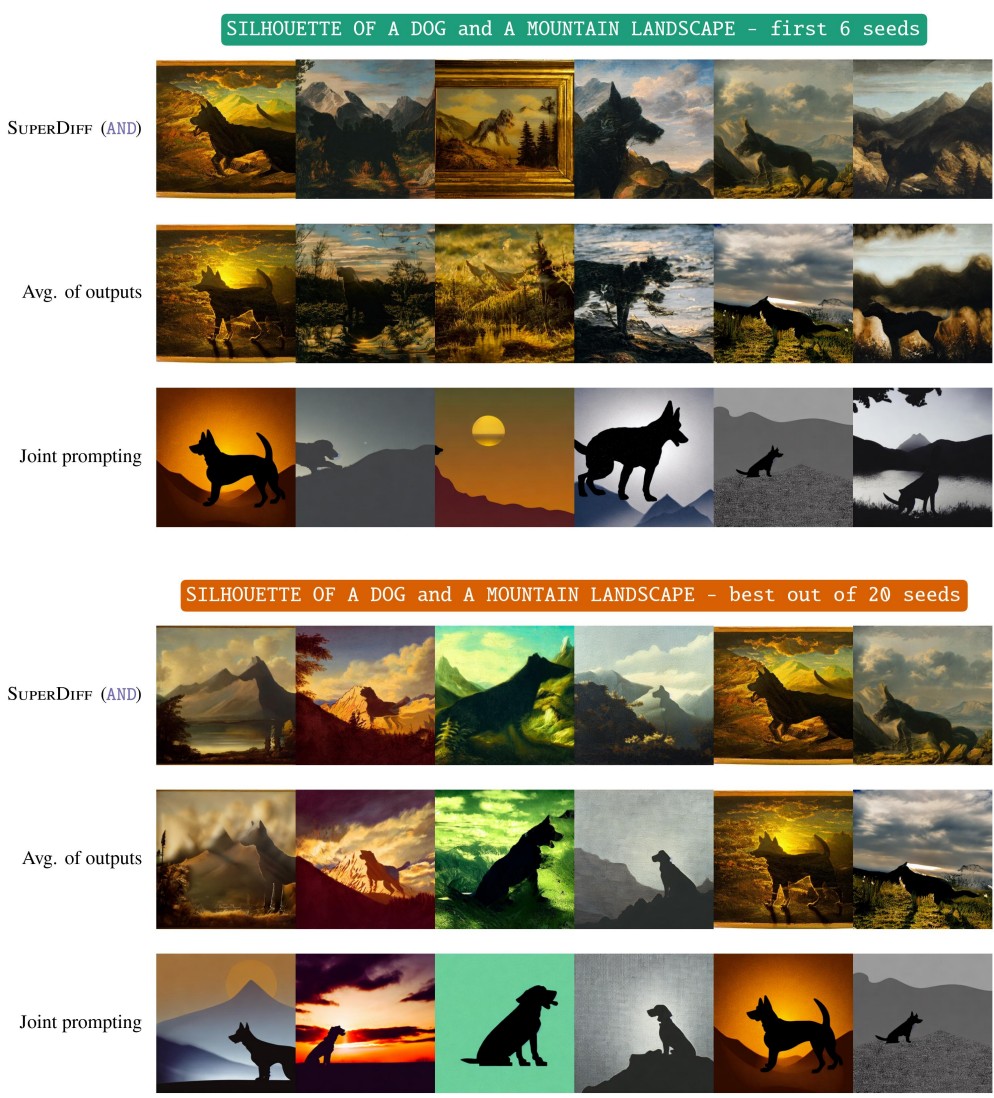

Figure A20: Image compositions generated using SUPERDIFF (AND), averaging of outputs, or joint prompting for the concepts "a silhouette of a dog" and "a mountain landscape".

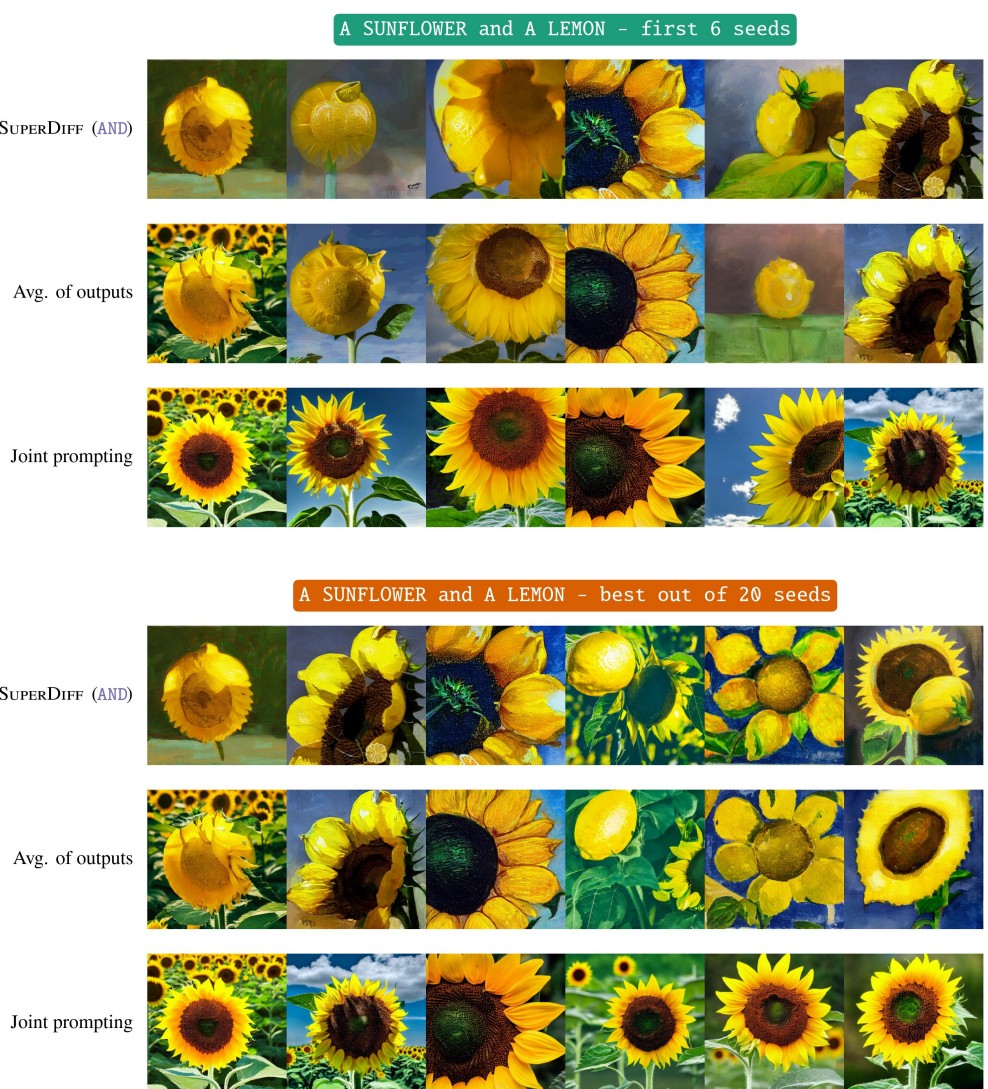

Figure A21: Image compositions generated using SUPERDIFF (AND), averaging of outputs, or joint prompting for the concepts "a sunflower" and "a lemon".

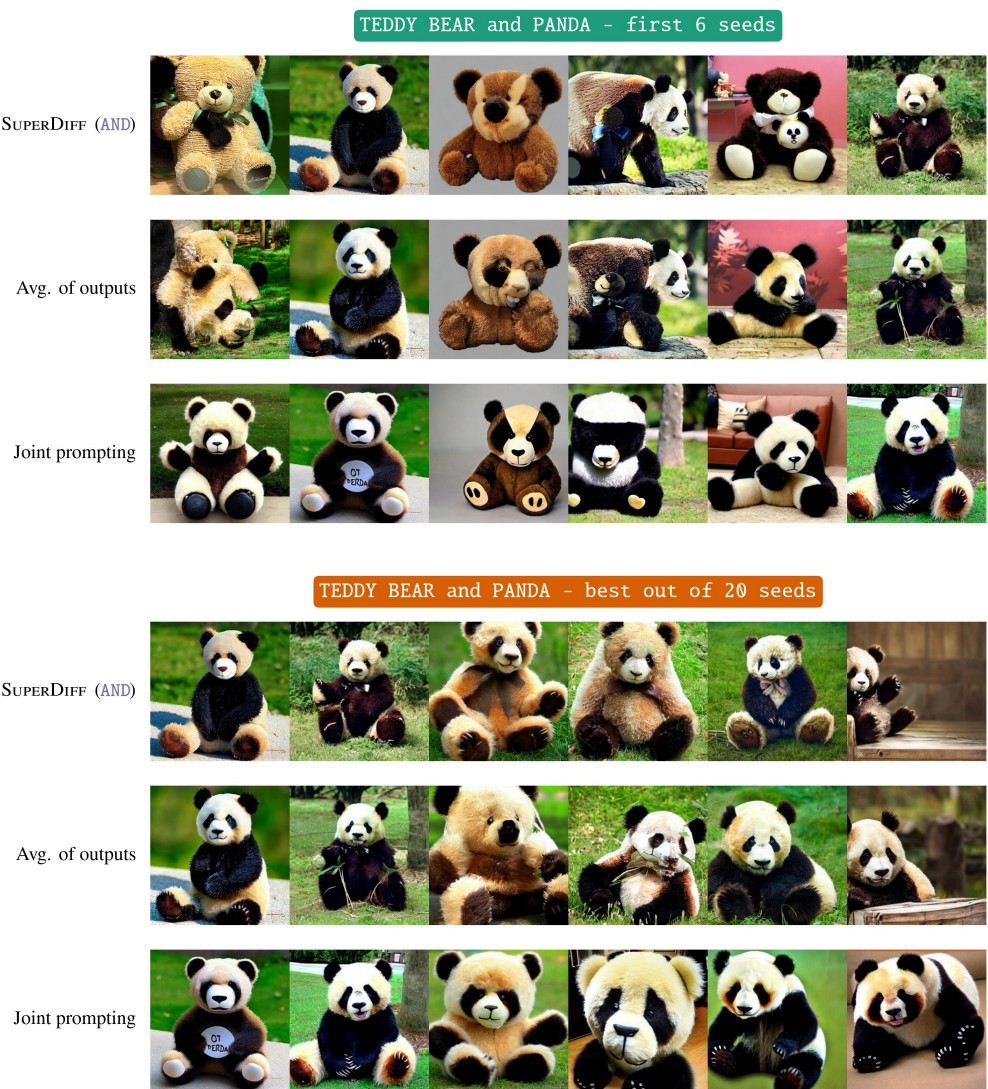

Figure A22: Image compositions generated using SUPERDIFF (AND), averaging of outputs, or joint prompting for the concepts "teddy bear" and "panda".

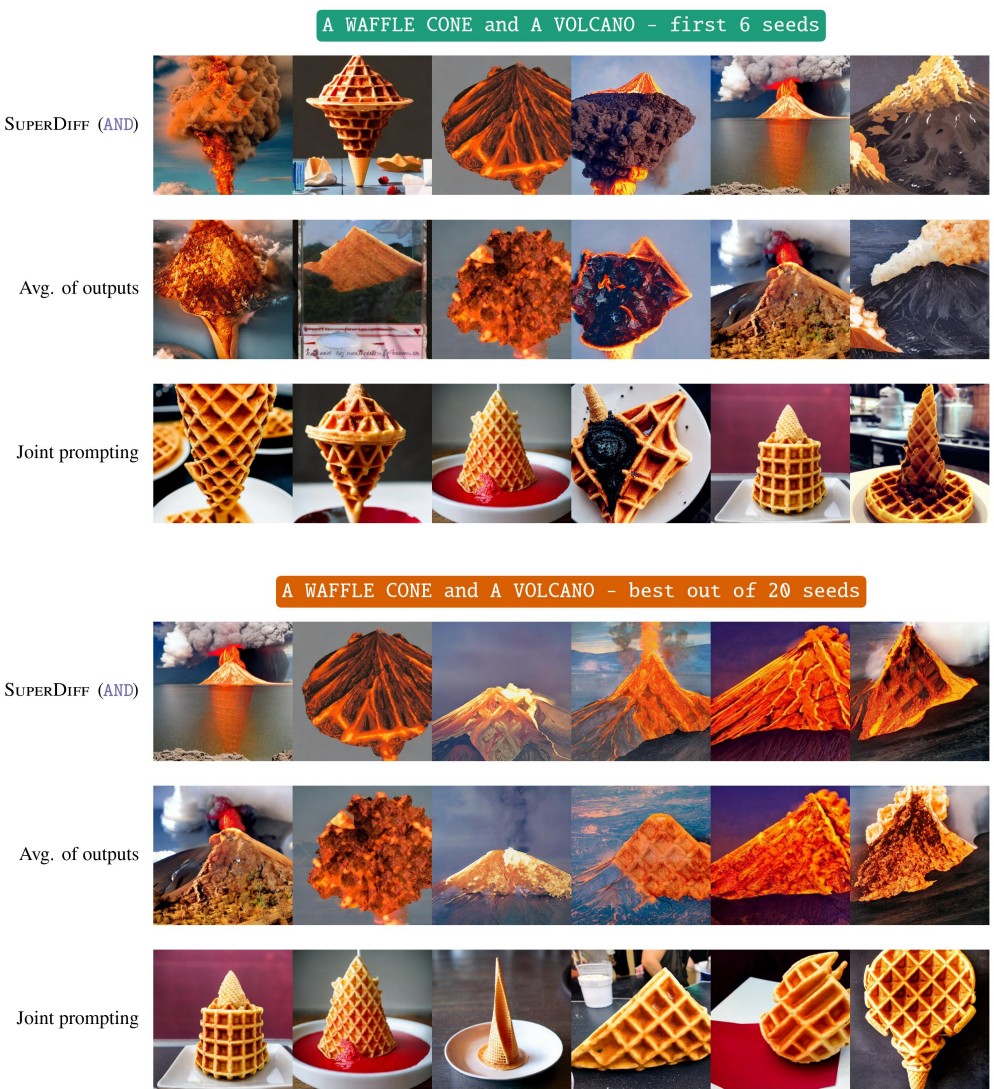

Figure A23: Image compositions generated using SUPERDIFF (AND), averaging of outputs, or joint prompting for the concepts "a waffle cone" and "a volcano".

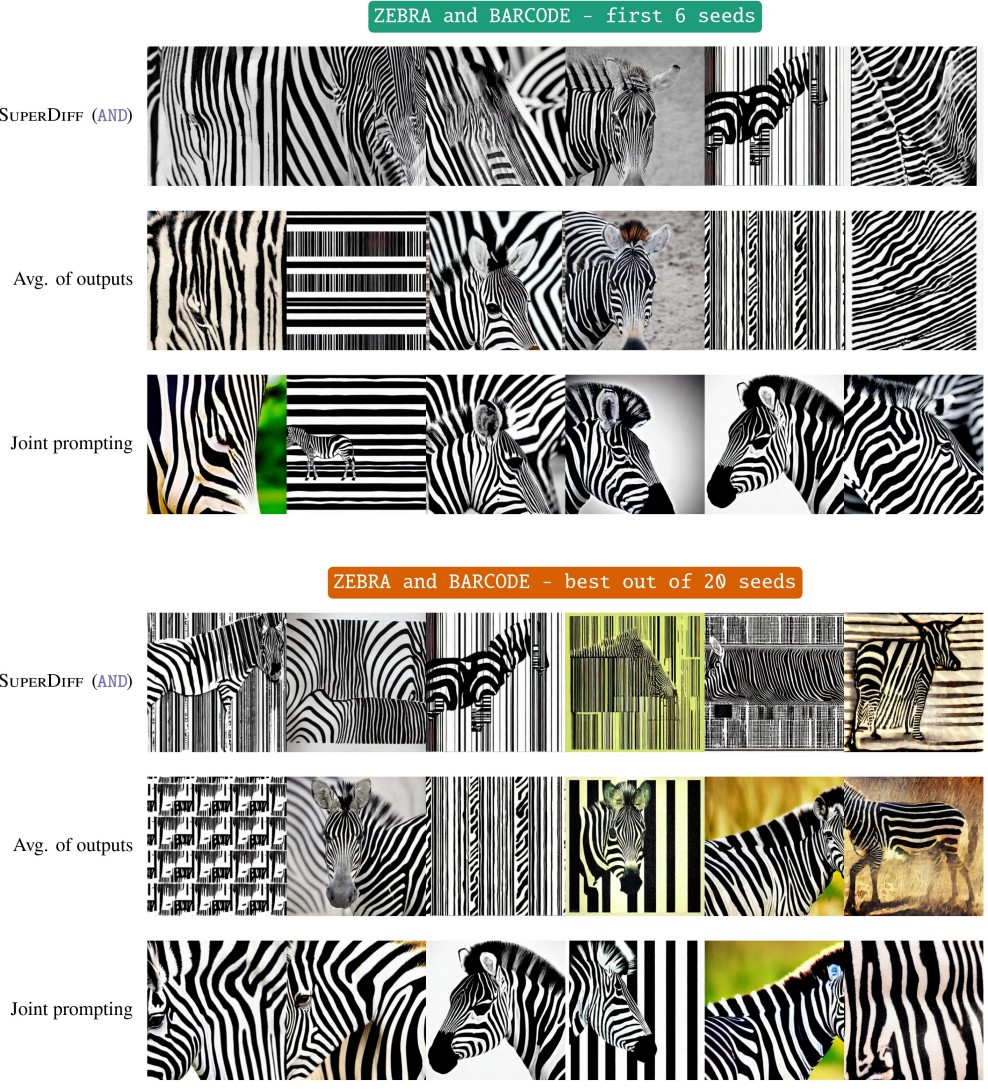

Figure A24: Image compositions generated using SUPERDIFF (AND), averaging of outputs, or joint prompting for the concepts "zebra" and "barcode".

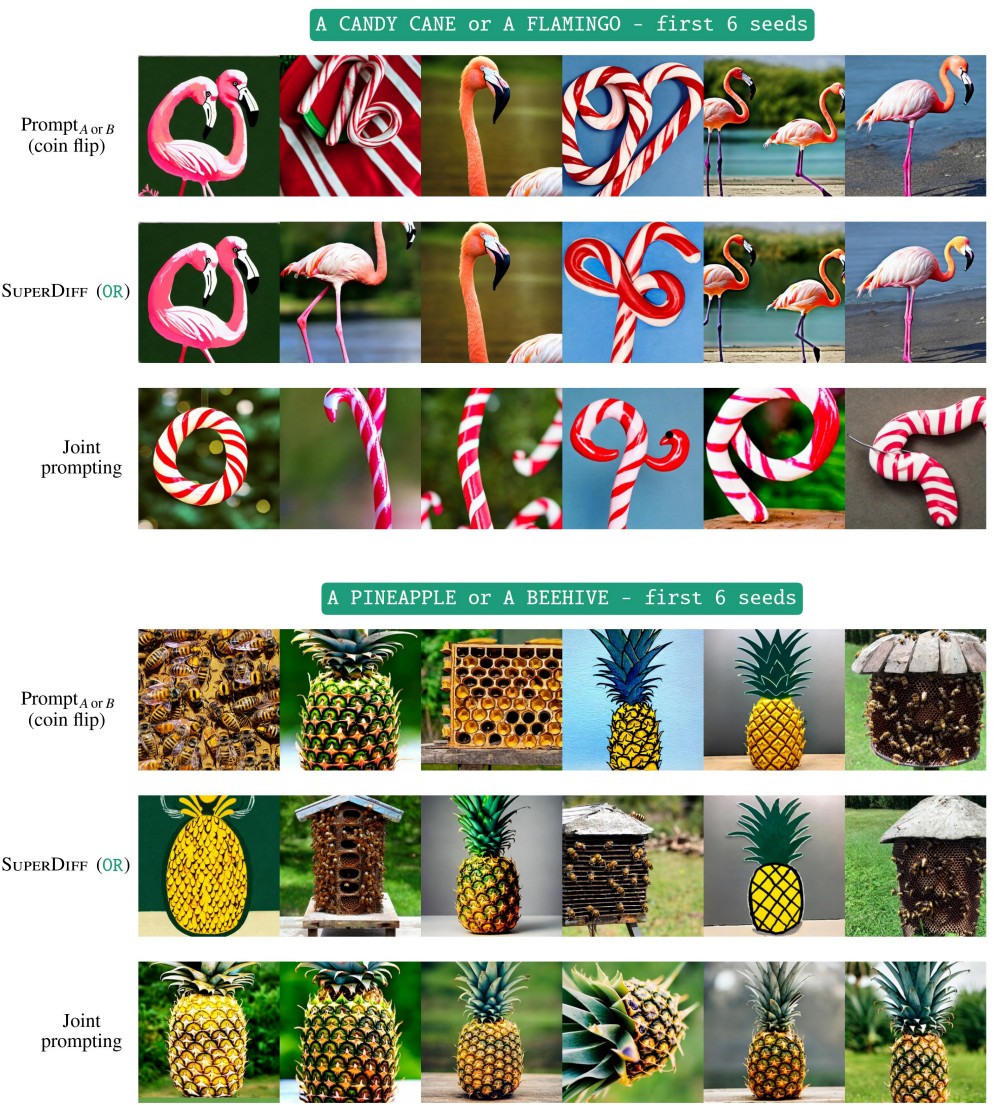

Figure A25: Concept selections using SUPERDIFF (OR), joint prompting, and randomly selecting a prompt of one concept. The top subfigure shows generated images for the concepts "a candy cane" or "a flamingo". The bottom subfigure shows generated images for the concepts "a pineapple" or "a beehive".

