# OpenReview forum: "The Superposition of Diffusion Models Using the Itô Density Estimator"
_ICLR.cc/2025/Conference — ICLR 2025 Spotlight_

### Official Review · Reviewer_1WGb · 2024-11-03

**Soundness:** 3
**Presentation:** 2
**Contribution:** 4
**Rating:** 10
**Confidence:** 5

**Summary:**

This paper introduces a novel, principled, and efficient way to combine diffusion models trained on different datasets (or conditioned on different prompts) to generate images from the mixture and the "intersection" of the corresponding distributions. It is based on a clever way to evaluate the densities $\log p^i_t(x_t)$ of the current iterate $x_t$ under each (noisy) distribution $q^i_t$ during synthesis.

**Strengths:**

The main strength of the paper is the important observation that the probability density function of generated images can be efficiently evaluated without the need for computing the divergence of the score. It is leveraged to sample from mixtures of densities, where the weights can be defined implicitly and adaptively (in the case of the logical AND operator as defined here). The experimental results convincingly demonstrate the effectiveness of the resulting approach.

**Weaknesses:**

In my opinion, the main weakness of the paper is in the clarity of the presentation of the central theoretical result (culminating in Proposition 7) and the motivation for the approach. I believe it can be significantly improved, which could enhance the impact of the paper.
- I found section 2.1 to be unnecessary complicated and rather irrelevant for the rest of the exposition. To my understanding, the main ideas are (1) that SDEs define linear equations on the densities, so that a mixture of clean distributions $\sum w_i p^i$ leads to a mixture of noisy distributions $\sum w_i p_t^i$ and (2) the relationship $\nabla \log (\sum w_i p^i_t) = \sum w_i p^i_t \nabla \log p^i_t / \sum w_i p^i_t$. These motivate the need for evaluating $p^i_t$ to combine scores in the correct way to sample from mixtures.
- The equations are obscured by the use of general schedules with arbitrary $\alpha_t$ and $\sigma^2_t$. I encourage the authors to state the results in the main text with e.g. $\alpha_t = 1$ and $\sigma_2^t$ (known as the variance exploding SDE) to simplify the exposition and relegate the general case to the appendix.
- Some results are also less intuitive (in my opinion) due to the choice to work in discrete time. For example, Proposition 6 and Theorem 1 are nothing but approximating the kernels $k_{\Delta t}$ and $r_{\Delta t}$ with Euler-Maruyama discretizations of the corresponding forward or backward SDEs (and analyzing the discretization error in Theorem 2). Similarly, Proposition 7 can be obtained in continuous time first (and then discretized) by applying Itô's formula to $\log q_t(x_t)$ where $x_t$ is a solution of the backward SDE (and using the fact that $q_t$ solves a Fokker-Planck equation). As an example, in the variance-exploding case, one obtains that $\mathrm{d} \log q_t(x_t) = \frac{\mathrm{d}t}2 ||\nabla \log q_t(x_t)||^2 + \langle \mathrm{d}x_t, \nabla \log q_t(x_t)\rangle$, which is the $\Delta t \to 0$ limit of Proposition 7 with $\alpha_t = 1$ and $\sigma^2_t = t$. I believe this result to be of independent interest, and would thus benefit from being highlighted and stated as simply as possible.

Another issue I have is regarding the logical OR and AND operators as defined in this paper.
- The logical OR operator corresponds to a fixed-weight mixture of distributions, and it is thus trivial to sample from. One can simply select one diffusion model with probability corresponding to the mixture weight, and then use exclusively the score of the chosen diffusion model during generation. Using SuperDiff should be equivalent to this algorithm. So either the improved results in section 4 can also be achieved with this simple baseline, in which case the theoretical results are not needed, or the baseline underperforms, in which case the improvements come from unknown implementation choices which are completely orthogonal from the theoretical analysis. In both cases, this raises questions.
- The real strength of the approach, I think, is when the mixture weights are adaptive (i.e., they are allowed to depend on the current iterate $x_t$). In that case, however, it is not clear what density we are ultimately sampling from. If I understand correctly, here the logical AND operator is defined implicitly, and produces samples $x$ such that $q^1(x) = q^2(x)$. A perhaps more usual definition is that one would aim to sample from the normalized product $q^1(x)q^2(x)/Z$ (or geometric mean $\sqrt{q^1(x)q^2(x)}/Z$), but this seems difficult to achieve with the formalism of this paper. It could be beneficial to include a short discussion of this matter in the paper.

Finally, I could not see where the parameters $\omega$ and $T$ in Table 2 were explained.

**Questions:**

- How do the authors explain the source of their numerical improvements using SuperDiff OR?
- What density is being sampled from when using SuperDiff AND?

---

> ### Author Response · Authors · 2024-11-21
> **Response to Reviewer 1WGb (1/2)**
>
> We thank the reviewer for their positive appraisal of our work in addition to their detailed comments which aided us in enhancing the quality of the submission. We are heartened to hear that the reviewer found that a central contribution of this paper is that the pdf of generated images "can be efficiently evaluated without the need for computing the divergence of the score." Certainly, we think this key differentiating factor of SuperDiff as opposed to other methods and we thank the reviewer for acknowledging that leveraging SuperDiff leads to "results [that] demonstrate convincingly the effectiveness of the resulting approach".
>
> We now turn to address the main points raised by the reviewer grouped by theme. We also wish to highlight that we have updated the main paper PDF, with new content---i.e. experiments, new theory, and streamlined presentation---colored in blue. Finally, we also include a global response to all reviewers that aims to address the shared concerns among reviewers.
>
> > I found section 2.1 to be unnecessary complicated
>
> We simplified the exposition as suggested, we kindly invite the reviewer to view the updated Section 2.
>
> > The equations are obscured by the use of general schedules
>
> We opted to present all the results in terms of the general drift and diffusion coefficient of the forward SDE, which significantly simplified the exposition (please see updated Sections 2 and 3). We discuss the noising schedules only once in the updated Proposition 2 to highlight that the drift term of the forward SDE is simply a linear function.
>
> > Some results are also less intuitive (in my opinion) due to the choice to work in discrete time.
>
> We thank the reviewer for raising this insightful point. As suggested, we have changed the entire exposition to the continuous time and stated the main result in the continuous time using Itô's lemma. We moved the discrete-time derivations into the Appendix D for interested readers.
>
> > The comparison of the logical OR operator and its utility.
>
> Please note, that in the initial submission, we compared SuperDiff(OR) against the models that were trained on separate parts of CIFAR-10 and the model that was on the entire dataset, hence, the differences can be explained by the model capacities. In the updated version, we have added the comparison to randomly choosing the model and then sampling from it, and we see that it has the same performance as SuperDiff(OR), which empirically validates our theory. Although, the OR sampling procedure does not have an immediate application on its own, we believe it can have important down-stream applications, e.g. for the compression algorithms and the likelihood estimation [1] or for the continual learning and unlearning as considered in [3].
>
> > In the AND case, it is not clear what density we are ultimately sampling from.
>
> This is an excellent point. Indeed, usually, sampling from AND is approached as sampling from the product of distributions [2] rather than the set of points with equal densities. However, we note that AND is merely an interpretation of this density and no rigorous connections to Boolean logic exist. In the context of the current paper, sampling from the product of the densities can be approached via the importance sampling and the fact that we can efficiently evaluate the density. However, this would require a separate extensive empirical study, which we leave for the future.
>
> > Finally, I could not see where the parameters $\omega$ and $T$ in Table 2 were explained.
>
> The role of the temperature parameter $T$ is described in Algorithm 1. We renamed $\omega$ in Table 2 (because of the notation clash) to $\ell$ and described its role in equation (18).

---

> ### Author Response · Authors · 2024-11-21
> **Response to Reviewer 1WGb (2/2)**
>
> > How do the authors explain the source of their numerical improvements using SuperDiff OR?
>
> According to our empirical study in Table 1, the temperature parameter $T$ allows for marginal improvements over the OR sampling scheme.
>
> > What density is being sampled from when using SuperDiff AND?
>
> Due to the normalization of all the weights $\sum_i \kappa_i = 1$ that we use in Algorithm 1, we believe that SuperDiff(AND) is close to the standard generation procedure but with an additional projection step of the generated samples to the set of points $\{ x : q^i(x) = q^j(x)\}$. However, we leave the rigorous investigation of this question for an independent study.
>
> ### Closing comments
>
> We thank the reviewer again for their valuable feedback. We hope that our rebuttal addresses their questions and concerns, and that the updated PDF is more streamlined in presentation as requested by the reviewer. We are happy to address any further comments and concerns the reviewer might have otherwise we would be encouraged if the reviewer would continue hold a positive outlook on this work. We thank the reviewer again for their time.
>
> ### References
> [1] Theis, Lucas, Tim Salimans, Matthew D. Hoffman, and Fabian Mentzer. "Lossy compression with gaussian diffusion." arXiv preprint arXiv:2206.08889 (2022).
>
> [2] Du, Yilun, Conor Durkan, Robin Strudel, Joshua B. Tenenbaum, Sander Dieleman, Rob Fergus, Jascha Sohl-Dickstein, Arnaud Doucet, and Will Sussman Grathwohl. "Reduce, reuse, recycle: Compositional generation with energy-based diffusion models and mcmc." In International conference on machine learning, pp. 8489-8510. PMLR, 2023.
>
> [3] Golatkar, Aditya, Alessandro Achille, Ashwin Swaminathan, and Stefano Soatto. "Training data protection with compositional diffusion models." arXiv preprint arXiv:2308.01937 (2023).

---

> > ### Comment · Reviewer_1WGb · 2024-11-21
> >
> > I thank the authors for their very detailed response. I am delighted that the authors found my comments useful and that it led to an improvement of the paper. I think the writing is much more clear now (minor point: Ôti in Theorem 1 should be Îto). I believe this work should be highlighted at the conference and have raised my score accordingly.

---

> > > ### Author Response · Authors · 2024-11-21
> > >
> > > We are excited to see that our work is so well received! By reversing Itô (hence, Ôti), we meant to highlight that Itô's lemma is applied to the time-reverse SDE, but we will keep working on the presentation of this result for the next version of the paper.

---

### Official Review · Reviewer_DUd3 · 2024-11-04

**Soundness:** 3
**Presentation:** 2
**Contribution:** 3
**Rating:** 6
**Confidence:** 3

**Summary:**

This paper proposes a novel algorithm for combining multiple pre-trained diffusion models at inference time, by the principle of superposition of vector fields. The method demonstrates more diverse generation results, better prompt following on image data, and improved structure design of proteins as well.

**Strengths:**

* The theoretical framework is solid.
* The method is well-motivated and supported by the theory.
* The method is training-free, and could be applied to diffusion models with different architectures.
* The results of protein generation outperform other baselines.

**Weaknesses:**

* The practical implications of AND, and OR operators are not explained clearly in both image and protein generation settings. What effect will the OR operator create on images, compared to the AND operator?
* Lacks quantitative results on SD. Could have used metrics such as TIFA Score [1]  and Image Reward [2]. I wonder if there is any reason that no such metric was used.
* Lacks comparison against other relevant methods [3-6]. In particular, [3,4,6] are all inference-time methods that sample from some sort of mixture of scores and demonstrate multiple practical uses, such as composing objects, styles, scenes, or improving text-image alignment. Need more discussions on the capabilities of the proposed method versus others: besides the different theoretical perspectives, how SUPERDIFF performs differently, the strengths and weaknesses of SUPERDIFF than the other methods. If experiments are not possible, please include a more detailed discussion. The comparison could help readers understand the proposed method in a broader context.

[1] Hu, Y., Liu, B., Kasai, J., Wang, Y., Ostendorf, M., Krishna, R., Smith, N.A.: Tifa: Accurate and interpretable text-to-image faithfulness evaluation with question answering. arXiv preprint arXiv:2303.11897 (2023)

[2] Xu, J., Liu, X., Wu, Y., Tong, Y., Li, Q., Ding, M., Tang, J., Dong, Y.: Imagereward: Learning and evaluating human preferences for text-to-image generation. Advances in Neural Information Processing Systems 36 (2024)

[3] Du, Y., Durkan, C., Strudel, R., Tenenbaum, J.B., Dieleman, S., Fergus, R., SohlDickstein, J., Doucet, A., Grathwohl, W.S.: Reduce, reuse, recycle: Compositional generation with energy-based diffusion models and mcmc. In: International conference on machine learning. pp. 8489–8510. PMLR (2023)

[4] Golatkar, A., Achille, A., Swaminathan, A., Soatto, S.: Training data protection with compositional diffusion models. arXiv preprint arXiv:2308.01937 (2023)

[5] Biggs, Benjamin, et al. "Diffusion Soup: Model Merging for Text-to-Image Diffusion Models." arXiv preprint arXiv:2406.08431 (2024).

[6] Liu, Nan, et al. "Compositional visual generation with composable diffusion models." European Conference on Computer Vision. Cham: Springer Nature Switzerland, 2022.

**Questions:**

* Why are there no quantitative results on SD, and detailed discussion of other very relevant methods as referenced earlier?
* FID statistics on CIFAR-10 are computed on the whole dataset. Is it fair to evaluate models trained on a partial dataset using such statistics, especially when the two partitions are generated by splitting the classes?
* What are the practical implications of the OR operator, especially in the field of image generation?

---

> ### Author Response · Authors · 2024-11-21
> **Response to Reviewer DUd3 (1/2)**
>
> We thank the reviewer for their time and detailed feedback. We appreciate the fact that the reviewer views our superdiffusion framework to be "well-motivated and supported by theory", and that the overall framework is "solid". We are also heartened to hear that the reviewer agrees that SuperDiff can be applied across a wide variety of diffusion models as it enjoys being "training-free". We also thank the reviewer for acknowledging that our protein generation results using SuperDiff "outperform other baselines".
>
> We focus here on addressing the key clarification points raised in this review while noting that we have updated the revision PDF with new experiments, theory, while increasing the clarity of exposition. For increased transparency, these changes are highlighted in blue. Finally, we also highlight that we have summarized these changes in our global response to all reviewers addressing shared concerns.
>
> > What effect will the OR operator create on images, compared to the AND operator?
>
> The OR operator corresponds to sampling from the mixture of densities of corresponding models, which we explain in Section 2.2. For instance, as we visually demonstrate in Fig. A21 of the updated manuscript, using SuperDiff(OR) one can sample from $p(\text{image}) = 0.5\cdot p(\text{image}|\text{flamingo}) + 0.5\cdot p(\text{image}|\text{candy cane})$, which will produce either an image of 'flamingo' or an image of 'candy cane'.
>
> > Lacks quantitative results on SD. Could have used metrics such as TIFA Score [1] and Image Reward [2].
>
> Thank you for suggesting additional quantitative metrics; we have included CLIP, TIFA, and ImageReward evaluations, which strengthen the empirical caliber of our paper. We have now included each additional metric in Table 2, as well as increased the number of total tasks. Our updated findings show that SuperDiff outperforms all baselines across all metrics in both OR and AND settings, indicating that our model is able to consistently and faithfully interpolate concepts (AND) and select concepts (OR).
>
> > Lacks comparison against other relevant methods [3-6]. In particular, [3,4,6]
>
> Thank you for suggesting this relevant literature; we added the corresponding discussion and the references into the related work section. First, we would like to note that we already compare against [6] which we called "averaging of outputs" in the original submission; we have made it more clear that this method is from [6] in the updated manuscript. However, unfortunately, comparing to all of the proposed works is not possible. In particular, the algorithm proposed in [3] requires an access to the density of samples, which is not accessible for Stable Diffusion (SD); in fact, efficient estimation of the density during the generation is our major contribution. Analogously, [4] proposes to train a separate neural network that estimates the densities of samples; the comparison here is complicated by the absence of an open-sourced version of their model. Finally, [5] proposes a method to "merge information between different models trained on different subsets of data into a single model" by averaging their weights. This is not possible in the considered settings (and actually in most practical scenarios) because: (i) for the proteins, the models have different architectures; hence, it is not clear how to average these models in weight-space, and (ii) we use a single pre-trained checkpoint of SD with different captions, and it is not clear how to apply [5] for the same model with different captions. We have added these discussions into our related works section.
>
> > Why are there no quantitative results on SD, and detailed discussion of other very relevant methods as referenced earlier?
>
> As per your suggestion, we have added quantitative results for the experiments with Stable Diffusion (see Table 2 of the updated manuscript) and the detailed discussion of the relevant methods.

---

> ### Author Response · Authors · 2024-11-21
> **Response to Reviewer DUd3 (2/2)**
>
> > FID statistics on CIFAR-10 are computed on the whole dataset. Is it fair to evaluate models trained on a partial dataset using such statistics, especially when the two partitions are generated by splitting the classes?
>
> The purpose of this experiment is to verify the feasibility of the OR sampling procedure. Namely, we want to make sure that using SuperDiff(OR), we can efficiently join two models trained on different modalities. Thus, we evaluate models trained on partial data to demonstrate the importance of different modalities. Indeed, otherwise, one could imagine the dataset where removing one of the modalities wouldn't lead to significant degradation of the model. Also, we updated our baselines in Table 1 (see the updated manuscript).
>
> > What are the practical implications of the OR operator, especially in the field of image generation?
>
> In the current paper, we study the feasibility of the OR operator for using it in practice, which complement our theoretical findings. Although, the OR sampling procedure does not have an immediate application on its own, we believe it can have important down-stream applications, e.g. for compression algorithms and likelihood estimation [1] or for continual learning and unlearning as considered in [4].
>
> ### Closing comment
>
> We hope that our responses were sufficient in clarifying all the great questions asked by the reviewer. We thank the reviewer again for their time, and we politely encourage the reviewer to consider updating their score if they deem that our responses in this rebuttal along with the new experiments merit it.
>
> ### References
>
> [1] Theis, Lucas, Tim Salimans, Matthew D. Hoffman, and Fabian Mentzer. "Lossy compression with gaussian diffusion." arXiv preprint arXiv:2206.08889 (2022).
>
> [3] Du, Y., Durkan, C., Strudel, R., Tenenbaum, J.B., Dieleman, S., Fergus, R., SohlDickstein, J., Doucet, A., Grathwohl, W.S.: Reduce, reuse, recycle: Compositional generation with energy-based diffusion models and mcmc. In: International conference on machine learning. pp. 8489–8510. PMLR (2023)
>
> [4] Golatkar, A., Achille, A., Swaminathan, A., Soatto, S.: Training data protection with compositional diffusion models. arXiv preprint arXiv:2308.01937 (2023)
>
> [5] Biggs, Benjamin, et al. "Diffusion Soup: Model Merging for Text-to-Image Diffusion Models." arXiv preprint arXiv:2406.08431 (2024).
>
> [6] Liu, Nan, et al. "Compositional visual generation with composable diffusion models." European Conference on Computer Vision. Cham: Springer Nature Switzerland, 2022.

---

> > ### Author Response · Authors · 2024-11-24
> >
> > Dear Reviewer,
> >
> > We are very grateful for the time you have spent providing constructive and valuable feedback on our paper. As the the rebuttal period is quickly approaching the end, we would like to have the opportunity to answer any remaining questions or doubts that you may still have. We would like to note that in our rebuttal, we have followed your great suggestions and included quantitative evaluations using three image metrics. We also improved our related works section in the context of the references provided by the reviewer and clarified the implications of the operators we studied.
> >
> > We would be happy to engage in further discussions regarding these updates or clarify any remaining doubts that the reviewer still has, please let us know! We thank the reviewer again for their time spent evaluating our paper so far. If the reviewer finds our new experiments and discussions useful for improving the paper, we would appreciate it if they could potentially consider reassessing their evaluation of our paper.

---

> > > ### Comment · Reviewer_DUd3 · 2024-11-25
> > >
> > > Thanks for your comments, and my main concerns are addressed. I will raise my score.

---

> > > > ### Author Response · Authors · 2024-11-26
> > > >
> > > > We thank the reviewer for acknowledging that our new experiments and discussions were satisfactory and enabled them to more positively endorse our paper. We are also happy to answer any more remaining questions the reviewer might have and we are very appreciative and grateful for the reviewer for their time and effort during this rebuttal period.

---

### Official Review · Reviewer_h88w · 2024-11-10

**Soundness:** 2
**Presentation:** 4
**Contribution:** 3
**Rating:** 6
**Confidence:** 4

**Summary:**

The authors propose the method to combine the multiple pre-trained diffusion models at the inference time without retraining the models. They come up with the theoretical principles using the continuity equation to show how diffusion models can be viewed in the superposition of elementary vector fields. Here, they implement two algorithms to combine pre-trained diffusion models. One is a mixture of densities (sampling from one model OR another), and the other is equal densities(samples that are likely to belong to one model AND another). They also overcome the challenges with existing diffusion models, such as (1. Differences of Marginal super-positional vector field between different models) and (2. Divergence operation’s time complexity) by introducing their density estimator. They apply their approach in various ways, such as combining models trained on disjoint datasets, concept interpolation in image generation, and improving the structure of protein design.

**Strengths:**

1. The paper is well-written and easy to understand. There are almost no grammatical errors. By developing the idea of superposition and using theoretical principles, the authors prove the idea's potential and present a reasonable result.
2. They apply their work to two individual tasks, which could be divisive among readers, but I found it interesting.
3. Also, it is interesting that the authors discover their model follows traditional operators such as logical OR and logical AND, making it intuitive. Similarly, the background of explaining how the superposition emerged from the diffusion models by using the vector fields and propositions is interesting.
4. They use nine propositions, two theorems, and one lemma to support their idea, which helps readers understand why their algorithms work.

**Weaknesses:**

**(Main) Qualitative Results and the Quantitative Results with figures**
1. Figure 1 is weak to verify the novelty of the model. I also think the generated images in the appendix, as well as the qualitative results, are mediocre.
2. The author only uses the AND operation (sampling equal densities) for qualitative results, and OR operation for the quantitative results. I believe that including the results for the OR operation in qualitative results and the AND operation in quantitative results would strengthen the paper. This would provide a more comprehensive view of the statement on line 104 on page 2: "improvements in designability and novelty generation".
3. Figure 2 does not show how the generated images are actually arranged. It is necessary to verify if the same results occur when directly arranging the generated images with the trained datasets.

**Evaluation metrics and ablation study**
1. The comparative group for the paper's qualitative results is insufficient. Comparisons with other recent models that produce dual images, such as factorization diffusion or visual anagram (Geng et al.,2024), could be added. Since it is clear that the latent diffusion result for just adding the prompt 'that looks like' would indeed be worse than the proposed method.
2. Similarly, in the process of making baselines for concept interpolation, I wonder if the value of the ablation study would have increased if the direction of A->B and B->A was changed and the comparison group was chosen by using the better result.
3. The execution times for the experiments were not provided. The authors claim to have solved the computational expense issue, but no results support this claim.

**Clarity of the paper**
1. Proposition 8 appears to be quite important but confusing because it was cut off the page. Listing the individual terms of $Aκ = b + o(Δt)$ on the same page would improve comprehension.
2. The related work section comes out almost at the end of page 10, and I think this part should come out more front. It comes out so out of the blue that it somewhat interferes with understanding.
3. The protein generation part is not clearly introduced. The authors compare Designability, Novelty, and Diversity, and there is no separate explanation of how this part is meaningful in protein generation. I didn't feel that logic was connected smoothly.

**Questions:**

**Major Questions**
1. I am curious why text-based evaluation metrics such as Clip Score were not used. It seems like an obvious choice to do.
2. In section 2.1, how were the mixing coefficients $wj$ actually set? Is the model capable of adjusting the weights for mixing? I am also curious about how $N$ for the individual forward process was actually set.
3. The method overview on page 5 mentions that pre-trained diffusion models can be used, but I am curious if the only one actually used is CIFAR-10, as shown in Table 1. (The experiment by providing the models with CIFAR-10 with two sets of labels divided into five and five) I think if the authors provide the results using the output of various datasets, the paper will be stronger.

**Minor Questions**
1. I think there should be punctuation after *"...a superposition of elementary vector fields"* on page 3, lines 140 and 141.
2. I think the introduction of the abstract is too long. This could be reduced since the intro occupies 1/3 of the entire amount.
3. It would have been interesting if there was a comparison according to the distance of the disjoint set.

---

> ### Author Response · Authors · 2024-11-21
> **Response to Reviewer h88w (1/4)**
>
> We would like to thank the reviewer for their time and effort spent reviewing our work. We are pleased to hear that the reviewer finds that our paper was "well-written and easy to understand", and that our framework of superposition is "interesting" in how it leads to traditional compostion operators like logical OR and logical AND. We now address the key points raised in the review, while we note that additional experiments and updates to the main paper are provided in the revision PDF (colored in blue text) and the main changes addressing common concerns are listed in the common response.
>
>
> ### Presentation Concerns
>
> We appreciate reviewer's nuanced feedback on the presentation of the paper.
>
> > Figure 1 is weak to verify the novelty of the model. I also think the generated images in the appendix, as well as the qualitative results, are mediocre.
>
> As per your suggestion, we have added quantitative results for the experiments with Stable Diffusion (see Table 2 of the updated manuscript). We have also included several more generated images in Appendix H from 14 additional concept pairs; we hope the reviewer finds these interesting.
>
> >The author only uses the AND operation (sampling equal densities) for qualitative results, and OR operation for the quantitative results. I believe that including the results for the OR operation in qualitative results and the AND operation in quantitative results would strengthen the paper.
>
> We thank the reviewer for this great suggestion. We have included both quantitative and qualitative results for OR and AND images generated by Stable Diffusion (see Table 2 and Figure A21). For the quantitative evaluation, we look at three metrics: CLIP Score [1], ImageReward [2], and TIFA [3]. We find that our method outperforms baselines (averaging of scores (based on [4]) and joint prompting) across all metrics. We also include quantitative results for proteins generated using AND (see Table 3). We find that our method can design almost double the number of novel proteins compared to the next best method, which motivates the use case of our method in discovery settings.
>
> > Figure 2 does not show how the generated images are actually arranged. It is necessary to verify if the same results occur when directly arranging the generated images with the trained datasets.
>
> Figure 2 demonstrates SuperDiff(OR) and SuperDiff(AND) on a mixture of 2D Gaussians to illustrate the intuition behind the proposed sampling procedures. The images generated using SuperDiff(AND) for the Stable Diffusion model are presented in Figure 1 (as well as Appendix H), and we compare them quantitatively in Table 2. In case that is not what you mean by "directly arranging the generated images with the trained datasets" we would be happy to provide further clarifications upon request.
>
> ### References
>
> [1] Hessel, Jack, Ari Holtzman, Maxwell Forbes, Ronan Le Bras, and Yejin Choi. "Clipscore: A reference-free evaluation metric for image captioning." arXiv preprint arXiv:2104.08718 (2021).
>
> [2] Xu, J., Liu, X., Wu, Y., Tong, Y., Li, Q., Ding, M., Tang, J., Dong, Y.: Imagereward: Learning and evaluating human preferences for text-to-image generation. Advances in Neural Information Processing Systems 36 (2024)
>
> [3] Hu, Y., Liu, B., Kasai, J., Wang, Y., Ostendorf, M., Krishna, R., Smith, N.A.: Tifa: Accurate and interpretable text-to-image faithfulness evaluation with question answering. arXiv preprint arXiv:2303.11897 (2023)
>
> [4] Liu, Nan, et al. "Compositional visual generation with composable diffusion models." European Conference on Computer Vision. Cham: Springer Nature Switzerland, 2022.

---

> ### Author Response · Authors · 2024-11-21
> **Response to Reviewer h88w (2/4)**
>
> ### Evaluation Concerns
>
> We thank the reviewer for suggesting ways to improve how we evaluate our method. In this update, we include several quantitative evaluations of our method.
>
> > The comparative group for the paper's qualitative results is insufficient. Comparisons with other recent models that produce dual images, such as factorization diffusion or visual anagram (Geng et al.,2024), could be added. Since it is clear that the latent diffusion result for just adding the prompt 'that looks like' would indeed be worse than the proposed method.
>
> Besides joint prompting. we compare to the averaging of scores proposed in [1]. We extended the comparison for images and proteins (see updated Section 4). The method proposed in [2] is principally different from what we consider, as it is based on the hand-crafted decomposition of images into different representations (e.g. low- and high-pass filters, color, and scale), which define the domain of the generated illusions (e.g. different prompts are visible on different scales). Our paper does not have a goal to produce optical illusions but rather to control the outputs based on the density to generate samples that are impossible to generate otherwise. This allows us to apply the proposed algorithm on completely different domains such as proteins where the application of [2] is impossible.
>
> >Similarly, in the process of making baselines for concept interpolation, I wonder if the value of the ablation study would have increased if the direction of A->B and B->A was changed and the comparison group was chosen by using the better result.
>
> This is a great point and we have taken this into account in our update. In our quantitative evaluation of SD-generated images, we generate images in both directions (A->B and B->A) and keep the direction that results in the higher score for each metric. We also show the better direction in Appendix H.
>
> > The execution times for the experiments were not provided. The authors claim to have solved the computational expense issue, but no results support this claim.
>
> Estimating the divergence of the vector field parameterized as a deep neural network is notoriously expensive. When generating a single image, our density estimator does not introduce _any_ overhead and generation takes 80.2 ± 0.5 seconds on average, while generation with estimating divergence takes 300 ± 4 seconds and requires 1.5x more memory (averages were computed for 3 runs, 100 steps each). This is expected, as estimating the divergence requires computing the Jacobian-vector product, which is significantly more expensive than evaluating the function, even using Autodiff [3].
>
> ### References
> [1] Liu, Nan, et al. "Compositional visual generation with composable diffusion models." European Conference on Computer Vision. Cham: Springer Nature Switzerland, 2022.
>
> [2] Geng, Daniel, Inbum Park, and Andrew Owens. "Factorized diffusion: Perceptual illusions by noise decomposition." In European Conference on Computer Vision, pp. 366-384. Springer, Cham, 2025.
>
> [3] Google. "Autodiff Cookbook: Jacobians and Hessians using jacfwd and jacrev." JAX Documentation, Google, https://jax.readthedocs.io/en/latest/notebooks/autodiff_cookbook.html#jacobians-and-hessians-using-jacfwd-and-jacrev. Accessed 20 Nov. 2024.

---

> ### Author Response · Authors · 2024-11-21
> **Response to Reviewer h88w (3/4)**
>
> ### Clarity concerns
>
> > Proposition 8 appears to be quite important but confusing because it was cut off the page. Listing the individual terms of on the same page would improve comprehension.
>
> We completely agree with the reviewer and have updated the manuscript to ensure all propositions, theorems etc. are contained within a single page.
>
> > The related work section comes out almost at the end of page 10, and I think this part should come out more front. It comes out so out of the blue that it somewhat interferes with understanding.
>
> We updated the related work section discussing the possible applications of different composition methods, mapping the existing literature, and comparing the proposed approach to it. This more practical discussion, creates a streamlined transition between the empirical study and the conclusion sections.
>
> > The protein generation part is not clearly introduced. The authors compare Designability, Novelty, and Diversity, and there is no separate explanation of how this part is meaningful in protein generation. I didn't feel that logic was connected smoothly.
>
> We acknowledge the reviewer's comments regarding the potential lack of clarity regarding the metrics used for our protein generation experiments. We have updated the clarity of this section by describing how each metric evaluates different facets of unconditional protein structure generation which are quite standard in the literature and employed by several seminal works [1-3]. At an intuitive level, similar to common ML metrics, designability targets how close the generated structure from our approach mimics the generated structure of a computational folding model (e.g. AlphaFold2/ESMFold). We care about this because such a metric (designability) is known to be highly correlated with actual wet lab synthesizability---a key goal of rational design. Of the generated protein structures that are designable we additionally care about diversity, which measures the number of clusters using an off the shelf protein clustering algorithm, and novelty which roughly measures the closest generated structure to a training set sample. These metrics have a similar interpretation to precision, recall, and FID, which are standard metrics for generative modeling. Finally, we note that Appendix G.2 already contained a detailed description of these metrics and their computation. We hope our answer here and updates to the manuscript fully address the very reasonable concern raised by the reviewer.
>
> ### References
>
> [1] Watson, Joseph L., et al. "De novo design of protein structure and function with RFdiffusion." Nature 620.7976 (2023): 1089-1100.
>
> [2] Yim, Jason, et al. "SE (3) diffusion model with application to protein backbone generation." International Conference on Machine Learning (2023).
>
> [3] Bose, Avishek Joey, et al. "Se (3)-stochastic flow matching for protein backbone generation." The International Conference on Learning Representations (ICLR) (2023).

---

> ### Author Response · Authors · 2024-11-21
> **Response to Reviewer h88w (4/4)**
>
> ### Major questions
>
> >I am curious why text-based evaluation metrics such as Clip Score were not used. It seems like an obvious choice to do.
>
> Thank you for this extremely valuable suggestion -- in this update, we report three quantitative metrics for Stable Diffusion-generated images vs baselines: Clip Score [1], ImageReward [2], and TIFA [3]. We find that our method outperforms the baselines across all metrics.
>
> > In section 2.1, how were the mixing coefficients $w_j$ actually set? Is the model capable of adjusting the weights for mixing? I am also curious about how $N$ for the individual forward process was actually set.
>
> The mixture coefficients $w_j$ are assumed to be dictated by the application. For instance, if we consider mixture of two equal densities (as we do for SuperDiff(OR) for CIFAR-10 and Stable Diffusion) then $w_1 = w_2 = 0.5$ and $N=2$. The final weights of the vector are indeed adjusted automatically according to Propositions 3, 4 or 6 in the updated PDF (correspondingly, Propositions 1 and 8 in the original submission). Finally, note that $N$ can be the dataset size when the superposition principle is applied to the training of the entire model but we moved this discussion to Appendix B to clarify the exposition in the main body.
>
> > The method overview on page 5 mentions that pre-trained diffusion models can be used, but I am curious if the only one actually used is CIFAR-10, as shown in Table 1. (The experiment by providing the models with CIFAR-10 with two sets of labels divided into five and five) I think if the authors provide the results using the output of various datasets, the paper will be stronger.
>
> All the experiments in the paper use pre-trained diffusion models without any fine-tuning or re-training. Besides the experiments on CIFAR-10 (different models trained on different datasets), this includes experiments with Stable Diffusion (same model but different conditioning) and protein generative models (different models trained on different datasets). Note that we updated our empirical studies for all three experiments incorporating the suggestions of the reviewers (see Section 4 of the updated version).
>
> ### Minor questions
> > I think there should be punctuation after "…a superposition of elementary vector fields" on page 3, lines 140 and 141.
>
> We re-wrote this part, thank you.
>
> > I think the introduction of the abstract is too long. This could be reduced since the intro occupies 1/3 of the entire amount.
>
> We appreciate the reviewer's suggestion. We have now condensed the abstract to enhance clarity and compactness.
>
> > It would have been interesting if there was a comparison according to the distance of the disjoint set.
>
> As the reviewer suggested, we conducted additional experiments with CIFAR-10 where instead of taking the disjoint set of labels, we take a random split of CIFAR-10 (still disjoint but having all labels). In Appendix F, we report Table A1, which is analogous to Table 1 in the body for disjoint labels.
>
> ### Closing comment
>
> We thank the reviewer again for their valuable feedback. We hope that our rebuttal addresses their questions and concerns, and we kindly ask the reviewer to consider a fresher evaluation of our paper if the reviewer is satisfied with our responses. We are also more than happy to answer any further questions that arise.
>
> ### References
> [1] Hessel, Jack, Ari Holtzman, Maxwell Forbes, Ronan Le Bras, and Yejin Choi. "Clipscore: A reference-free evaluation metric for image captioning." arXiv preprint arXiv:2104.08718 (2021).
>
> [2] Xu, J., Liu, X., Wu, Y., Tong, Y., Li, Q., Ding, M., Tang, J., Dong, Y.: Imagereward: Learning and evaluating human preferences for text-to-image generation. Advances in Neural Information Processing Systems 36 (2024)
>
> [3] Hu, Y., Liu, B., Kasai, J., Wang, Y., Ostendorf, M., Krishna, R., Smith, N.A.: Tifa: Accurate and interpretable text-to-image faithfulness evaluation with question answering. arXiv preprint arXiv:2303.11897 (2023)

---

> > ### Comment · Reviewer_h88w · 2024-11-22
> >
> > Thank you for a detailed response to my review with improvements. I think this paper is good, but there are still many things to develop. I will maintain my score. However, I’m happy to raise my confidence level since I have already given a positive score. Good luck.

---

> > > ### Author Response · Authors · 2024-11-22
> > >
> > > We thank the reviewer for their time and effort in reviewing our rebuttal. We are glad to see our updates to the paper, and rebuttal allows the reviewer to be more confident in their positive assessment of the paper. We are more than happy to answer any lingering concerns or questions the reviewer might still hold during the remainder of this rebuttal period. Please do let us know!

---

### Author Response · Authors · 2024-11-21
**Response to Everyone**

We thank all the reviewers for their time and constructive comments. We are ecstatic that the reviewers found our paper to make an important contribution regarding efficient on-the-fly density estimation (R 1WGb) and that our method of superimposing diffusion models is novel, principled, and theoretically well-motivated (R 1WGb, DUd3, h88w). We are also very glad to hear that the reviewers enjoyed the breadth of experimental domains we investigated (R DUd3), found the experimental results to convincingly demonstrate the effectiveness our approach (R 1WGb), as well as valued the fact that our method is training-free and can easily be applied to different architectures (R DUd3). We now address shared concerns raised by the reviewers, and summarise our new experiments and ablations included in the supplementary PDF. All changes are higlighted in the attached PDF in blue.


### Summary of changes
- As suggested by Reviewer 1WGb, we opted to present all the results in terms of the general drift and diffusion coefficient of the forward SDE, which significantly simplified the exposition (see updated Sections 2 and 3).
- As suggested by Reviewer 1WGb, we have changed the entire exposition to continuous time and stated the main result in continuous time using Itô's lemma. We moved the discrete time derivations into the Appendix D for interested readers.
- As suggested by Reviewers h88w and DUd3, we have added extensive comparisons of the proposed algorithm for Stable Diffusion. Namely, we have added quantitative comparisons for SuperDiff(AND) and SuperDiff(OR) using CLIP [1], ImageReward [2], and TIFA [3], where we demonstrate the practicality of the proposed scheme (see Table 2). We also extended the qualitative comparison for SuperDiff(AND) and provided qualitative comparison for SuperDiff(OR) (see Appendix H).
- As suggested by Reviewer h88w, we have included a quantitative evaluation for SuperDiff(AND) for designing proteins. Excitedly, we report that SuperDiff(AND) is able to generate almost 2x more novel, designable proteins than the next best model, motivating the utility of our method in discovery settings.
- As suggested by Reviewers 1WGb and h88w, we extended our empirical study for CIFAR-10. Namely, we added the comparison to the random choice of models A and B (see Table 1), and provided comparison for another split where both parts A and B contain all the labels (see Appendix F, Table A1).

---

> ### Author Response · Authors · 2024-11-21
> **Increasing quantitative experiments (R h88w, DUd3)**
>
> Reviewer comments:
>
> >(h88w) I believe that including the results for the OR operation in qualitative results and the AND operation in quantitative results would strengthen the paper.
>
> >(DUd3) Lacks quantitative results on SD. Could have used metrics such as TIFA Score and Image Reward.
>
> We agree with reviewers' suggestions about including quantative experiments on images generated by Stable-Diffusion (SD) and are appreciative to Reviewers DUd3 and h88w for suggesting three metrics to use: CLIP [1], ImageReward [2], and TIFA [3]. We increased the number of generated images in our experiment 3-fold and scored them using these three metrics. We found that our method outperfomed baselines on all metrics; notably, our method had a TIFA score of 39.92, while the next best method ([4]) had a score of 32.48.
>
> We also increased the breadth of our evaluation: we showcase images from the OR setting and evaluate them quantitively, also outperforming baseline methods. Finally, we extend our protein evaluation to the AND setting and find that our method is able to generate the highest number of novel proteins, which is really motivating for a discovery setting.

---

> ### Author Response · Authors · 2024-11-21
> **Closing general response comment and references**
>
> We have addressed individual questions and concerns in our reviewer responses. We would like to thank all reviewers for their valuable time and effort in reviewing our manuscript; their suggestions were instrumental in helping us improve our work!
>
> ### References
>
> [1] Hessel, Jack, Ari Holtzman, Maxwell Forbes, Ronan Le Bras, and Yejin Choi. "Clipscore: A reference-free evaluation metric for image captioning." arXiv preprint arXiv:2104.08718 (2021).
>
> [2] Xu, J., Liu, X., Wu, Y., Tong, Y., Li, Q., Ding, M., Tang, J., Dong, Y.: Imagereward: Learning and evaluating human preferences for text-to-image generation. Advances in Neural Information Processing Systems 36 (2024)
>
> [3] Hu, Y., Liu, B., Kasai, J., Wang, Y., Ostendorf, M., Krishna, R., Smith, N.A.: Tifa: Accurate and interpretable text-to-image faithfulness evaluation with question answering. arXiv preprint arXiv:2303.11897 (2023)
>
> [4] Liu, Nan, et al. "Compositional visual generation with composable diffusion models." European Conference on Computer Vision. Cham: Springer Nature Switzerland, 2022.

---

### Meta-Review · Area_Chair_Cqdg · 2024-12-17

**Metareview:**

This paper introduces a novel algorithm for combining multiple pre-trained diffusion models at inference time based on the principle of superposition of vector fields. The method achieves more diverse generation results, enhanced prompt adherence in image data, and improved protein structure design.

All reviewers agree that the work is interesting, grounded with theoretical justifications, and well-validated on protein generation problems.

**Additional Comments On Reviewer Discussion:**

During the rebuttal, the authors conducted more comprehensive investigations, especially on stable diffusions that addressed the reviewers' concerns.

---

### Decision · Program_Chairs · 2025-01-22

Accept (Spotlight)